



# On the consistency of methane isotopologue retrievals using TCCON and multiple spectroscopic databases.

Edward Malina[1], Ben Veihelmann[1], Dietrich G. Feist[2,3,4], and Isamu Morino[5]

[1]Earth and Mission Science Division, ESA/ESTEC, Keplerlaan 1, Noordwijk, the Netherlands.
[2]Lehrstuhl für Physik der Atmosphäre, Ludwig-Maximilians-Universität München, Munich, Germany
[3]Deutsches Zentrum für Luft- und Raumfahrt, Institut für Physik der Atmosphäre, Oberpfaffenhofen, Germany.
[4]Max Planck Institute for Biogeochemistry, Jena, Germany
[5]Satellite Remote Sensing Section and Satellite Observation Center, Center for Global Environmental Research, National Institute for Environmental Studies, Onogawa 15-2, Tsukuba, Japan.

**Correspondence:** Edward Malina (edward.malina.13@alumni.ucl.ac.uk)

**Abstract.** In this study we perform retrievals of the two main methane isotopologues $^{12}CH_4$ and $^{13}CH_4$ using measurements from the Total Carbon Column Observing Network (TCCON) from two sites, namely Ascension Island in the Atlantic Ocean and Tsukuba, Japan.

Using the TCCON GGG2014 retrieval environment retrievals are performed using four separate spectroscopic databases
and a set of spectral fit windows. Databases used include the TCCON spectroscopic database; the HITRAN2016 database; the GEISA2015 database; and the ESA SEOM-IAS database. We assess the retrievals using standard TCCON methane windows, and specific windows (in the 4190-4340 cm$^{-1}$ range) based on the sensitivity of the instruments TROPOMI present on Copernicus Sentinel-5P (S5P) and the future Sentinel 5 (S5) mission present on MetOp-SG. We assess the biases in retrieving methane isotopologues using these different spectral windows and different spectroscopic databases. The sensitivity of these
retrievals (across windows and databases) to errors in the a priori information, specifically pressure, temperature, methane and water vapour are also assessed.

We find significant biases between retrievals calculated using differing spectroscopic databases and windows for both methane isotopologues, with up to a 3% bias between $^{12}CH_4$ retrievals and 20% bias in $^{13}CH_4$ retrievals. Retrievals using the 4190-4340 cm$^{-1}$ spectral range show the results with the least variation between spectroscopic databases, and we there-
fore recommend that this band should be used in future TCCON methane retrievals. Results obtained ng with the SEOM-IAS database show the lowest fit residuals. Uncertainty on $^{13}CH_4$ retrievals are relatively high ($0.1 – 2$ ppb, combination of systematic and random). The sensitivity to a priori assumptions are shown to be significant for both $^{12}CH_4$ and $^{13}CH_4$. Uncertainty in the pressure cross sections is shown to be the most significant, with variations across all spectroscopic databases and spectral windows.

*Copyright statement.* TEXT





## 1   Introduction

Methane is widely acknowledged to have a significant impact on the global climate (IPCC, 2014), but the processes via which it enters and is removed from the atmosphere are still not as widely understood as carbon dioxide, with bottom up estimations not agreeing with top down estimations (Kirschke et al., 2013; Saunois et al., 2019). This disconnect has led to the development

of multiple satellite missions, with the aim of developing the knowledge of the global methane budget.

The recent launch of the S5P satellite, with the TROPOspheric Ozone Monitoring (TROPOMI) instrument (Veefkind et al., 2012), and the future S5 mission with its Ultra-Violet Near infrared Shortwave infrared (UVNS) instrument (Ingmann et al., 2012), represent a significant advancement in space-based Greenhouse Gas (GHG) retrievals.

TROPOMI and UVNS exploit the $4190 - 4340$ cm$^{-1}$ spectral range, which has not been explored in any great depth from

space-based instruments for methane retrievals. The Scanning Imaging Absorption spectrometer for Atmospheric Cartogra-pHY (SCIAMACHY) (Bovensmann et al., 1999) onboard ENVISAT was sensitive to this spectral range, but was plagued with detector issues (ice build-up) and was overlooked in favour of other windows. The Measurements Of Pollution In The Tropo-sphere (MOPITT) (Drummond and Mand, 1996) is also sensitive to this spectral range, but is also affected by technical issues and has never successfully retrieved methane in this spectral window. In addition, the wide spectral sensitivity of the limb

viewing Canadian Atmospheric Chemistry Experiment (ACE)-FTS (Bernath et al., 2005) includes this spectral window, but again the methane products of ACE-FTS do not include retrievals in this window. S5P/TROPOMI and S5/UVNS will therefore be relying on spectroscopy that has limited assessment from space-based instruments (Checa-Garcia et al., 2015; Galli et al., 2012). There may be other satellite instruments sensitive to this window, but there are no publicly available methane products from this window. This extends to TCCON, which although sensitive to this spectral range, has primarily provided its methane

abundances in the $6000$ cm$^{-1}$ spectral range.

In this study, we make retrievals of the two main methane isotopologues using the TCCON GGG2014 (Toon, 2015) retrieval environment. Spectra are taken from two different TCCON sites and over multiple seasons, in order to introduce atmospheric variations. We assess the differences in abundances of the isotopologues, the retrieval errors and the quality of the fits when retrieved from the standard TCCON spectral windows, and methane spectral windows in the TROPOMI/UVNS spectral range.

We will also quantify the variations in retrieval abundances when using four separate spectral databases, and the application of non-Voigt line broadening shapes. We briefly investigate some of the differences in key spectroscopic parameters w.r.t. to each of the spectroscopic databases. The sensitivity of the retrievals to errors introduced into the a priori data are assessed, allowing for the assessment of how differing windows and spectroscopic databases are sensitive to these errors.

TCCON is a global network of 27 ground based Fourier Transform Spectrometers (FTS) (Wunch et al., 2010), with the

primary aim of providing reference total column abundances of numerous atmospheric species calibrated against aircraft pro-files (Wunch et al., 2010, 2011), including methane, for validation and cross-calibration purposes. TCCON operates in a wide spectral range ($4000 - 15000$ cm$^{-1}$) and takes measurements in the solar occultation configuration. TCCON is currently one of the key sources of reference data for the validation of satellite-based GHG retrievals (Yoshida et al., 2011; Crisp et al., 2012; Parker et al., 2015). Examples include the Orbiting Carbon Observatory (OCO)-2 and the Greenhouse Gases Observing



Satellite (GOSAT) (Kuze et al., 2009). TCCON instruments have both high spectral resolution (0.02 cm$^{-1}$), and high Signal
to Noise ratios (SNR) due to solar direct viewing geometry. TCCON and TROPOMI/UVNS both have overlapping spectral
windows in the Shortwave Infrared (SWIR) methane absorption regions, highlighted in Table 1.

**Table 1.** Methane SWIR windows commonality between S5P/S5 and TCCON

| Methane window | S5P/TROPOMI | S5/UVNS | TCCON |
|---|---|---|---|
| 5970-6289 cm$^{-1}$ | N | Y | Y |
| 4190-4340 cm$^{-1}$ | Y | Y | Y |

TCCON methane products generated using the standard TCCON windows will be compared with TROPOMI/UVNS prod-
ucts obtained using the $4190 - 4340$ cm$^{-1}$ window, therefore potential biases associated with the choice of fit windows needs
to be quantified and understood. Indeed, if the $4190 - 4340$ cm$^{-1}$ window proves to be accurate/stable, then there is justifi-
cation to integrate TCCON retrievals from this window into future TCCON retrieval products. Building on potential differ-
ences in methane retrievals from different windows, numerous algorithms will be used to provide methane data products from
TROPOMI/UVNS (e.g. (Hu et al., 2016; Schneising et al., 2019)), all of which will potentially be based on differing spec-
troscopic databases and therefore subject to differing biases. The high SNR and high spectral resolution makes TCCON data
an excellent resource to perform retrievals of methane isotopologues, and assess any potential variations due to differences in
the spectroscopic databases, building on examples of similar past studies (Checa-Garcia et al., 2015; Galli et al., 2012). By
investigating the biases present in a TCCON site we can comment on some of the potential spectroscopic related biases in
satellite retrievals, based on variations in the spectroscopic source. For example whether or not these are persistent over time
and location, and the stability of these biases depending on local temperature and pressure variations. We therefore hope to
inform as to the potential source of biases related to ongoing TROPOMI validation, and future S5/UVNS validation. We also
will make recommendations on what spectral windows and databases should be used for methane retrievals going forward.

TCCON methane products are the result of a standardised process where an average of three retrieved values from three
TCCON fit windows (described in Table 2 below) are presented as the TCCON methane products. Assessments of the biases
present in these windows, with respect to the spectroscopic databases may help inform the future of TCCON methane, if
updates to spectroscopy are currently in consideration.

In addition to assessing the window and spectroscopic source biases for the two main methane isotopologues, the opportunity
is taken to calculate the $\delta^{13}$C metric. This is a metric which has been used in numerous studies globally to differentiate
methane source types (Fisher et al., 2017; Nisbet et al., 2016; Rigby et al., 2017; Rella et al., 2015), e.g. industrial or wetlands.
Calculating total column values of this metric would be highly beneficial towards understanding the global methane budget.
From TCCON it would be possible to constrain the regional origin of methane, or understand variations in the hydroxyl methane
sink in the troposphere. Calculation of the $\delta^{13}$C metric requires the concentration of the two main methane isotopologues $^{12}$CH$_4$
and $^{13}$CH$_4$, which make up roughly 98% and 1.1% of global atmospheric methane respectively. Almost all measurements of
this metric are limited to in situ studies or airborne flask measurements, which although highly accurate, by their nature are





spatially limited. Some effort has gone into satellite based retrievals (Buzan et al., 2016; Weidmann et al., 2017; Malina et al.,

2018, 2019), but the results of these studies show this to be a challenging task. Therefore the calculation of the $\delta^{13}C$ metric is a target of secondary importance in this study. Nevertheless, we assess if the $\delta^{13}C$ metric can be derived from TCCON measurements, how the $\delta^{13}C$ metric varies depending on the spectral window and spectroscopic database and if the results can be interpreted in a meaningful way.

This paper is structured as follows, Section 2 outlines the methods used in this study, including details about the TCCON

sites and spectra used, as well as the retrieval method. Information about the spectroscopic databases used in this study are also given. The results of this study based on the methods in section 2 are shown in section 3. Section 4 outlines an assessment of the sensitivity of the retrievals to introduced errors in the a priori data. Section 5 discusses the results shown in sections 3 and 4, and conclusions are shown in section 6.

## 2 Methods, tools, datasets and requirements

### 2.1 TCCON spectra and tools

We use TCCON spectra from two different sites, firstly the Ascension Island site, found in the middle of the Atlantic Ocean near the equator. The second site is the Tsukuba site, near Tokyo in Japan, and at a higher latitude than Ascension Island. Ascension Island has an arid climate with a little precipitation, which remains largely constant through the year and does not have designated seasons but is subject to some variation. While Tsukuba is subject to seasonal effects, with hot wet

summers and cold dry winters. These two sites represent a wide range of atmospheric conditions, the Tsukuba spectra are generally captured in a narrow spread of solar zenith angles (SZA), typically 35°<SZA<60°, while Ascension Island spectra are captured under a much wider range of SZA, typically 10°<SZA<90°. This means that there will be larger variations in SNR for Ascension Island than Tsukuba, therefore leading to higher levels of radiometric noise and an increase in the random error, which could be significant for $^{13}CH_4$ retrievals.

In this study we use the GGG2014 environment which includes the GFIT retrieval algorithm (Wunch et al., 2010), the standard algorithm used by the TCCON sites for processing and distributing trace gas column abundances. GFIT is described in detail in Wunch et al. (2010), and is summarised briefly, GFIT employs a nonlinear least-squares optimal estimation scheme based on (Rodgers, 2000). A forward model (radiative transfer model) is used to calculate synthetic irradiance spectra based on a set of fit parameters (state vector elements) and model parameters. This is then fit to the measured irradiance spectrum in

order to provide an optimal result, normally a trace gas abundance. In the case of GFIT the state vector includes the following.

- first target gas scaling factor (desired output).

- interfering gas scaling factor.

- continuum level of the irradiance spectrum.

- continuum tilt





– continuum curvature

– frequency shift

– zero level offset

– solar scaling (differences in shifts of atmospheric and solar lines)

– fit channel fringes

Note that while all of the above can be included in the GFIT state vector, not all are routinely included. The continuum curvature especially is not commonly included in the state vector, since the option is designed to remove instrument features, but may also attempt to remove other effects due to the spectroscopic database, as noted in the TCCON wiki (TCCON). GFIT assumes a fixed profile shape for all trace gases, and the sub-column amount for each altitude/pressure level are not independently scaled. Unlike in most satellite retrieval algorithms, aerosol and albedo terms are not included in the state

vector, this is because TCCON operates in solar direct viewing, where scattering is considered unimportant and surface terms are not necessary. The retrieved column amount scaling factors are multiplied by the a priori vertical column abundances in order to determine the retrieved vertical column. Dry air Mole Fractions (DMF) are calculated by dividing the scaled trace gas column with the total column $O_2$, retrieved from a wide window in the 7885 $cm^{-1}$ (1286 nm) spectral range multiplied by the volume mixing ratio of $O_2$ 0.2095.

Because of the high spectral resolution of the TCCON instruments (0.02 $cm^{-1}$), all spectral lines are resolved. Radiative transfer calculations are performed on a line-by-line basis. GGG includes a spectroscopic database in its environment, which is similar to other more widely adopted databases (see below). TCCON has a standard set of spectral windows for methane retrievals, all of which are in the 6000 $cm^{-1}$ methane absorption window range. In this study we include the TROPOMI/UVNS SWIR spectral windows (4190-4340 $cm^{-1}$). This window along with a description of all of the windows considered in this

study are described in Table 2 below.

**Table 2.** Spectral windows used in study.

| Window | Window spectral range ($cm^{-1}$) | Target species | Background species | Window source |
|---|---|---|---|---|
| 1 | 4190-4340 | $^{12}CH_4$ | $^{12}CH_4$, $CO_2$, $H_2O$, HDO, CO, HF, $N_2O$, $O_3$ | Sentinel 5 baseline |
| 2 | 5880-5996 | $^{12}CH_4$ | $^{13}CH_4$, $CO_2$, $H_2O$, $N_2O$ | TCCON standard |
| 3 | 5996.45-6007.55 | $^{12}CH_4$ | $^{13}CH_4$, $CO_2$, $H_2O$, $N_2O$, HDO | TCCON standard |
| 4 | 6007-6145 | $^{12}CH_4$ | $^{13}CH_4$, $CO_2$, $H_2O$, $N_2O$, HDO | TCCON standard |
| 5 | 4190-4340 | $^{13}CH_4$ | $^{12}CH_4$, $CO_2$, $H_2O$, HDO, CO, HF, $N_2O$, $O_3$ | Sentinel 5 baseline |
| 6 | 6007-6145 | $^{13}CH_4$ | $^{12}CH_4$, $CO_2$, $H_2O$, $N_2O$, HDO | TCCON standard |

Windows 2-4 are standard TCCON methane retrieval windows which in this study are used for $^{12}CH_4$. Window 6 is as window 4, but with the target species of $^{13}CH_4$, note that $^{13}CH_4$ is not a standard TCCON product, and there are no standard





TCCON windows for this isotopologue. We therefore decided to use just one wide spectral window where known positions of $^{13}CH_4$ spectral lines exist. Windows 1 and 5 are based on the TROPOMI spectral window identified in Galli et al. (2012); Hu

et al. (2016), given that no standard windows exist in this spectral window for TCCON.

### 2.1.1  Spectroscopic Databases

The introduction of $^{13}CH_4$ into spectroscopic databases in the TROPOMI spectral region is relatively recent, and in the case of HITRAN, was only introduced in the 2012 release. Indeed Gordon et al. (2017) reports that numerous new $^{13}CH_4$ lines were introduced into the latest HITRAN2016 release, this implies that the spectroscopy of $^{13}CH_4$ in this region is not yet settled. A

review of the documents released with numerous spectroscopic databases (e.g. (Gordon et al., 2017; Jacquinet-Husson et al., 2016; Birk et al., 2017) suggest that $^{13}CH_4$ are not all sourced from the same laboratory studies. We therefore decided to compare methane isotopologue retrieval from four separate spectroscopic databases, which are as follows: 1) the database included with GGG2014 (Toon, 2015), which currently assumes a Voigt line shape for all lines. 2) HITRAN2016, HITRAN is a well-established spectroscopic database that has been used in numerous satellite based studies previously (e.g. Galli et al.

(2012)). The current release HITRAN2016 (Gordon et al., 2017) has been revised from the previous release (HITRAN2012) in terms of methane, with new lines and parameters included for both of the main isotopologues. HITRAN2016 does include the additional parameters required to model non-Voigt lines shapes, however the current version does not include these parameters for methane (at the time of writing). 3) The GEISA2015 database (Jacquinet-Husson et al., 2016) is another spectroscopic database, similar in design and goals to the HITRAN databases. The GEISA database does not currently include non-Voigt line

shape parameters. 4) SEOM-IAS (Birk et al., 2017), specifically developed for the TROPOMI spectral window and designed around non-Voigt atmospheric line shape profiles. This database only has data within the 4190-4340 cm$^{-1}$ spectral range, and can therefore only contribute to windows 1 and 5 of this study.

Some work has been performed previously comparing spectroscopic databases e.g. (Jacquinet-Husson et al., 2016; Armante et al., 2016), but this study is the first case with respect to the TROPOMI spectral window with TCCON.

### 2.1.2  Voigt vs non-Voigt line shape profiles

Ngo et al. (2013) states that the standard Voigt profiles used for spectral line broadening may be inadequate for trace gas retrievals (based on laboratory studies), which can lead to errors larger than instrument precision requirements. In order to calculate more accurate line shapes for remote sensing purposes, numerous models have been proposed. In this paper we use the quadratic Speed Dependent Hard Collision (qSDHC) model introduced in Ngo et al. (2013); Tran et al. (2013). This model

includes additional parameters based on speed dependence of collisional broadening and velocity changes of molecules due to collisions, on top of the standard parameters of pressure-induced air broadening, and pressure induced line shift. Note that only the SEOM-IAS database use these additional parameters, the remaining spectroscopic databases do not include these parameters for methane at the time of this paper. We use the FORTRAN routines provided with Ngo et al. (2013) to implement the qSDHC model into the GFIT algorithm, modified to include first order Rosenkranz line mixing effects. Mendonca et al.

(2017) report that incorporating speed dependent and line mixing has a significant effect on calculated methane columns when





compared against assuming Voigt dependency. They find a 1.1% difference in total methane column abundances from 131,124 spectra. The implication being that it is important to account for the additional physical parameters included in non-Voigt models, when retrieving methane.

## 2.2 Metrics

Our main assessment metrics in this study are as follows.

- Averaging Kernels (AK): the AKs capture the sensitivity of the retrieved state vector to the truth, and is defined as $\mathbf{A} = \partial \hat{\mathbf{x}} / \partial \mathbf{x}$, where $\hat{\mathbf{x}}$ is the retrieved state vector and $\mathbf{x}$ is the truth. AKs are typically used in satellite and ground-based remote sensing to characterise the vertical sensitivity profile of a retrieval.

- Transmission spectra.

- RMSE of the residual between the calculated transmission spectra, and the TCCON measurement transmission spectra, expressed as the Root Mean Square Error (RMSE).

- The quality of the fit, expressed via the $\chi^2$ test, quantitatively defined as:

$$\chi^2 = \sum_i [\mathbf{y_{measured}} - \mathbf{y_{calculated}}]^2. \tag{1}$$

Where $\mathbf{y_{measured}}$ refers to the measured TCCON spectrum, and $\mathbf{y_{calculated}}$ is the synthetic spectrum calculated by the
forward model.

- A posteriori error

- Standard deviation of the DMF for each window ($\sigma_{window}$).

- Standard deviation of the DMF between all windows for a specific database ($\sigma_{inter-window}$)

- Bias (b) of the retrieved mean of the DMF for each window against the retrieved mean of the equivalent window
using the TCCON spectroscopic database, which is taken as the reference in the present study due to its pedigree in validations for satellite missions.

- A posteriori error

- Total uncertainty in the retrieved abundances of the methane isotopologues, including systematic and random errors. Wunch et al. (2010) states that systematic errors typically dominate for TCCON retrievals.

- $\delta^{13}$C: Methane isotopologues abundances are typically expressed in the form of the following metric.

$$\delta^{13}C = \left( \frac{(^{13}CH_4/^{12}CH_4)_{sample}}{VPDB} - 1 \right) \times 1000\%o, \tag{2}$$



where VPDB refers to Vienna Pee Dee Belemnite, an international reference standard for 13C assessment. Tropospheric methane typically exhibits a $\delta^{13}$C value of roughly -47‰ (Rigby et al., 2017), and total column measurements from TCCON are unlikely to deviate from this value to a significant degree. Therefore this tropospheric $\delta^{13}$C value acts as a useful proxy,

to determine the stability and variability associated with retrievals of methane isotopologues from different spectral windows, spectroscopic databases, location and time using the tropospheric $\delta^{13}$C value as a baseline.

In an ideal scenario we would compare our results with some reference results, however we are currently unaware of total column $^{13}CH_4$ retrieval data. We therefore perform our comparisons with respect to the TCCON spectroscopic database, under the assumption that biases are already present, which can be assessed at a later date if there is benefit to doing so.

## 2.3   Analysis criteria

There are two key aspects to this study, the primary aspect is an assessment of the biases between spectral windows and spectroscopic databases w.r.t the two main methane isotpologues. TCCON typically aims for precision of <0.3% on methane retrievals , and has a rough estimate of 1% systematic uncertainties (dominated by in-situ calibration which vary depending on site (Wunch et al., 2015)). Therefore it is possible to judge the variations of the $^{12}CH_4$ between windows and databases based

on these biases and precisions. In terms of $^{13}CH_4$, there are no published precision and accuracy requirements or statistics with TCCON. We therefore assume precision and accuracy values of carbon monoxide as an appropriate proxy, given that CO has similar DMF value to $^{13}CH_4$ in the lower atmosphere. TCCON aims for precision of <1% on carbon monoxide retrievals, and roughly 7.5% systematic uncertainties (Wunch et al., 2015).

In order to judge inter-window/spectroscopic database biases, we compare the relative difference of the retrievals with

respect to window 4 of the TCCON spectral database, henceforth described as the 'reference value'. We choose this window because it is the most commonly used in space based retrievals at this time (e.g. GOSAT), and the TCCON spectral database as it is the most established with TCCON retrievals. The relative difference is calculated as the difference between the retrieval and the reference value, divided by the reference value.

Secondly the $\delta^{13}$C value has been used to differentiate between methane source types (Fisher et al., 2017; Nisbet et al., 2016;

Rigby et al., 2017; Rella et al., 2015), and variations of this value has been linked with variations in the global methane budget (Rigby et al., 2017; Mcnorton et al., 2016). Therefore if TCCON sites could accurately resolve the $\delta^{13}$C value, there would be significant benefit to the GHG community. Several studies (e.g. Nisbet et al. (2016); Rigby et al. (2017); Weidmann et al. (2017); Malina et al. (2018, 2019)) all show that an uncertainty of <1‰ in the $\delta^{13}$C metric is required in order to determine natural annual variability. This roughly equates to achieving a total uncertainty of <0.02 ppb on $^{13}CH_4$ retrievals, or roughly

0.1% of the total column. This is clearly an unrealistic target for individual retrievals, given the uncertainty requirements for carbon monoxide described above. Nevertheless precision errors will be low due to the nature of TCCON, and through the fact that TCCON sites are situated in a fixed position, allowing for long term averaging to reach a required precision target. Therefore one of the key aims of this study is to identify how far away TCCON uncertainty is from the desired uncertainty of <1‰ $\delta^{13}$C.





### 2.4 Sensitivity analysis

#### 2.4.1 Spectroscopic differences

Section 2.1.1 introduces the various spectroscopic databases used in this study. Differences in line intensity and the lower state energy (both of these variables being temperature and pressure dependent) are assessed through comparing the max, min, and mean/total values of both of the variables. This potentially allows us to understand the differences in TCCON retrievals caused by variations between the spectroscopic databases.

#### 2.4.2 A priori error

Malina et al. (2019) reports that retrievals of $^{13}CH_4$ are highly sensitive to errors in a priori temperature and pressure. This has led to the inclusion of the sensitivity of TCCON $^{13}CH_4$ retrievals to a priori errors into this study. Building on any potential spectroscopic differences outlined in sect. 2.1.1, Equation (3) below suggests that if there are significant differences between the spectroscopic databases, most notably in the lower state energy level, then the temperature and pressure dependency uncertainty of the retrievals will vary depending on the database. Thus implying that the bias between spectroscopic databases can vary depending on spatial and temporal conditions. In order to quantify the effect of a priori errors, we use linear regression to compare the retrieved concentrations of $^{12}CH_4$ and $^{13}CH_4$ from cases where a priori errors have been introduced, and the original unperturbed cases.

The a priori atmospheric data for the GFIT algorithm is based on two sources; the pressure, temperature and humidity data are drawn from the National Centers for Environmental Prediction (NCEP)/National Center for Atmospheric Research models (NCAR). In addition to these profiles, the trace gas profiles are built from empirical models developed from a combination of data from atmospheric balloon borne sensors and from the satellite instrument Atmospheric Chemistry Experiment-FTS (Wunch et al., 2010, 2011).

A priori profiles derived from different models can vary e.g. (Rahpoe et al., 2013). Based on these types of examples, we investigate the sensitivities to the following profiles.

- A priori methane profile.

- A priori water vapour profile.

- Pressure.

- Temperature.

For methane we assume 2% uncertainty on the total methane column. This applies to both the $^{12}CH_4$ and $^{13}CH_4$ profiles. Aside from methane, water vapour is the main dominating trace gas in the spectral windows identified in Table 2. We therefore investigate the effects of imprecise knowledge of the water vapour column, and in this study we assume a 10% uncertainty.

We note that GFIT retrieves methane and water vapour, but while pressure and temperature are model parameters of the forward model, they are not included in the state vector. This means that dependencies on pressure and temperature will not be





removed in the retrieval process. A 2% error into the a priori pressure profile is introduced, and for GFIT retrievals, pressure errors can affect methane retrievals in two ways. The first is through the retrieval of $O_2$ which is used to convert the total column concentration of methane into DMFs. The second is through pressure dependence of spectroscopic absorption. Finally a 2 K error is introduced into the total column a priori profile for temperature. As with pressure, errors are introduced through

the spectroscopic cross sections, Eq (3) describes the temperature dependency of the line intesity (e.g. An et al. (2011)).

$$\frac{S(T)}{S(T_0)} = \frac{Q(T_0)}{Q(T)} exp(-\frac{hcE_0}{k}(\frac{1}{T} - \frac{1}{T_0})), \tag{3}$$

where $S(T)$ is the line intensity at temperature T, $Q(T)$ is the total partition function of the absorbing molecule at temperature T, $S(T_0)$ and $Q(T_0)$ are as before but at temperature $T_0$, $E_0$ is the lower state energy, and h, c and k are constants.

In this work the results from the sensitivity to temperature biases are shown in the main text. However the results from the

other values of interest (pressure etc) are shown in Appendix C in order to keep the text concise.

## 3   Results

### 3.1   Averaging kernels

Figure 1 shows example column averaging kernels for retrievals of $^{12}CH_4$ and $^{13}CH_4$ using windows 1, and 4-6 from selected observations at both Tsukuba and Ascension Island.





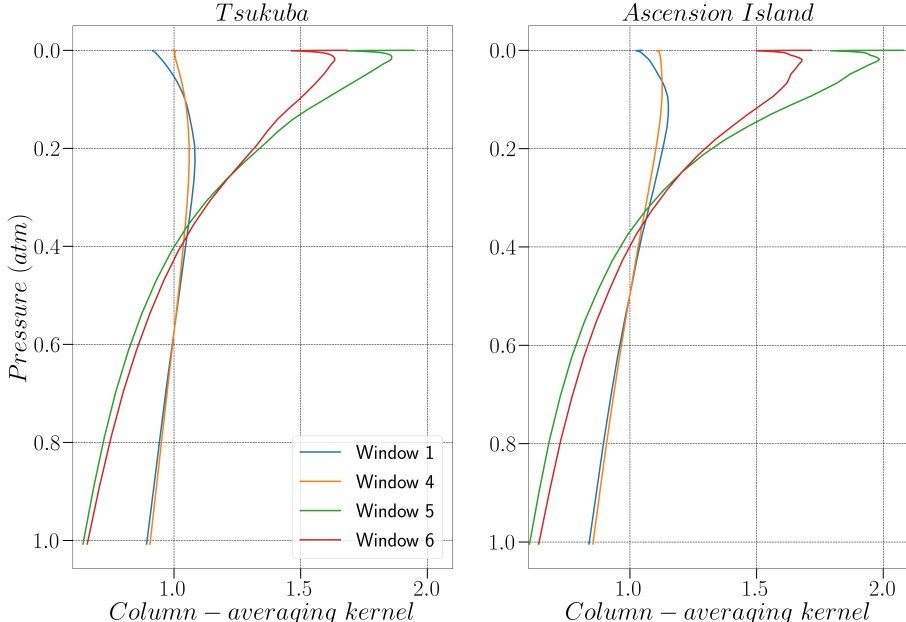

**Figure 1.** Column averaging kernels for typical retrievals of $^{12}CH_4$ and $^{13}CH_4$ from the Tsukuba TCCON site (left) and Ascension Island TCCON site (right) using the internal TCCON spectral database. The legend indicates the spectral window, for which the averaging kernels were calculated.

The $^{12}CH_4$ averaging kernels for both windows and both sites show little variation, and are similar to the $CH_4$ averaging kernels shown in (Wunch et al., 2011). The $^{13}CH_4$ averaging kernels show larger variation in the upper atmosphere, especially in the Ascension Island case, however this is not significant given the low concentration of $^{13}CH_4$ in the upper stratosphere. The shape of $^{13}CH_4$ averaging kernels is very similar to the shape of averaging kernels of CO from TCCON (Wunch et al., 2011), for all cases analysed in the present study. The similarity of the averaging kernels for the different windows shown

in Fig 1 shows that the total columns retrieved from different fit windows can be compared directly, and that biases between windows can be attributed to other sources.





## 3.2 Transmission fit accuracy

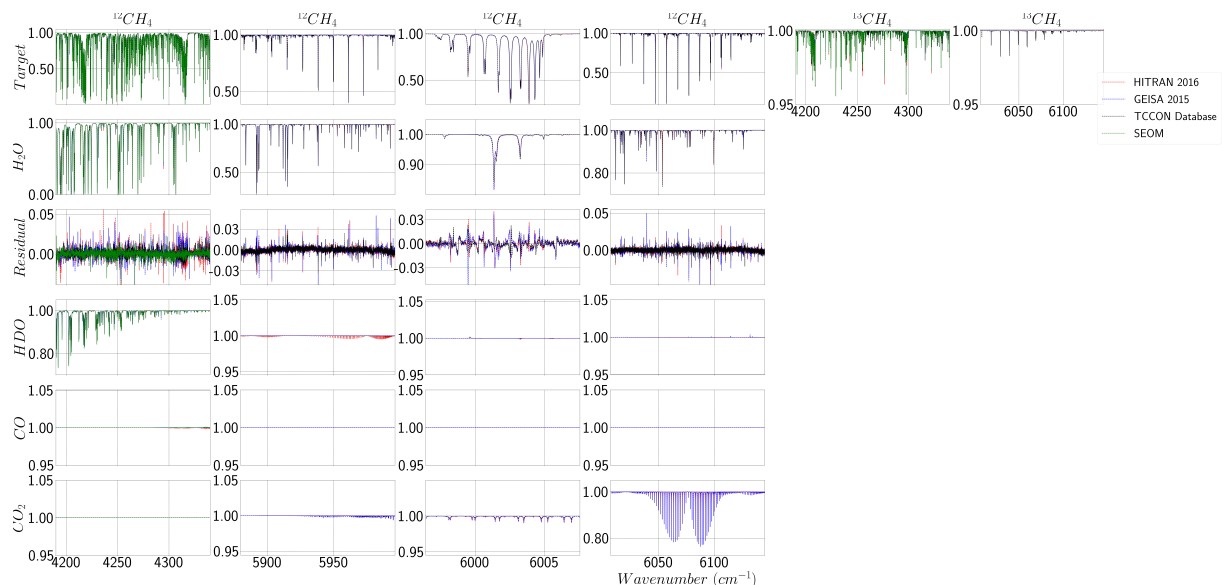

**Figure 2.** Example transmission spectra calculated from retrievals of $^{12}CH_4$ and $^{13}CH_4$ from the Tsukuba site in April 2016. Each column represents the transmission for a specific window, with columns 1 to 6 as windows 1 to 6. The top row indicates the transmission of the target species ($^{12}CH_4$ or $^{13}CH_4$). The second row shows the transmission of $H_2O$, for window 5 & 6, see the transmission in windows 1 & 4. The third row shows the residual transmission between the measured and calculated transmissions. Following the residual, minor background trace gases are shown, row 4 is HDO, row 5 is CO and row 6 is $CO_2$. The red lines represent cross sections computed from HITRAN2016, blue for GEISA2015, green for SEOM-IAS and black for TCCON spectroscopic database.

Figure 2 shows the relative strengths of the absorption of the trace gases of interest in the selected spectral windows. For $^{12}CH_4$ and $H_2O$, we see that window 1 is a complex region with a large number of absorption lines including strong lines
that saturate in the centre, and pronounced spectral overlap of lines. Windows 2-4 all show high levels of absorption but less line mixing/overlapping lines. The absorption by $^{13}CH_4$ in the spectral windows 5 and 6 is weak; window 5 contains significantly larger number of spectral lines compared to window 6. Both HITRAN2016 and GEISA2015 show significant residual transmission peaks in all of the windows. Quantitatively, the transmission statistics listed in Table 3 indicate that the SEOM-IAS retrieval has the best fit in window 1, with the TCCON database showing similar values. Note that the fit
characteristics of window 1 are several times worse than any of the other windows explored in this study. We must take into account here that window 1 is wide, and that better fits could be obtained by splitting the window. Note that the column densities for all trace gases are fitted simultaneously, therefore changing the target gas for retrieval purposes (e.g. $^{12}CH_4$ or $^{13}CH_4$) does not change the fit residuals, hence this is why only the transmission value for $^{13}CH_4$ are shown for windows 5 and 6.





For comparison purposes, Fig A1 and Table A1 show the same fit parameters, as shown in Fig. 1 and Table 3, but for an

example of Ascension Island retrieval in October of 2016. The quality of fit is several times worse for the example of Ascension

Island spectra. The key point is the relative fit values between the windows and spectral databases, which are similar to those

shown in Fig 2 and Table 3. This implies that the differences observed between the windows and databases exist, irrespective

of time and location. Suggesting site and season are not major contributors to biases in window and spectroscopic database.

The example transmission spectra shown in Fig 2 were captured with a solar zenith angle of 43°, with a similar air mass to

other spectra captured on the same day, while the Ascension Island spectra were captured with a solar zenith angle of 22°,

with a similar air mass to the Tsukuba spectra.

This analysis also shows that the GEISA database does not include any $^{13}CH_4$ lines in window 5, meaning that there will be

no further analysis on this window w.r.t GEISA.

**Table 3.** Statistics for Fig 1, based on metrics identified in sect 2.2.

| | Window 1 | Window 2 | Window 3 | Window 4 |
|---|---|---|---|---|
| RMSE | TCCON: $4.438x10^{-3}$<br>HITRAN: $6.803x10^{-3}$<br>GEISA: $5.678x10^{-3}$<br>SEOM: $4.268x10^{-3}$ | TCCON: $3.076x10^{-3}$<br>HITRAN: $3.747x10^{-3}$<br>GEISA: $3.910x10^{-3}$<br>SEOM: nan | TCCON: $3.846x10^{-3}$<br>HITRAN: $5.392x10^{-3}$<br>GEISA: $6.01x10^{-3}$<br>SEOM: nan | TCCON: $2.680x10^{-3}$<br>HITRAN: $3.578x10^{-3}$<br>GEISA: $3.722x10^{-3}$<br>SEOM: nan |
| $\chi^2$ | TCCON: 0.392<br>HITRAN: 0.922<br>GEISA: 0.642<br>SEOM: 0.363 | TCCON: 0.146<br>HITRAN: 0.216<br>GEISA: 0.235<br>SEOM: nan | TCCON: 0.0218<br>HITRAN: 0.0414<br>GEISA: 0.0532<br>SEOM: nan | TCCON: 0.132<br>HITRAN: 0.235<br>GEISA: 0.254<br>SEOM: nan |

Note that the RMSE and $\chi^2$ values for the $^{13}CH_4$ retrievals are identical to those indicated for $^{12}CH_4$ in the same window,

and are therefore not repeated in Table 3.

### 3.3    Retrieval accuracy

Figure 3 shows a time series of 40 DMFs of $X^{12}CH_4$ from measurements made on the 1st of April 2016 at the Tsukuba TCCON

site. The top panel shows the time series over the course of the day for each spectral window and database in consideration. We

see here that the maximum bias in retrieved $X^{12}CH_4$ DMFs is roughly 50 ppb, between the HITRAN and GEISA 6002 cm$^{-1}$

windows. The statistics in Table 4 suggest that window 3 has the largest deviation in DMFs w.r.t. spectroscopic databases,

while window 1 has the lowest deviation. In general Table 4 suggests that there are significant variations in the retrieved DMFs

in both spectral windows and spectral databases. The middle panel in Fig. 3 reveals a clear and constant bias between the

reference values and the other windows, of up to 2%.

Table 4 suggests that window 1 shows the least variation, while window 3 has the most (largely driven by the GEISA

retrievals). However, the inter-window variation suggests the least variation from the GEISA retrievals, while the HITRAN

retrievals show the most. The bias values from the equivalent TCCON windows in general show the largest biases from the

GEISA retrievals. However window 1 from the SEOM-IAS database shows the largest bias in this regard.





Retrieval uncertainties shown in the bottom panel of Fig 3 suggest a typical range of between 5 and 10 ppb. With the GEISA

and HITRAN retrievals showing the highest errors. These errors are significantly lower than the persistent differences noted

between the windows, meaning that these biases cannot be attributed to random retrieval uncertainties, and are likely due to

differences in the spectroscopic databases.

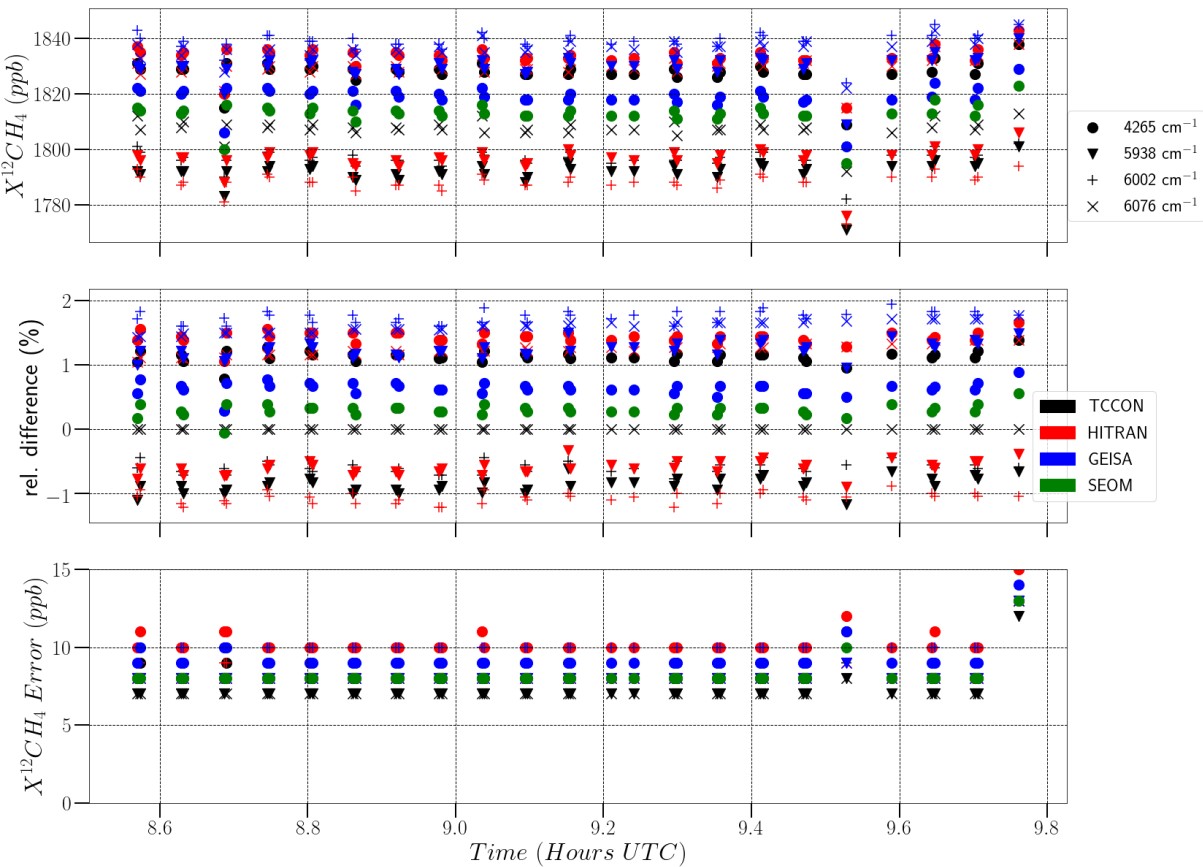

**Figure 3.** Retrieval time series for $X^{12}CH_4$ from the Tsukuba site on 01/04/2016. The top panel indicates the retrieved DMFs of $^{12}CH_4$ in

ppb for differing spectral windows and spectroscopic databases. The spectral windows are differentiated by line style, as shown in the legend

in the top left corner. The databases are differentiated by colour, as indicated in the top right hand corner. The middle panel shows the relative

difference of the retrievals with respect to retrievals from window 4 of the TCCON database in ppb, illustrating the persistent behaviour of

the spectroscopic-dependent and fit-window dependent differences. The bottom panel shows the total retrieval uncertainties in ppb.

Figure 4 shows the time series for DMFs of $X^{13}CH_4$ for the same time period as Fig 3. Nisbet et al. (2016) suggests that

a global background $\delta^{13}C$ value of -47‰ is typical, roughly equating to a range of between 18.6 and 19.3 ppb for $X^{13}CH_4$,

for the $X^{12}CH_4$ range represented in Fig 3. There is a significant range in retrieval values, with window 5 of HITRAN and

SEOM-IAS databases slightly underestimating the background $X^{13}CH_4$ value, unless the local air mass is significantly depleted

in $X^{13}CH_4$. Despite the differences in the retrieved DMFs between the windows, like in Fig 3, each window shows similar





characteristics in the time series. Table 4 indicates the metrics associated with $X^{13}CH_4$ retrievals, we note that HITRAN shows the lowest standard deviation between the windows. The closest agreement is between window 5 of TCCON and SEOM-IAS, which show just a 7 ppt bias.

The retrieval errors shown in the lower panel are noticeably lower for the window 5 retrievals (roughly x2) as opposed to the window 6 retrievals, but all are quite large (up to 25% of the total column in window 6). This scale of error could be reduced through long term averaging (i.e. days or weeks), but as Wunch et al. (2010) identifies, the majority of retrieval errors will be systematic in nature and cannot be removed through averaging.

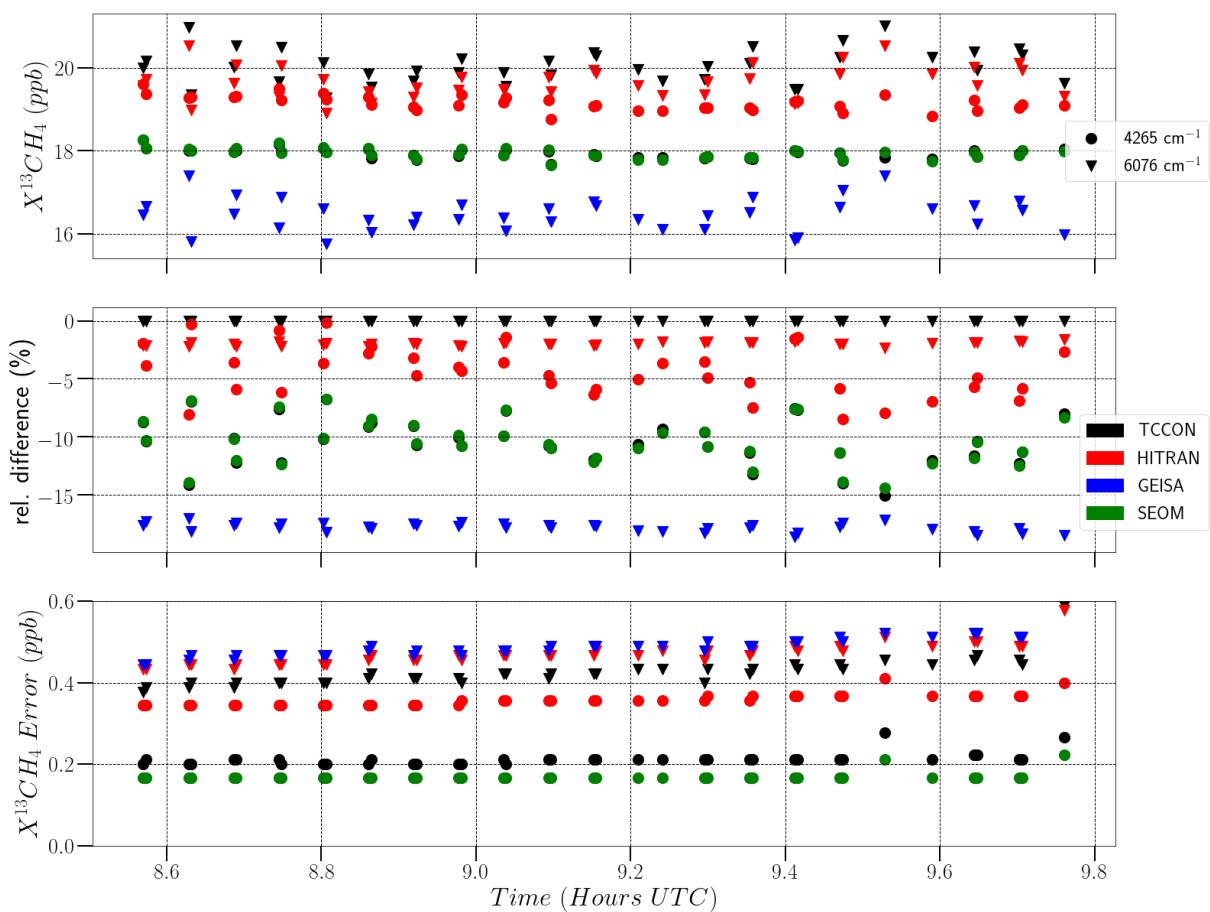

**Figure 4.** Retrieval time series for $X^{13}CH_4$ from the Tsukuba site on 01/04/2016. The top panel indicates the retrieved DMFs of $X^{13}CH_4$ in ppb for differing spectral windows and spectroscopic databases. The spectral windows are differentiated by line style, as shown in the legend in the top left corner. The databases are differentiated by colour, as indicated in the top right hand corner. The middle panel shows the relative difference of the retrievals with respect to retrievals from window 6 of the TCCON database in ppb. The bottom panel shows the total retrieval uncertainties in ppb.





**Table 4.** Statistics for Fig 3 and Fig 4 based on metrics identified in retrieval abundancies subsection of section 2.2.

| Window | 1 | 2 | 3 | 4 | 5 | 6 |
|---|---|---|---|---|---|---|
| $\sigma_{window}$ (ppb) | 8.98 | 17.5 | 22.5 | 12.6 | 0.589 | 1.64 |
| Database | | | TCCON | HITRAN | GEISA | SEOM-IAS |
| $\sigma_{inter-window}X^{12}CH_4$ (ppb) | | | 16.5 | 20.0 | 9.15 | N/A |
| $\sigma_{inter-window}X^{13}CH_4$ (ppb) | | | 1.09 | 0.389 | N/A | N/A |
| Database | HITRAN | | GEISA | | SEOM-IAS | |
| b (ppb; window 1) | 5.36 | | 8.90 | | 15.1 | |
| b (ppb; window 2) | 4.92 | | 38.1 | | N/A | |
| b (ppb; window 3) | 8.49 | | 42.4 | | N/A | |
| b (ppb; window 4) | 22.3 | | 28.9 | | N/A | |
| b (ppb; window 5) | 1.22 | | N/A | | 0.00655 | |
| b (ppb; window 6) | 0.394 | | 3.57 | | N/A | |

Figure 5 shows a time series of retrievals from the Ascension Island site on 01/10/2016. The results in this fig. show similar values to those shown in Fig. 3 in terms of the DMF magnitude from the windows and spectroscopic databases. This is highlighted by the $\sigma_{window}$ values shown in Table 5 which are similar to those indicated in Table 4. The $\sigma_{inter-window}$ values also show similar magnitudes to those indicated in Table 4, with HITRAN showing the largest intrawindow variation, and GEISA the lowest. In general the bias values are similar in Table 4 and Table 5, however there are some differences. There is

a significantly larger bias for window 2 in HITRAN, and lower for window 4 in HITRAN. Window 4 bias for GEISA is also significantly lower.

The total errors on the $X^{12}CH_4$ retrievals are significantly higher than those indicated in Fig 3, but the structure of the errors of the windows are similar to those shown in Fig 3. For example window 4 of TCCON shows the lowest total uncertainty, while window 1 of HITRAN shows the highest uncertainty. Apart from the strange behaviour of window 1 from the GEISA

database, which can be attributed to a single point. We note that the retrieval uncertainties increase across the time axis of Fig 5. As stated in the introduction, SNR decreases as SZA increases which is the case here.

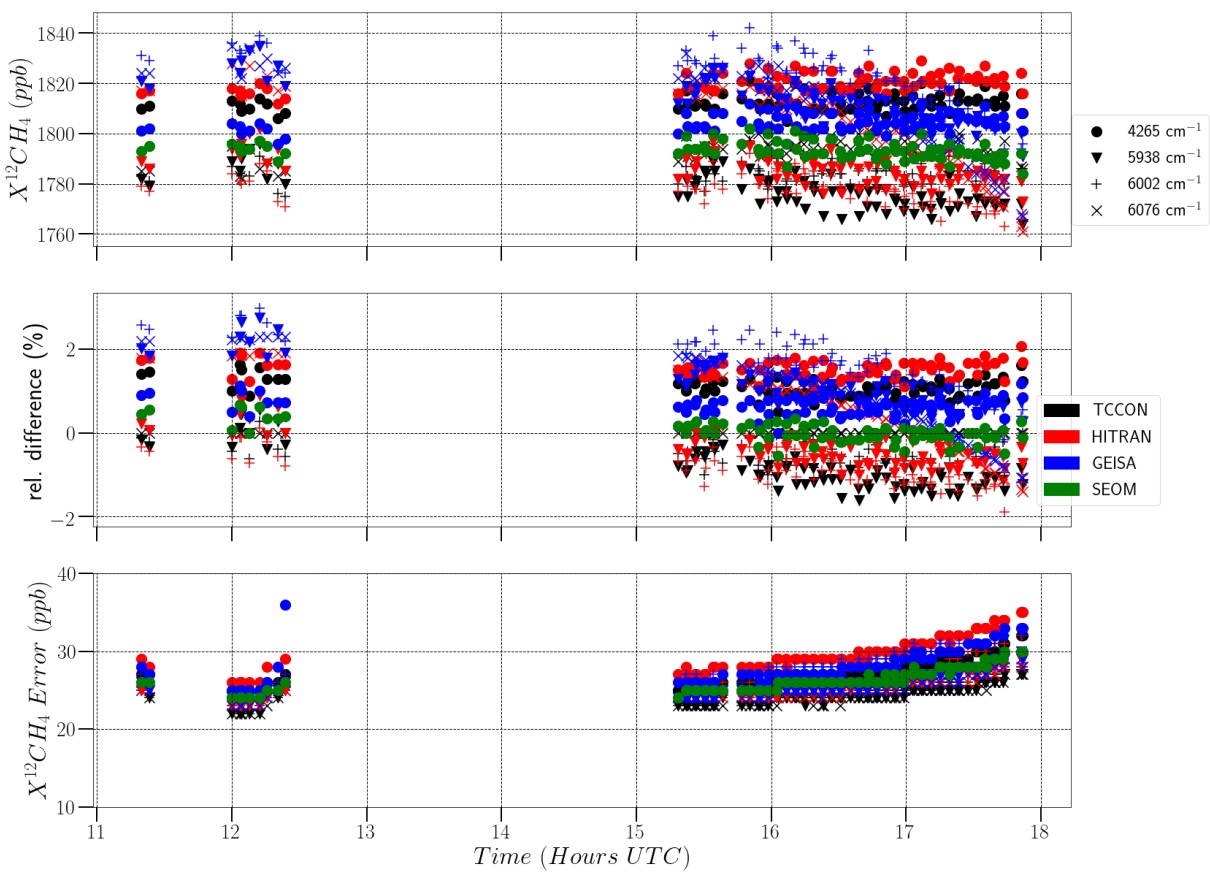

**Figure 5.** As Fig 3 but focused on 79 retrievals from the Ascension Island site on 01/10/2016.

The time series of $X^{13}CH_4$ retrievals shown in Fig 6 suggest close agreement with window 5 of the TCCON, HITRAN and SEOM-IAS databases, as indicated by the bias values of window 5 shown in Table 5. The DMFs of $X^{13}CH_4$ from these databases also show more realistic values, rather than window 6 of the retrievals. Which in addition to large over/underestimations

of the $X^{13}CH_4$ DMFs, show significant disagreement between the spectroscopic databases, as highlighted by the bias values in Table 5. The retrieval errors for window 5 are the lowest, and show similar patterns to those indicated in Fig 4. The overall magnitude of the retrievals errors are larger than that of the Tsukuba case, which matches the pattern shown in the $X^{12}CH_4$ errors shown in Fig 5.

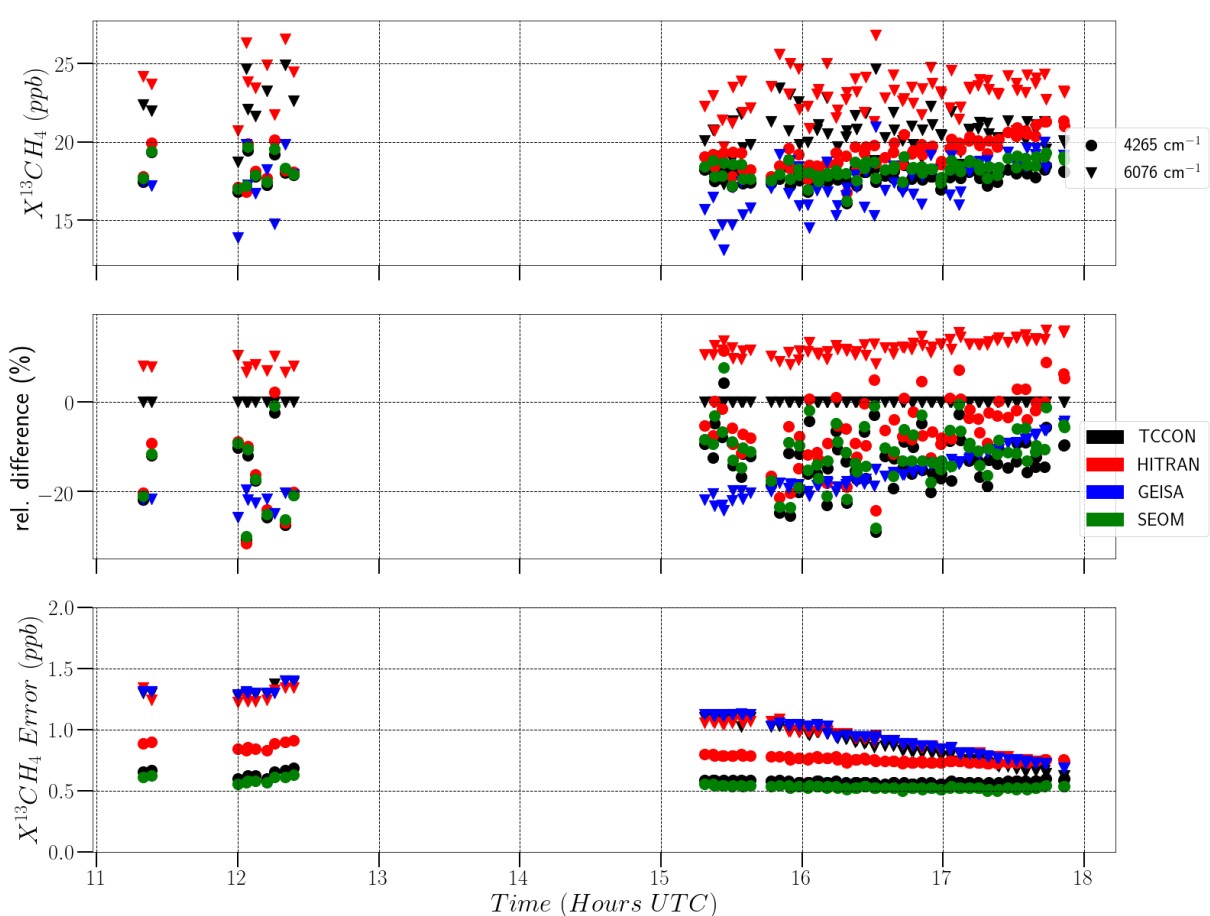

**Figure 6.** As Fig 4 but focused on 79 retrievals from the Ascension Island site on 01/10/2016.





**Table 5.** Statistics for Fig 5 and Fig 6 based on metrics identified in retrieval abundancies subsection of section 2.2.

| Window | 1 | 2 | 3 | 4 | 5 | 6 |
|---|---|---|---|---|---|---|
| $\sigma_{window}$ (ppb) | 10.6 | 16.6 | 21.2 | 14.8 | 0.981 | 2.79 |
| Database | | | TCCON | HITRAN | GEISA | SEOM-IAS |
| $\sigma_{inter-window}X^{12}CH_4$ (ppb) | | | 16.3 | 19.4 | 11.1 | N/A |
| $\sigma_{inter-window}X^{13}CH_4$ (ppb) | | | 1.83 | 2.30 | N/A | N/A |

| Database | HITRAN | GEISA | SEOM-IAS |
|---|---|---|---|
| b (ppb; window 1) | 7.76 | 8.47 | 19.2 |
| b (ppb; window 2) | 9.15 | 35.4 | N/A |
| b (ppb; window 3) | 7.41 | 37.6 | N/A |
| b (ppb; window 4) | 10.6 | 16.7 | N/A |
| b (ppb; window 5) | 1.45 | N/A | 0.414 |
| b (ppb; window 6) | 2.45 | 3.38 | N/A |

In addition to the retrievals presented in Fig 3, Fig 4, Fig 5 and Fig 6. We also present additional retrievals examples in Fig

B1, Fig B2, Fig B3 and Fig B4. Where Fig B1 and B2 show $X^{12}CH_4$ and $X^{13}CH_4$ retrievals from Tsukuba in July of 2016, and Fig B3 and B4 show retrievals from Ascension Island in August of 2016. Comparing the $X^{12}CH_4$ retrievals from all figures and tables, except for the bias values of window 3 and 4 of HITRAN which show more variable values. We note that relative difference of the DMFs are similar between the figures for the same spectral window/database, apart from a small increase in magnitude (roughly 0.1%). Similar responses can be seen when comparing Fig 5 and Fig B1, with notable exceptions on the

beginning of the time series of these figures.

For $X^{13}CH_4$ we see little variation between all figures, with small biases in window 5 between HITRAN, TCCON and SEOM-IAS. However, large variations are seen between all of the databases in window 6, with similar patterns in the magnitude variation. Suggesting that the errors are systematic and spectroscopic based, as opposed to seasonally or site based.

### 3.4   Calculation of $\delta^{13}C$ values

Based on Eq 2, we can calculate the $\delta^{13}C$ values for both Tsukuba and Ascension Island TCCON sites for the days shown in this study. Here we used the same spectral windows to calculate these values, i.e. windows 1 and 5, and windows 4 and 6. For the $\delta^{13}C$ values we calculate an averaged value for the whole day.



**Table 6.** Daily averaged values of $\delta^{13}$C from both TCCON sites for two $^{12}$CH$_4$ and $^{13}$CH$_4$ window combinations for each spectral database.

| $\delta^{13}$C | TCCON windows 1 & 5 | TCCON windows 4 & 6 | HITRAN windows 1 & 5 | HITRAN windows 4 & 6 | GEISA windows 4 & 6 | SEOM windows 1 & 5 |
|---|---|---|---|---|---|---|
| Tsukuba 01/04/2016 | -116‰ | -1.52‰ | -59.1‰ | -33.1‰ | -193‰ | -109‰ |
| Tsukuba 07/07/2016 | -173‰ | 74.5‰ | -159‰ | 296‰ | -202‰ | -143‰ |
| Ascension Island 23/08/2016 | -108‰ | -92.4‰ | -104‰ | -8.47‰ | -297‰ | -95.0‰ |
| Ascension Island 01/10/2016 | -115‰ | 43.6‰ | -46.7‰ | 160‰ | -134‰ | -84.2‰ |

Table 6 shows that the $\delta^{13}$C values calculated from the windows 1 and 5 combination are largely consistent across sites and dates (with the exception of the July retrievals from Tsukuba), while the values calculated from windows 4 and 6 show much more variation. However, there is still significant variation across all cases that cannot be accounted for purely by precision errors. We therefore assert that in the case of TCCON retrievals, the dominant error in $\delta^{13}$C retrievals are spectroscopic errors

## 3.5 Sensitivity analysis

### 3.5.1 Spectroscopic databases

Here we analyse the differences between the spectroscopic databases for both $^{12}$CH$_4$ and $^{13}$CH$_4$.





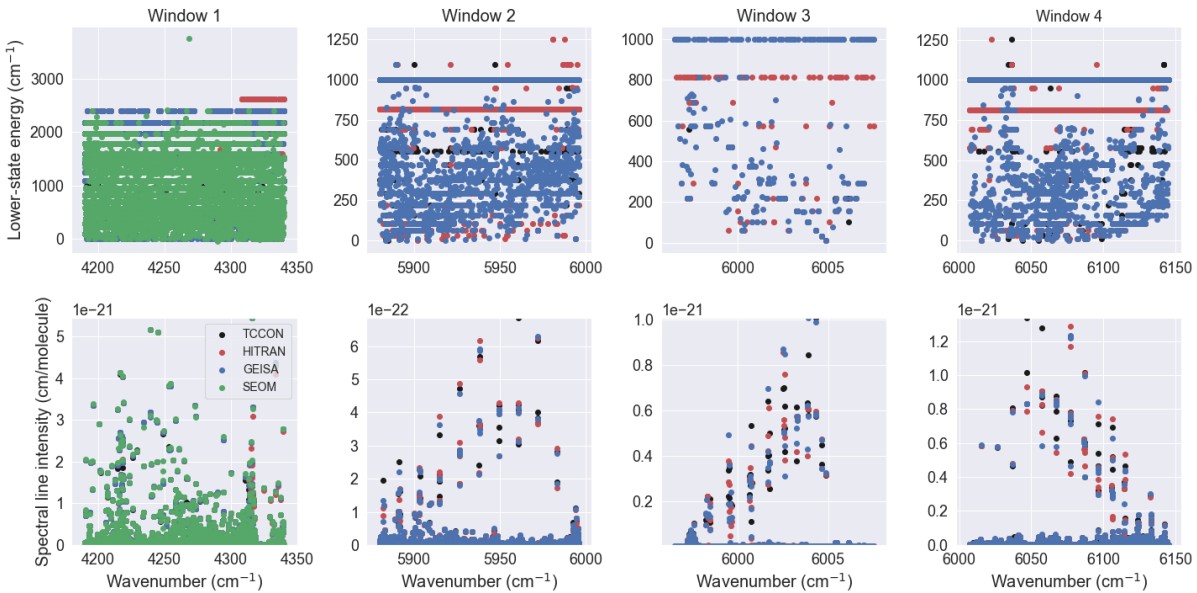

**Figure 7.** Comparisons of spectral line intensity and lower state energy values for $^{12}CH_4$, based on all spectroscopic databases under assessment in this study. The colours represent the spectroscopic databases, in the same format as used in all of the previous figures. The top row shows the lower state energy level and the bottom row show the spectral line intensity, each column identifies the window.

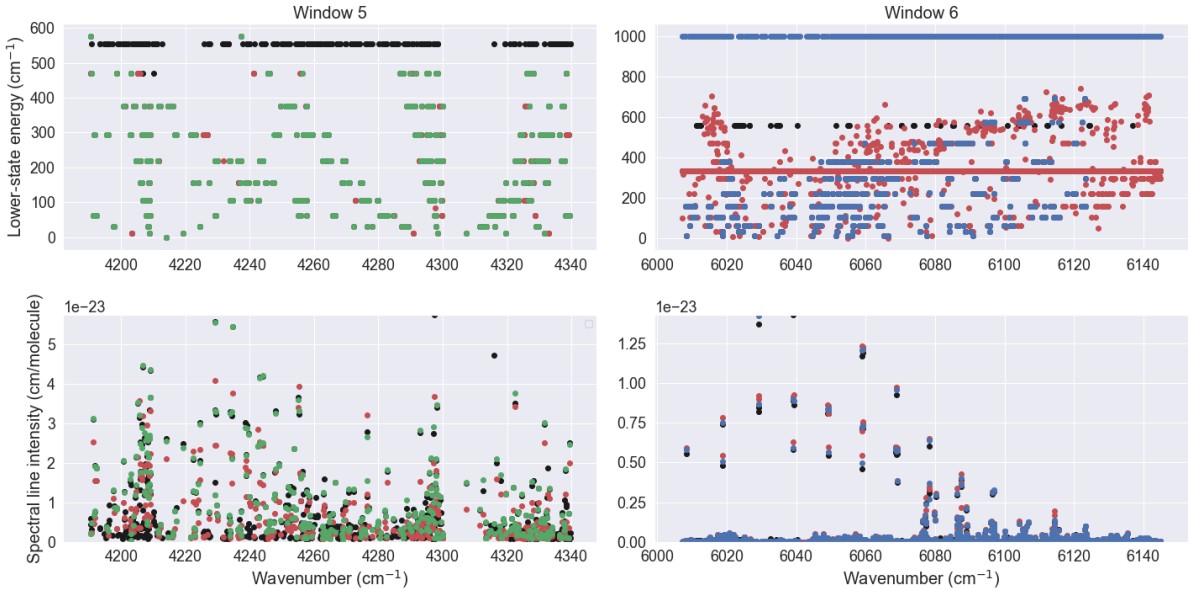

**Figure 8.** As Fig. 7, but with $^{13}CH_4$ under investigation, for windows 5 and 6.





**Table 7.** Statistics for Fig. 7 and Fig. 8 identifying the variations between spectroscopic databases for windows 1-6.

| Window (->) Spectroscopic Database | 1 | 2 | 3 | 4 | 5 | 6 |
|---|---|---|---|---|---|---|
| TCCON | No Lines: 12710 min $E_0$: 0.0 max $E_0$: 2400 mean $E_0$: 1163 $S_{min}$:1e$^{-34}$ $S_{max}$:5.45e$^{-21}$ $S_{sum}$:4.38e$^{-19}$ | No Lines: 445 min $E_0$: 0.0 max $E_0$: 1096 mean $E_0$: 442 $S_{min}$:1e$^{-24}$ $S_{max}$:6.84e$^{-22}$ $S_{sum}$:2e$^{-20}$ | No Lines: 70 min $E_0$: 10.5 max $E_0$: 815 mean $E_0$: 371 $S_{min}$:6.57e$^{-24}$ $S_{max}$:1e$^{-21}$ $S_{sum}$:1.98e$^{-20}$ | No Lines: 411 min $E_0$: 0.0 max $E_0$: 1251 mean $E_0$: 365 $S_{min}$:5.24e$^{-34}$ $S_{max}$:1.34e$^{-21}$ $S_{sum}$:3.84e$^{-20}$ | No Lines: 680 min $E_0$: 0 max $E_0$: 575 mean $E_0$: 335 $S_{min}$:2.32e$^{-41}$ $S_{max}$:5.74e$^{-23}$ $S_{sum}$:4.65e$^{-21}$ | No Lines: 1351 min $E_0$: 10.5 max $E_0$: 690 mean $E_0$: 321 $S_{min}$:3.75e$^{-27}$ $S_{max}$:9.60e$^{-24}$ $S_{sum}$:3.78e$^{-22}$ |
| HITRAN | No Lines: 13356 min $E_0$: 0.0 max $E_0$: 2627 mean $E_0$: 1218 $S_{min}$:1e$^{-29}$ $S_{max}$:5.46e$^{-21}$ $S_{sum}$:4.38e$^{-19}$ | No Lines: 2206 min $E_0$: 0.0 max $E_0$: 1252 mean $E_0$: 597 $S_{min}$:1.38e$^{-26}$ $S_{max}$:7.26e$^{-22}$ $S_{sum}$:2.28e$^{-20}$ | No Lines: 203 min $E_0$: 10.5 max $E_0$: 815 mean $E_0$: 468 $S_{min}$:2e$^{-25}$ $S_{max}$:1.08e$^{-21}$ $S_{sum}$:2.03e$^{-20}$ | No Lines: 1801 min $E_0$: 0.0 max $E_0$: 1252 mean $E_0$: 550 $S_{min}$:2.14e$^{-26}$ $S_{max}$:1.52e$^{-21}$ $S_{sum}$:3.96e$^{-20}$ | No Lines: 446 min $E_0$: 0 max $E_0$: 575 mean $E_0$: 216 $S_{min}$:5e$^{-25}$ $S_{max}$:4.08e$^{-23}$ $S_{sum}$:2.85e$^{-21}$ | No Lines: 2747 min $E_0$: 0 max $E_0$: 742 mean $E_0$: 329 $S_{min}$:1.202e$^{-28}$ $S_{max}$:1e$^{-23}$ $S_{sum}$:3.99e$^{-22}$ |
| GEISA | No Lines: 12678 min $E_0$: 0.0 max $E_0$: 2400 mean $E_0$: 1156 $S_{min}$:9.51e$^{-28}$ $S_{max}$:5.45e$^{-21}$ $S_{sum}$:4.41e$^{-19}$ | No Lines: 3255 min $E_0$: 0.0 max $E_0$: 1096 mean $E_0$: 697 $S_{min}$:1.6e$^{-26}$ $S_{max}$:7.05e$^{-22}$ $S_{sum}$:2.27e$^{-20}$ | No Lines: 293 min $E_0$: 10.5 max $E_0$: 999 mean $E_0$: 623 $S_{min}$:1.94e$^{-26}$ $S_{max}$:1.01e$^{-21}$ $S_{sum}$:2e$^{-20}$ | No Lines: 3367 min $E_0$: 0.0 max $E_0$: 999 mean $E_0$: 735 $S_{min}$:6.94e$^{-27}$ $S_{max}$:1.5e$^{-21}$ $S_{sum}$:3.97e$^{-20}$ | No Lines: na min $E_0$: na max $E_0$: na mean $E_0$: na $S_{min}$:na $S_{max}$:na $S_{sum}$:na | No Lines: 1312 min $E_0$: 10.5 max $E_0$: 999 mean $E_0$: 753 $S_{min}$:3.68e$^{-27}$ $S_{max}$:9.95e$^{-24}$ $S_{sum}$:3.85e$^{-22}$ |
| SEOM | No Lines: 10587 min $E_0$: 0.0 max $E_0$: 3747 mean $E_0$: 1030 $S_{min}$:0 $S_{max}$:5.45e$^{-21}$ $S_{sum}$:4.46e$^{-19}$ | No Lines: na min $E_0$: na max $E_0$: na mean $E_0$: na $S_{min}$:na $S_{max}$:na $S_{sum}$:na | No Lines: na min $E_0$: na max $E_0$: na mean $E_0$: na $S_{min}$:na $S_{max}$:na $S_{sum}$:na | No Lines: na min $E_0$: na max $E_0$: na mean $E_0$: na $S_{min}$:na $S_{max}$:na $S_{sum}$:na | No Lines: 362 min $E_0$: 0 max $E_0$: 575 mean $E_0$: 210 $S_{min}$: 1.28e$^{-25}$ $S_{max}$: 5.89e$^{-23}$ $S_{sum}$: 3.56e$^{-21}$ | No Lines: na min $E_0$: na max $E_0$: na mean $E_0$: na $S_{min}$:na $S_{max}$:na $S_{sum}$:na |

The results indicated in Figs. 7 & 8 as well as Table 7 show that the spectral line intensity values are largely consistent across the spectroscopic databases, despite distinct variations in the number of spectral lines. The key differences seem to occur with the lower state energy level values, especially for the mean values across spectroscopic databases for the same windows. Except for window 1, which shows relatively consistent results for all databases, possibly due to the wider spectral range and large number of spectral lines. We note that window 1 shows the most consistency in the spectroscopic values, and it also shows the

lowest variation in terms of retrieval values. This suggests that variations in the spectroscopic databases are highly significant in terms of the retrievals.





### 3.5.2 A priori error

The impact on adding errors into the a priori temperature knowledge as described in sect. 2.4.2 is explored in the following section. The dependency of other factors such as pressure are shown in Appendix C.

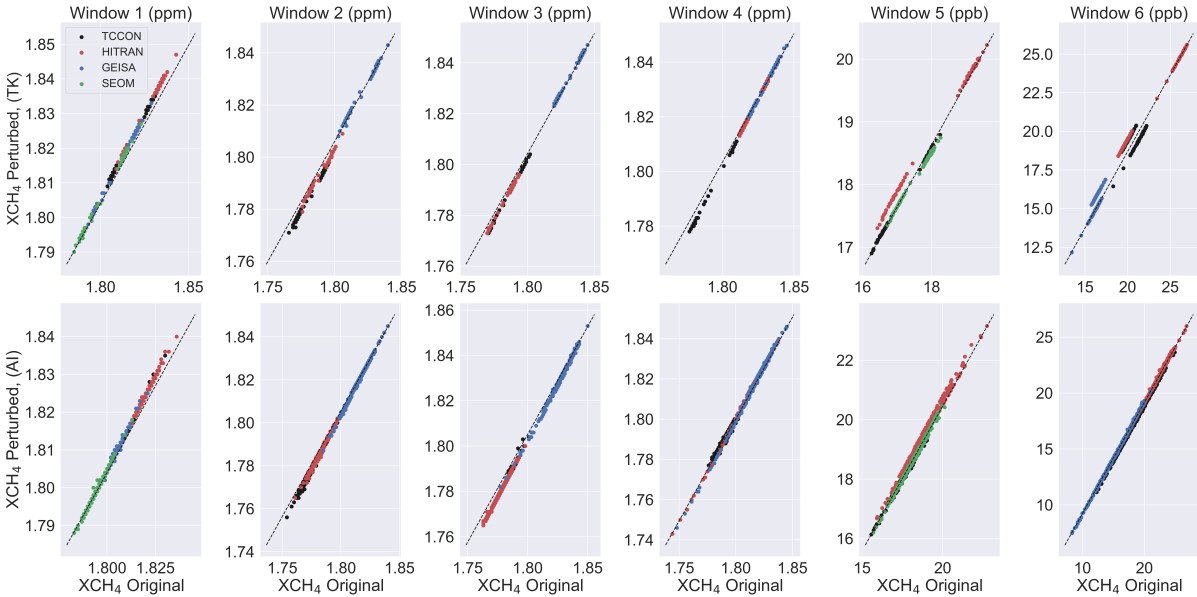

**Figure 9.** Series of scatter plots indicating the differences between retrieved values of $X^{12}CH_4$ and $X^{13}CH_4$ from the standard cases shown in sect. 3.3 and when a 2 K temperature shift is applied to the a priori atmosphere. Each column indicates the window under consideration, the top row is for Tsukuba data (TK), and the bottom row is for Ascension Island data (AI). The colours in the plots are consistent with the rest of this paper in indicating the spectroscopic database.

Adding a 2 K temperature bias into the a priori profile has a notable impact on the retrievals of $X^{12}CH_4$ and $X^{13}CH_4$, with windows 2 and 6 showing large variations qualitatively. These are explored quantitatively in Fig. 10 below.





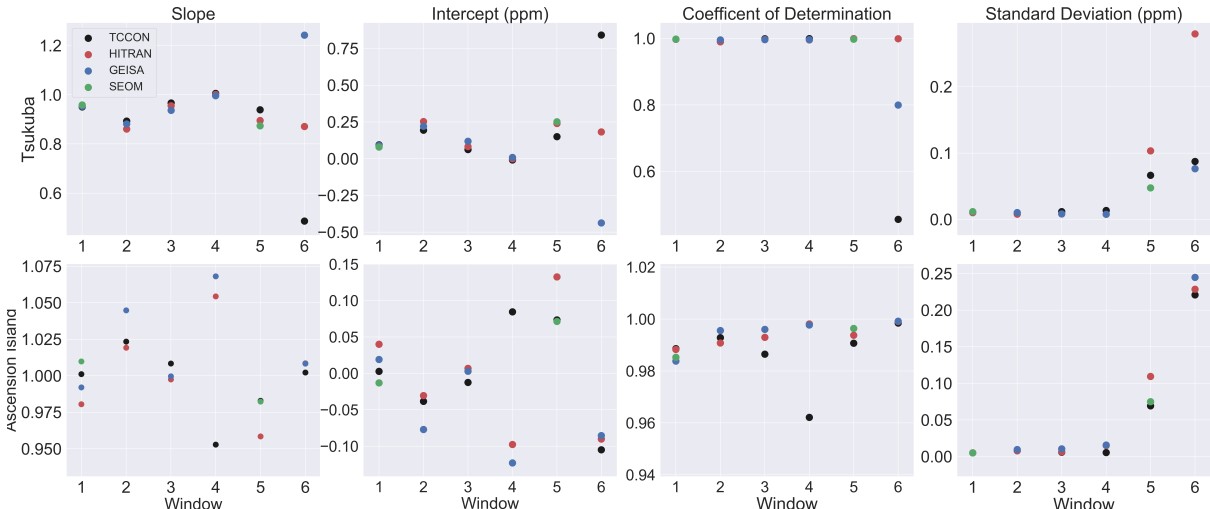

**Figure 10.** Plot indicating the linear statistical relationship between the standard retrievals from sect 3.3 and the perturbed pressure column retrievals. The x-axis for all plots indicates the retrieval window under consideration. The first column shows the values for the linear slope, the second column is the linear intercept, the third column is the coefficient of determination and the fourth column is the standard deviation. The first row indicates retrievals from Tsukuba and the second row shows retrievals from Ascension island. The colours in the plots are consistent with the rest of this paper in indicating the spectroscopic database.

The statistics indicated in Fig. 10 show a significant sensitivity to a priori temperature biases, particularly at the Ascension Island site. In relation to Tsukuba, window 2 shows the most sensitivity for $X^{12}CH_4$, and window 6 shows the most for $X^{13}CH_4$, with each of the spectroscopic databases showing similar values. However for the Ascension island retrievals, windows 2 and 390 4 show similar sensitivities to the temperature dependence, with the spectroscopic databases showing more variability in the results. These results suggest that retrievals of $X^{12}CH_4$ and especially $X^{13}CH_4$ are sensitive to temperature errors.

Further analysis on sensitivities to apriori errors are shown in Appendix C, with the retrievals showing the greatest sensitivity to errors in the pressure profile, and only minor sensitivity to errors in trace gas profiles. Given that both temperature and pressure dependencies exist in the calculation of spectroscopic values, these sensitivities provide a good argument for the 395 retrieval of the temperature and pressure columns.

## 4   Discussion

In sect 3.5.1 we note that there are significant variations in the spectroscopic databases across all of the considered windows. It is also found that typically the spectral line intensity does not deviate significantly between the databases, meaning that the key differences between the databases are caused by the other parameters. It is difficult to assess all of the differences between 400 the databases, due to the range of parameters used, and it is beyond the scope of this paper to provide a comprehensive analysis of all of the differences. There are some papers which describe the sources of the spectral lines for each of the databases (e.g.





Jacquinet-Husson et al. (2016); Brown et al. (2013), but these never go into significant detail. Confusingly, several of these databases state that they drawn data from the same sources (e.g. (Albert et al., 2009; Nikitin et al., 2015, 2017)), meaning that there should be little deviation, but these papers go on to say that not all of the lines from these studies are implemented based
on in house assessments of fit quality. It is clear that the differences in spectroscopic databases dominate the biases noted in this study, and we recommend further work should be done to assess these difference.

In addition to differences between the spectroscopic databases, we have shown that there are significant differences between the spectral windows used for retrieving methane (specifically the isotopologues). This is not a significant problem if the systematic biases between the windows are roughly constant, meaning that by taking an average of these windows, all of the
biases will be removed, which is what is currently done in the Level 2 TCCON retrievals. However, we have also shown that because each window responds differently to errors in starting assumptions such as the a priori profiles, there will never be a constant bias between the windows. Meaning that there is likely to be an underlying bias in all TCCON data that varies from retrieval to retrieval, and day to day.

One of the key driving factors behind the difficulties of retrieving constant $\delta^{13}C$ values in this study are the sensitivity of the
retrievals (especially $X^{13}CH_4$) to a priori input errors. The retrievals seem especially sensitive to pressure and water vapour errors (in high humidity conditions), meaning that the uncertainty associated with these retrievals will vary depending on the season. This is a significant hurdle, but better results could be obtained if the error on this a priori data is reduced significantly. We note that GFIT is a scaling retrieval algorithm, so improvements could be made if we had used an algorithm which uses profile fit methods, and allows for the inclusion of pressure and temperature terms into the state vector. The profile fit method
is potentially very important, especially in relation to water vapour, where several of the fit windows show themselves highly sensitive to water vapour a priori errors. We note that such advancements are currently being tested (Zhou et al., 2019) with the "SFIT4" algorithm, and the application of this algorithm to this study could well yield improved results. We did not assess systematic errors associated with errors in the profile shapes of $^{12}CH_4$ and $^{13}CH_4$, even though these are likely to be high. We judged that biases due to variations in the spectroscopic databases dominated all other uncertainties, and there would not be
much added benefit to including data on the profile shapes. Fundamentally the sensitivity analysis in this study assumed the worst case scenarios for a priori uncertainty (e.g. a total column temperature shift of 2 K). The uncertainties on the a priori data may not be as high as suggested by this study, and indeed may be 0, however it is very difficult to assess this and we therefore assume the worst.

This study includes a question that is of some interest to the community, "can we calculate realistic and constant $\delta^{13}C$ values
from TCCON". The results from this study suggest not yet, due to varity of factors influencing the retrievals. The $\delta^{13}C$ values calculated using windows 4 & 6 show large variability across both the spectroscopic databases and TCCON sites. The fact that such a wide spread in $\delta^{13}C$ values are calculated using the same measurements on the same day is of concern, especially when the measurements were averaged over at least 79 points, implying the random error should not induce such wide fluctuations in the $\delta^{13}C$. The results from windows 1 and 5 show much more realistic results, which are especially consistent between the
TCCON and SEOM-IAS spectroscopic databases, and between months. However, there is still a significant bias in these results, given that the tropospheric average $\delta^{13}C$ value is assumed to be -47‰ (Sherwood et al., 2016). We expect that large deviations



from this value is unlikely, given that TCCON retrieves total column estimates, and not in-situ samples. This assumption of -47‰ is a little unfair, since this is an assumption based on lower tropospheric averages, and does not take into account sink processes that occur further up into the atmosphere. For example Rigby et al. (2017) assume a -2.6‰ fractionation due to the chlorine sink in the stratosphere. However, it can be argued here that the priority in calculating an accurate value of $\delta^{13}$C from TCCON is a full assessment of all of the systematic biases present in the retrievals, most notably the spectroscopic biases, before discussion of the true $\delta^{13}$C value of the total column.

## 5 Conclusions

In this study we retrieve $X^{12}CH_4$ and $X^{13}CH_4$ DMFs from two TCCON sites, with the aim of understanding the biases associated with retrieving methane isotopologues in the TROPOMI spectral region as opposed to standard TCCON methane windows. We use four sources of spectroscopic parameters, the HITRAN2016, GEISA2015, SEOM-IAS and internal TCCON databases in order to assess the accuracy of $X^{13}CH_4$ spectral lines. Measurements are taken from two TCCON sites (Tsukuba, and Ascension island) to provide a range in atmospheric conditions.

We retrieve $X^{12}CH_4$ from four different spectral windows, and $X^{13}CH_4$ from two spectral windows covering the spectral range of the future S5/UVNS instrument and the current S5P/TROPOMI instrument. Three of the $X^{12}CH_4$ windows are routinely used in TCCON products, but the window in the 4265 cm$^{-1}$ range is not, and we therefore compare the variation of the retrievals from these standard and non-standard windows. There are currently no published studies on $X^{13}CH_4$ retrievals from TCCON, and we therefore assessed the accuracy of retrievals from two differing windows. The best retrieval fit metrics are found for the SEOM-IAS and internal TCCON spectroscopy databases (with the SEOM-IAS database limited to the 4265 cm$^{-1}$ spectral range). Where uniquely, the SEOM-IAS database includes non-Voigt line shapes for methane.

We find significant levels of bias between the retrieved $X^{12}CH_4$ DMFs both in terms of spectral window and spectroscopic database. In some cases, similar windows from different spectroscopic databases differ by as much as 50 ppb. These biases remain consistent between seasons and TCCON sites, implying systematic errors in the spectroscopic parameters. Window 1 (4265 cm$^{-1}$) shows the lowest variation between the databases, typically < 10 ppb. While the short window 3 (6002 cm$^{-1}$) shows the most variation, typically > 20 ppb. Window 5 (4265 cm$^{-1}$) retrievals for $X^{13}CH_4$ are highly consistent, typically < 0.5 ppb bias. While window 6 (6076 cm$^{-1}$) retrievals of $X^{13}CH_4$ are less consistent, with up to 4 ppb standard deviation ( 20%) between the databases. In addition to these biases, we find that retrievals of $X^{12}CH_4$ and $X^{13}CH_4$ are sensitive to uncertainty in the a priori data, especially pressure and temperature errors, and to instrument errors. Furthermore, we find that each of the spectroscopic databases and spectral windows assessed in this study show differing sensitivities to a priori uncertainties.

Based on the results presented in this study, we recommend including the TROPOMI SWIR spectral region into future TCCON methane retrievals, due to the consistency of the retrievals presented in this study. In addition, based on major deviations between retrievals based on different spectroscopic databases, we call for further investigation into these deviations.

The $\delta^{13}$C values at each TCCON site were calculated in order to determine if $\delta^{13}$C values could be assigned to TCCON sites. Uncertainty is high on $X^{13}CH_4$ retrievals (0.1-2 ppb) depending on site and database), which is not low enough to calculate





$\delta^{13}$C with the uncertainty required to determine annual variation (<1‰ uncertainty). However, given the stationary nature of
TCCON sites, the precision on the retrievals can be increased through long term averaging. This would require the averaging
of hundreds (perhaps thousands) of measurements, as well as highly accurate a priori data, knowledge of airmass fluctuations,
full characterisation of any instrumentation errors and very careful selection of spectroscopic database and spectral window.
Such accuracy from a priori data sources is not readily available, therefore $\delta^{13}$C value from TCCON is not a likely prospect
for the foreseeable future, at least until reliable spectroscopic data can be sourced for $^{13}$CH$_4$.

*Code and data availability.*   The GGG2014 retrieval environment is available at https://tccon-wiki.caltech.edu, and TCCON L1b spectra are
available upon discussion with the relevent site PI

## Appendix A:  Transmission

Example transmission for Ascension Island.

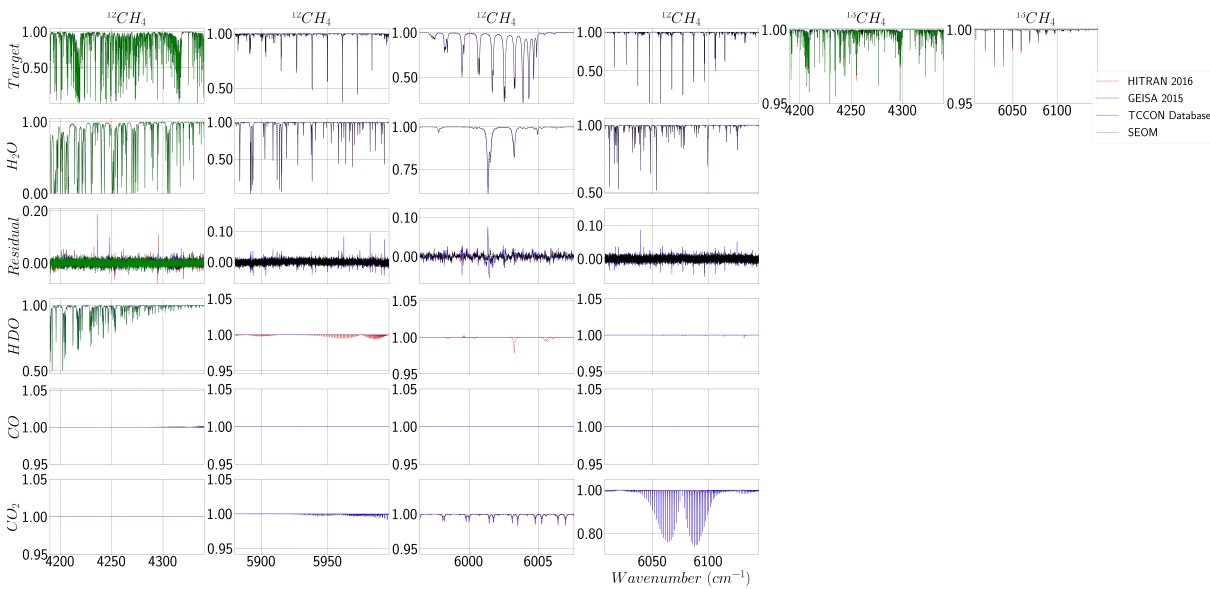

**Figure A1.** As Figure 2, but for Ascension Island in October 2016.





**Table A1.** Statistics for Fig A1, based on metrics identified in sect 2.2.

| | Window 1 | Window 2 | Window 3 | Window 4 |
|---|---|---|---|---|
| RMSE | TCCON: $8.748 \times 10^{-3}$<br>HITRAN: $1.061 \times 10^{-2}$<br>GEISA: $9.880 \times 10^{-3}$<br>SEOM: $8.508 \times 10^{-3}$ | TCCON: $6.488 \times 10^{-3}$<br>HITRAN: $7.035 \times 10^{-3}$<br>GEISA: $7.296 \times 10^{-3}$<br>SEOM: nan | TCCON: $6.482 \times 10^{-3}$<br>HITRAN: $8.738 \times 10^{-3}$<br>GEISA: $8.887 \times 10^{-3}$<br>SEOM: nan | TCCON: $6.253 \times 10^{-3}$<br>HITRAN: $6.723 \times 10^{-3}$<br>GEISA: $6.906 \times 10^{-3}$<br>SEOM: nan |
| $\chi^2$ | TCCON: 1.524<br>HITRAN: 2.241<br>GEISA: 1.944<br>SEOM: 1.441 | TCCON: 0.648<br>HITRAN: 0.762<br>GEISA: 0.820<br>SEOM: nan | TCCON: 0.0670<br>HITRAN: 0.112<br>GEISA: 0.116<br>SEOM: nan | TCCON: 0.716<br>HITRAN: 0.828<br>GEISA: 0.874<br>SEOM: nan |

**Appendix B: Retrievals**

Additional retrieval time series.

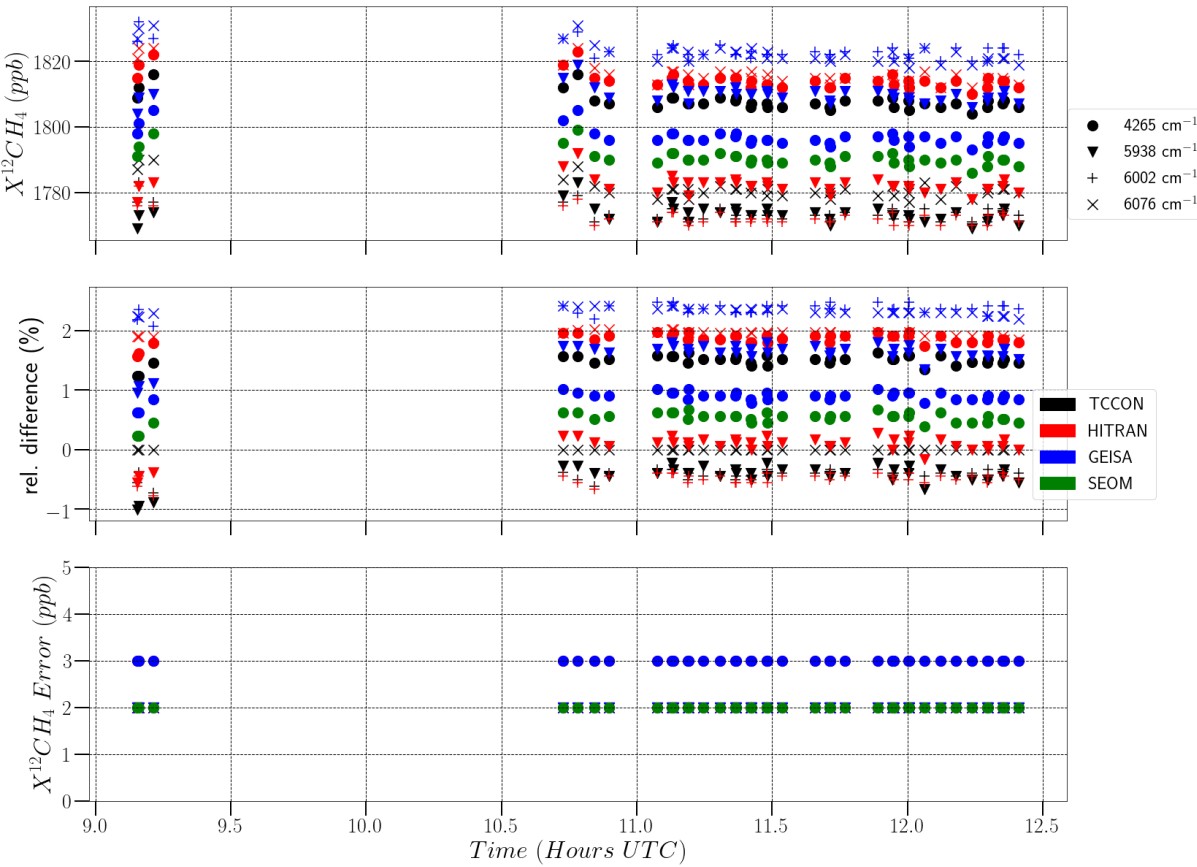

**Figure B1.** As Figure 3 but for 40 retrievals on 07/07/2016.





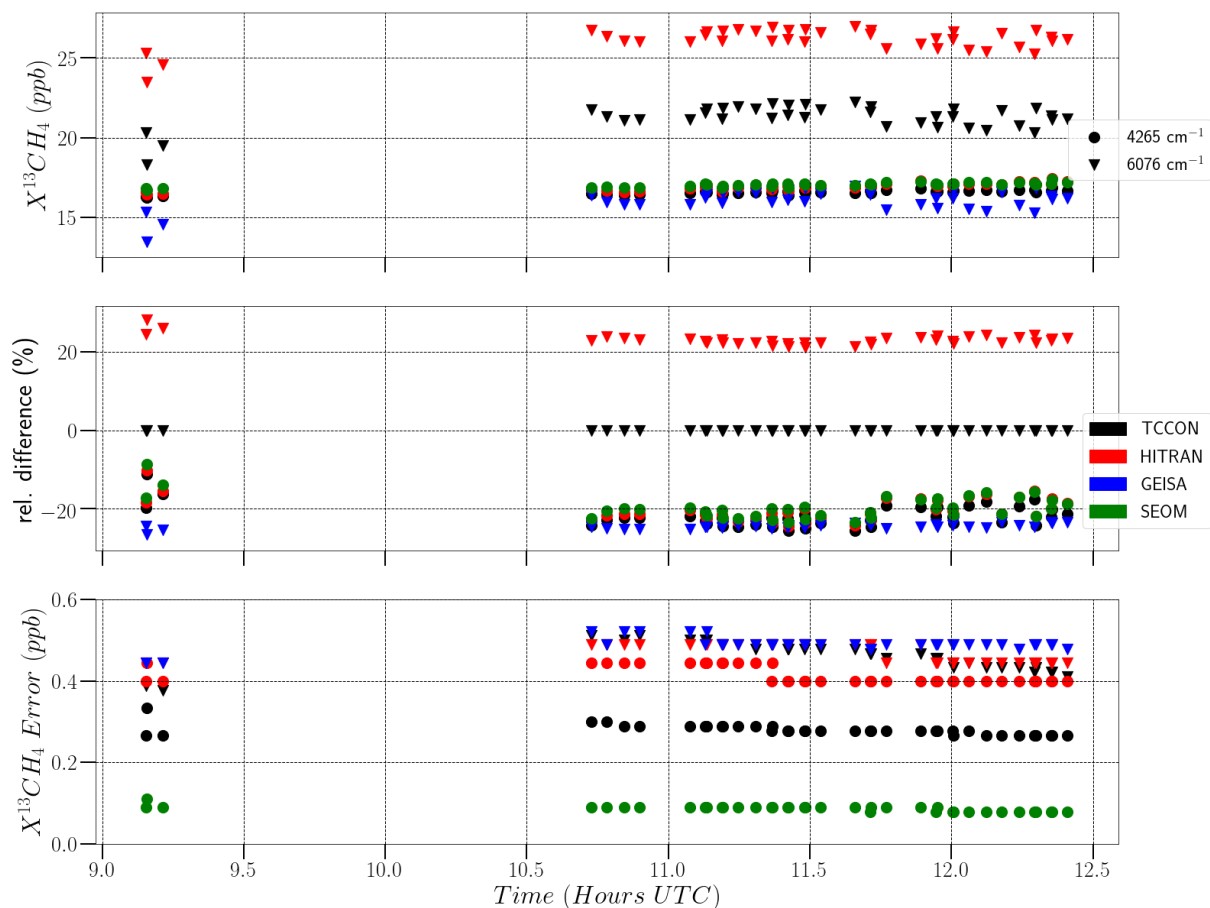

**Figure B2.** As Figure 4, but for 40 retrievals on 07/07/2016.





**Table B1.** Statistics for Fig B1 and Fig 2 based on metrics identified in retrieval abundances subsection of section 2.2.

| Window | 1 | 2 | 3 | 4 | 5 | 6 |
|---|---|---|---|---|---|---|
| $\sigma_{window}$ (ppb) | 9.42 | 15.7 | 23.7 | 18.5 | 0.26 | 4.14 |
| Database | | | TCCON | HITRAN | GEISA | SEOM-IAS |
| $\sigma_{inter-window}X^{12}CH_4$ (ppb) | | | 16.4 | 18.1 | 10.5 | N/A |
| $\sigma_{inter-window}X^{13}CH_4$ (ppb) | | | 2.40 | 4.63 | N/A | N/A |

| Database | HITRAN | GEISA | SEOM-IAS |
|---|---|---|---|
| b (ppb; window 1) | 6.54 | 9.74 | 17.2 |
| b (ppb; window 2) | 8.87 | 36.4 | N/A |
| b (ppb; window 3) | 1.23 | 49.4 | N/A |
| b (ppb; window 4) | 34.6 | 41.9 | N/A |
| b (ppb; window 5) | 0.338 | N/A | 0.307 |
| b (ppb; window 6) | 4.88 | 5.10 | N/A |



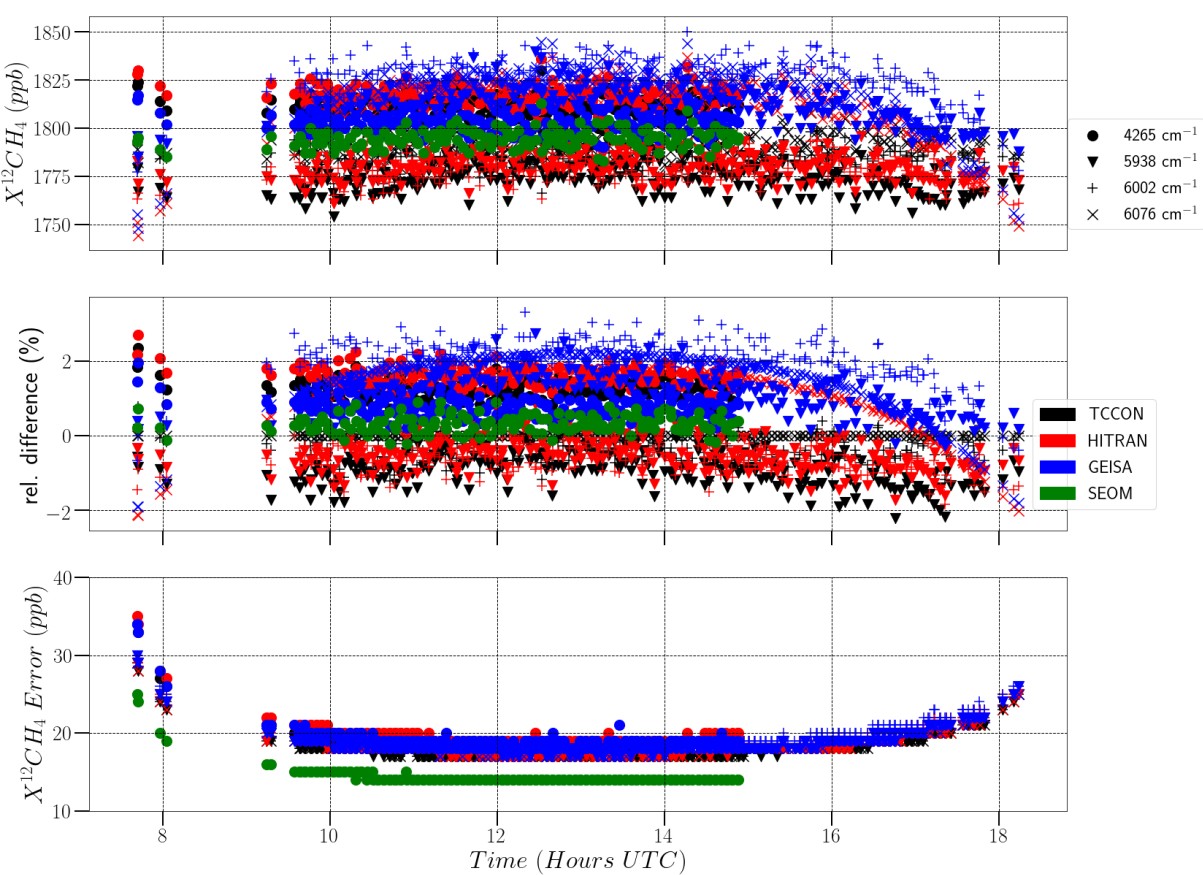

**Figure B3.** As Figure 5 but for 243 retrievals on 23/08/2016.



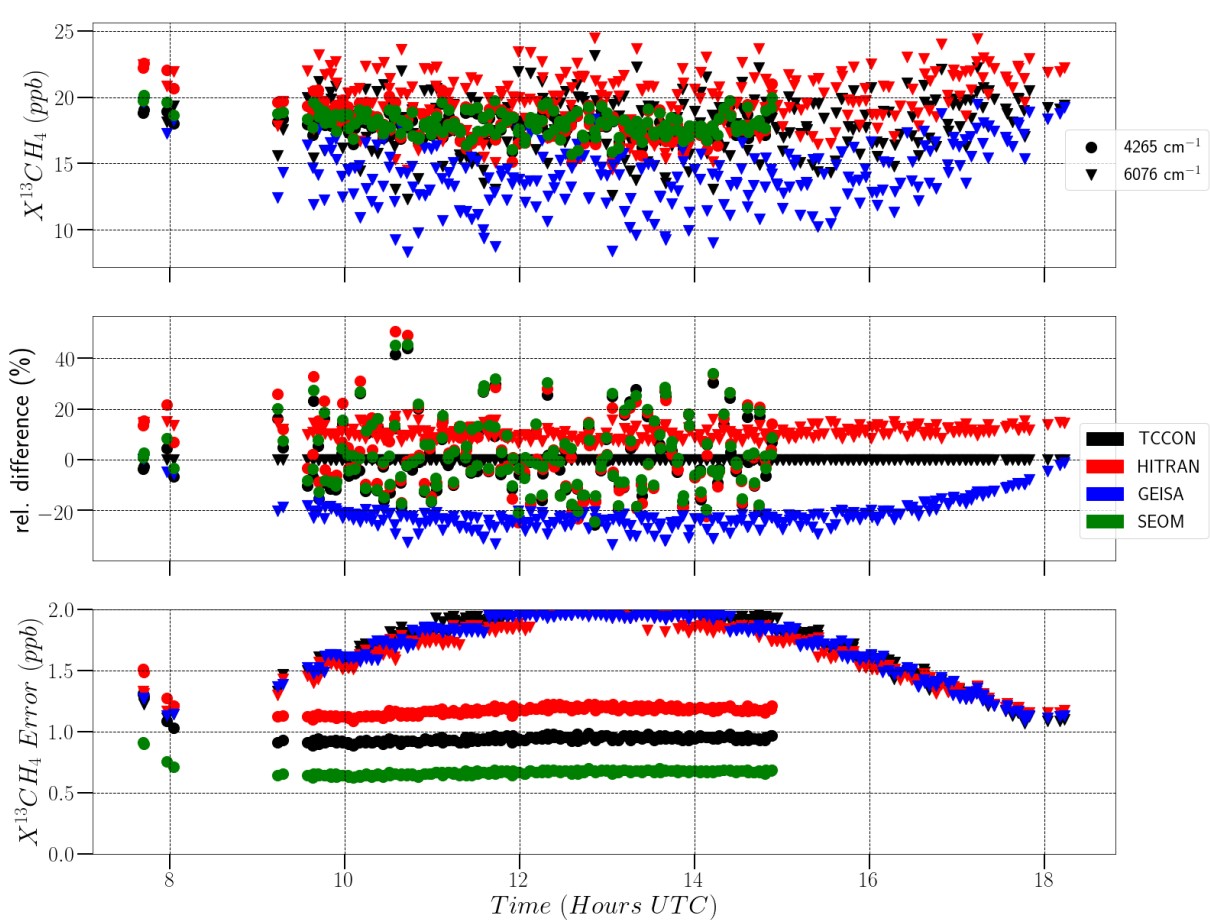

**Figure B4.** As Figure 4, but for 243 retrievals on 23/08/2016.





**Table B2.** Statistics for Fig B3 and Fig B4 based on metrics identified in retrieval abundances subsection of section 2.2.

| Window | 1 | 2 | 3 | 4 | 5 | 6 |
|---|---|---|---|---|---|---|
| $\sigma_{window}$ (ppb) | 10.0 | 18.3 | 23.2 | 17.9 | 1.02 | 3.22 |
| Database | | | TCCON | HITRAN | GEISA | SEOM-IAS |
| $\sigma_{inter-window}X^{12}CH_4$ (ppb) | | | 16.4 | 18.1 | 13.4 | N/A |
| $\sigma_{inter-window}X^{13}CH_4$ (ppb) | | | 1.69 | 1.99 | N/A | N/A |

| Database | HITRAN | GEISA | SEOM-IAS |
|---|---|---|---|
| b (ppb; window 1) | 6.30 | 8.99 | 17.0 |
| b (ppb; window 2) | 7.16 | 37.5 | N/A |
| b (ppb; window 3) | 5.71 | 42.8 | N/A |
| b (ppb; window 4) | 21.0 | 27.4 | N/A |
| b (ppb; window 5) | 0.368 | N/A | 0.307 |
| b (ppb; window 6) | 1.90 | 3.87 | N/A |



## Appendix C:  A priori errors

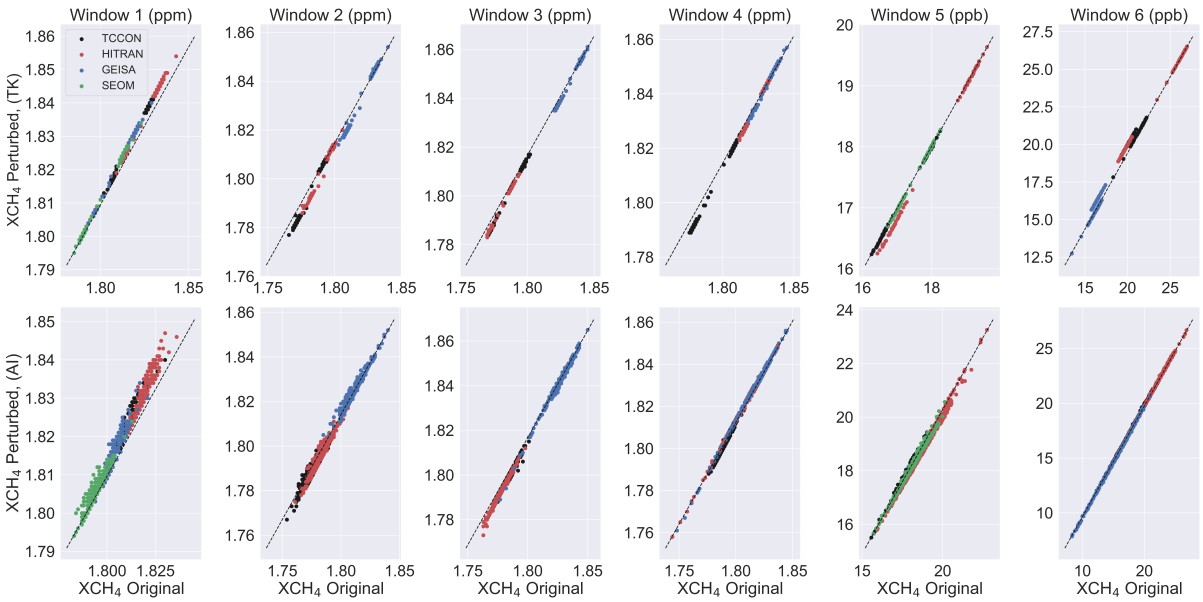

**Figure C1.** Series of scatter plots indicating the differences between retrieved values of $X^{12}CH_4$ and $X^{13}CH_4$ from the standard cases shown in sect. 3.3 and when a 2 % pressure shift is applied to the a priori atmosphere. Each column indicates the window under consideration, the top row is for Tsukuba data (TK), and the bottom row is for Ascension Island data (AI). The colours in the plots are consistent with the rest of this paper in indicating the spectroscopic database.

The results shown in Fig. C1 indicate that there is significant dependency on the accuracy of the a priori pressure column, which varies between the TCCON sites. Figure C2 below quantitatively explores the variations though linear regression statistics for 485 each of the cases indicated in Fig. C1.



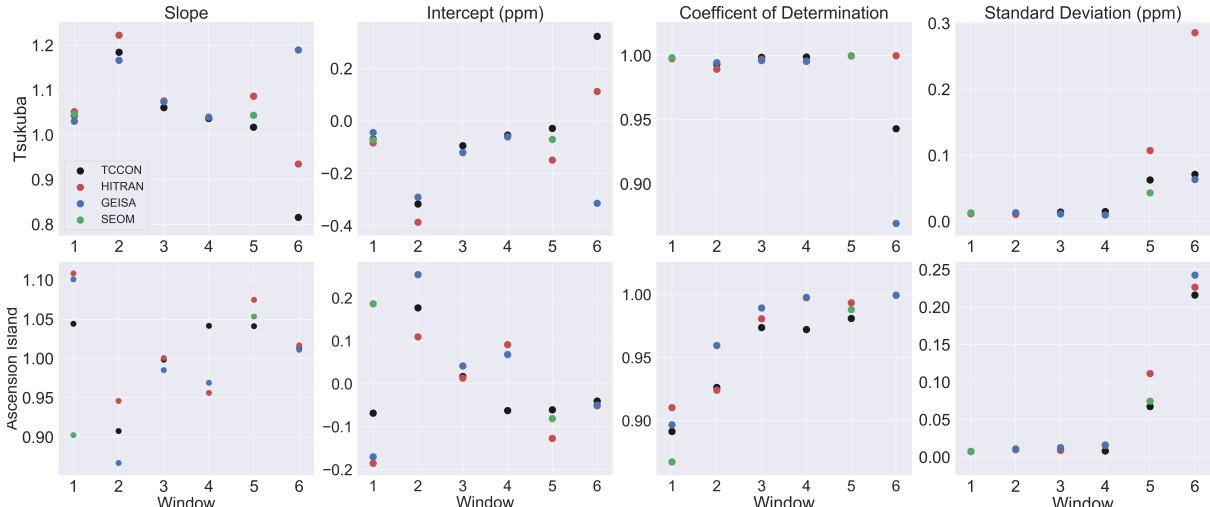

**Figure C2.** Plot indicating the linear statistical relationship between the standard retrievals from sect 3.3 and the perturbed pressure column retrievals. The x-axis for all plots indicates the retrieval window under consideration. The first column shows the values for the linear slope, the second column is the linear intercept, the third column is the coefficient of determination and the fourth column is the standard deviation. The first row indicates retrievals from Tsukuba and the second row shows retrievals from Ascension island. The colours in the plots are consistent with the rest of this paper in indicating the spectroscopic database.

Figure C2 shows that the sensitivity of $X^{12}CH_4$ and $X^{13}CH_4$ to a systematic bias in the pressure column varies depending on the spectroscopic database, and the window. While the fact that different spectroscopic databases and different windows react differently to a priori errors is not surprising, the significance of the differences in all cases is. For the $X^{12}CH_4$ cases (windows 1-4), window 2 typically shows the most sensitivity while for the $X^{13}CH_4$ cases (windows 5-6), window 6 shows

the most sensitivity. Interestingly, the results from Ascension island suggest greater insensitivity to the pressure error, but this could be attributed to a greater number of measurements available for this analysis.





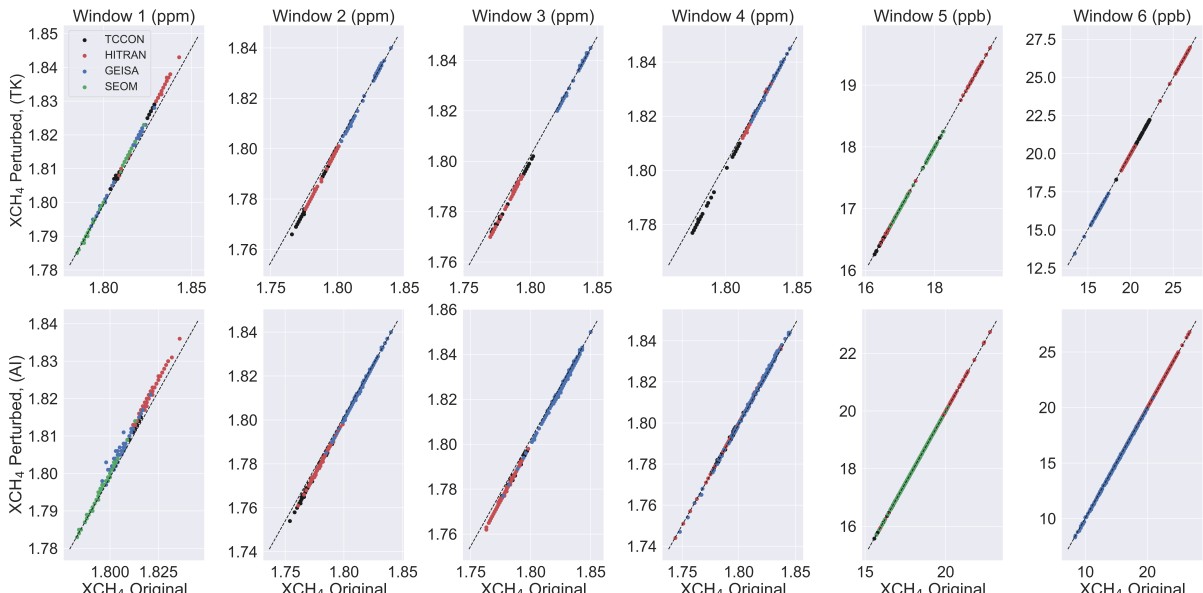

**Figure C3.** As Fig. C1, but for comparisons with a 2% methane bias.

Figure C3 suggests that there is some sensitivity to a priori methane errors, but that it is not as significant as those indicated for pressure and temperature errors.

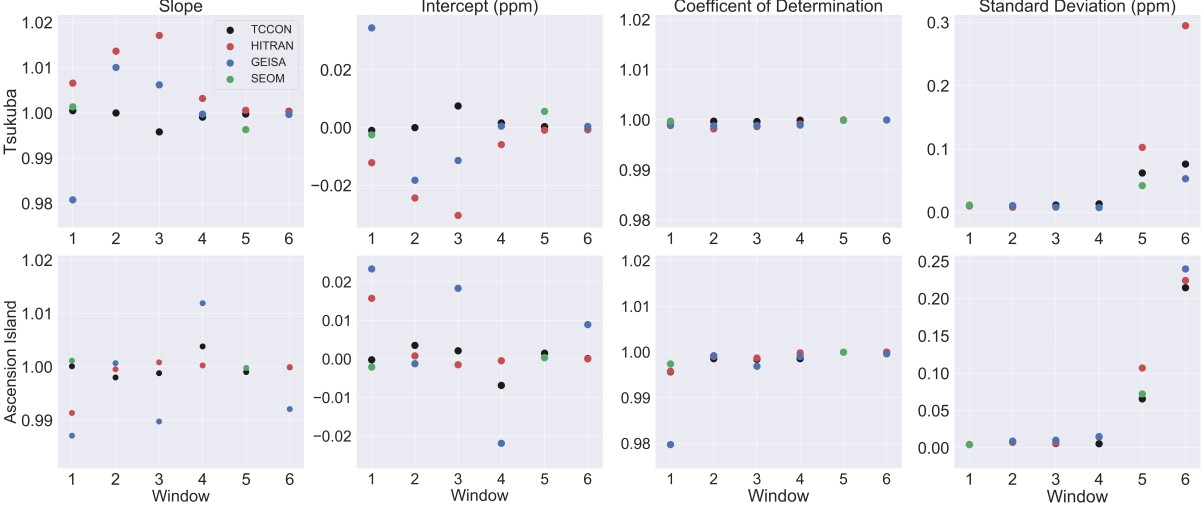

**Figure C4.** As Fig. C2, but for comparisons with a 2% methane bias.

Figure C4 shows significantly less dependency on errors in the methane column than any of the other cases investigated so far. It is interesting to note that the results from the TCCON and SEOM-IAS spectroscopic databases show more consistency





than those from the HITRAN and GEISA databases. There are no clear cases where specific windows show more sensitivity than any of the others, and in this case the $X^{13}CH_4$ retrievals show no more sensitivity than the $X^{12}CH_4$ retrievals.

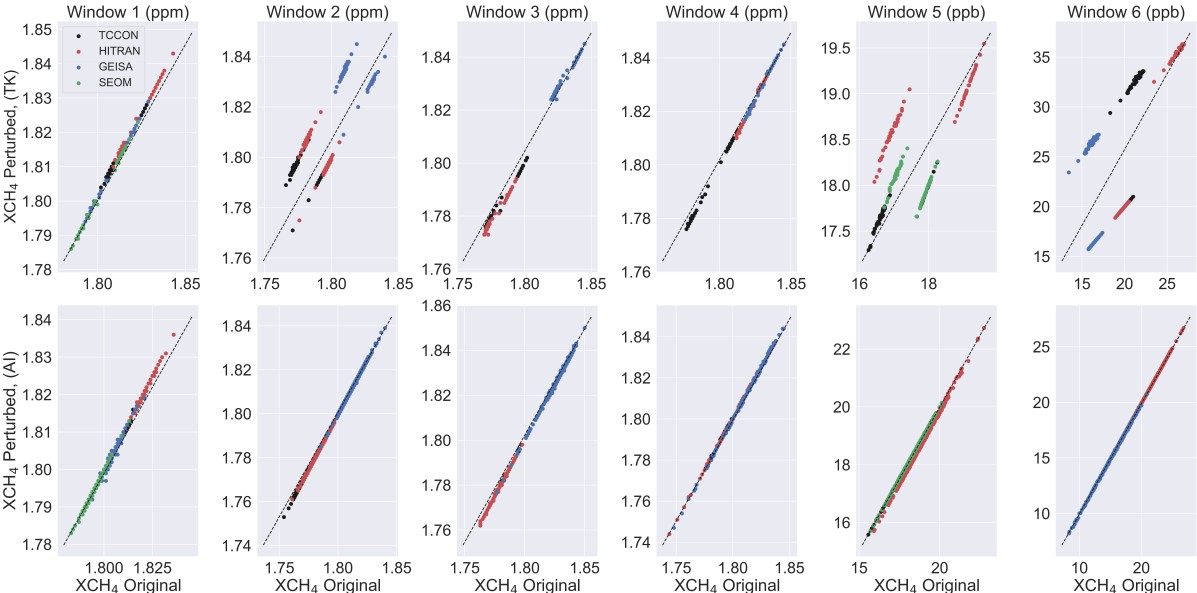

**Figure C5.** As Fig. C1, but for comparisons with a 10% water vapour bias.

Figure C5 shows that Tsukuba is apparently far more sensitive to errors in the a priori water vapour column in some spectral windows than Ascension island. We note that the conditions for Tsukuba site in July of 2016 (indicated in Table D1) show very high levels of water vapour, which is a possible cause for this apparent sensitivity. The linear statistics shown in Fig. C6 explore these differences in more detail.






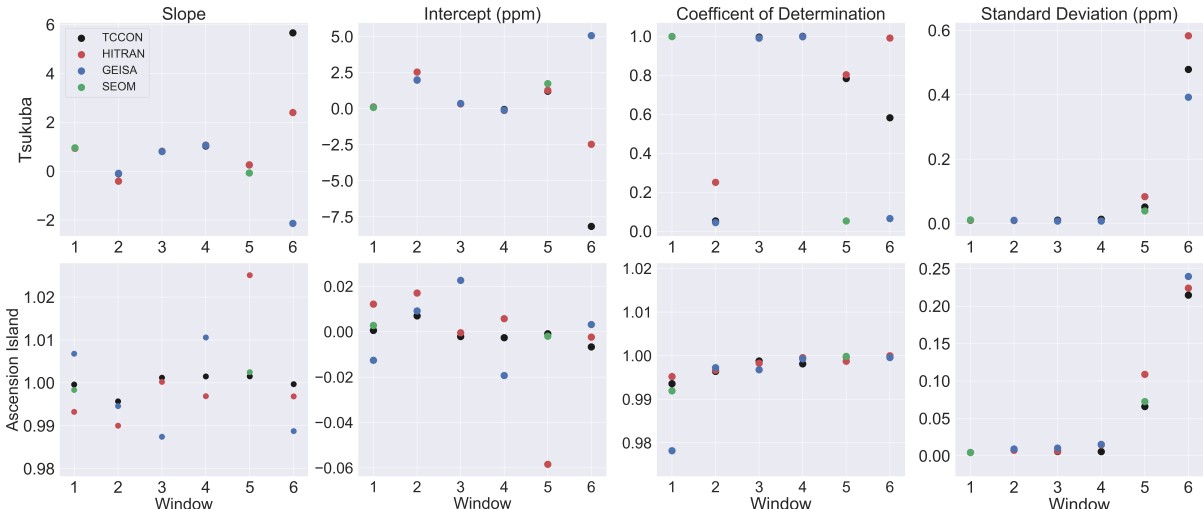

**Figure C6.** As Fig. C2, but for comparisons with a 10% water vapour bias.

The statistics for the Ascension island retrievals show similar values to those for the methane error values, shown in Fig. C4. Suggesting that the retrievals of $X^{12}CH_4$ and $X^{13}CH_4$ are sensitive to errors in the a priori water vapour column, but not to the same degree as with pressure or temperature. However, as we identified earlier in this paper Ascension island retains

a consistent year round humidity due to its location. Conversely Tsukuba has a wide range of seasons and therefore highly variable humidity, which is apparent in the statistics shown in Fig. C6. However, although some of the values are very large, again there is the pattern that windows 2 and 6 show far more sensitivity than any other of the windows. For example, in this case the slope and intercept for window 1 of the TCCON spectroscopic database are 0.939 and 0.112 ppm respectively, which are still large when compared to the Ascension island results, but are on the same scale as the results from the pressure and

temperature errors.

**Appendix D: Additional data**

**Table D1.** Daily ranges of a priori and measured surface temperatures, and averaged $XH_2O$ DMFs from both TCCON sites.

|  | A priori surface temperature (°C) | Site measured temperature (°C) | $XH_2O$ (average) |
|---|---|---|---|
| Tsukuba (01/April/2016) | 12.0 | 15.8-16.8 | 2069 ppm |
| Tsukuba (01/July/2016) | 25.1 | 29.2-33.2 | 6896 ppm |
| Ascension Island (23/Aug/2016) | 22.4 | 24-27.3 | 3752 ppm |
| Ascension Island (01/Oct/2016) | 22 | 24.3-25.9 | 4345 ppm |

Water vapour retrievals are taken from the 4565 cm$^{-1}$ spectral window.



*Author contributions.*  DGF provided the Ascension Island TCCON data, IM provided the Ascension Island TCCON data. EM devised and performed the study, analysed the data and wrote the paper. BV consulted on the interpretation of the results. All authors reviewed the paper.

*Competing interests.*  We declare no competing interests.

*Acknowledgements.*  This study has been performed in the framework of the postdoctoral Research Fellowship program of the European Space Agency (ESA). The GGG2014 retrieval environment was developed at the California institute of Technology (Caltech), and is available at https://tccon-wiki.caltech.edu. Thanks to Geoff Toon at Caltech for providing advice on the use of GGG2014. HITRAN2016 is available at https://hitran.org/, GEISA2015 is available from http://ara.abct.lmd.polytechnique.fr/index.php?page=geisa-2. The SEOM-IAS database is 520  available at https://www.wdc.dlr.de/seom-ias/. The TCCON line list is described in the GGG documentation. The Ascension Island TCCON station has been supported by ESA under grant 3-14737 and by the German Bundesministerium für Wirtschaft und Energie (BMWi) under grants 50EE1711C and 50EE1711E. The TCCON sites at Tsukuba is supported in part by the GOSAT series project. Thanks to Anu Dudhia at Oxford University for the Fortran routine to convert the GEISA line structure into a HITRAN line structure.





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
