# Peer review of "On the consistency of methane isotopologue retrievals using TCCON and multiple spectroscopic databases."

_Atmospheric Measurement Techniques, 2020_

## Referee Comment (RC1) · Anonymous Referee #1 · 18 Jun 2020

**Reviewer comments**

**Malina et al., On the consistency of methane isotopologue retrievals using TCCON and multiple spectroscopic databases**

**General comments**

This paper assesses comparative retrievals of both $^{12}CH_4$ and $^{13}CH_4$ total column amounts from TCCON spectra, with an emphasis on a new spectral window and line parameter data available for the TROPOMI instrument on ESA satellites (S5P, S5), but not currently used by TCCON. The study covers 4 separate line parameter lists and 4 spectral windows, with spectra of differing quality from 2 TCCON sites, Tsukuba and Ascension Island.

The study is thorough and detailed, covering the sensitivity of 4 x 4 x 2 = 32 different combinations of analysis to a number of potential sources of error. In general I have no argument with the thrust and rigour of the work, apart from specific comments below. However I find the level of detail difficult to assimilate, so that the key conclusions get lost in the (repetitive) detail and length of the paper. I would make the general recommendation to move much of the detail expressed in similar arrays of plots and figures to supplementary material or an appendix, while providing an essential summary of each comparison and sensitivity test in the main paper. I do not provide a detailed roadmap for this, but I have made some suggestions in the specific comments. This would shorten the main paper and improve its readability for a wide audience, while not removing the detail for those readers who need it. The conclusions do provide a reasonable summary of the main points from the study, but the road is long to get there.

I have made comments below about the feasibility of making *useful* remote sensing measurements of $\delta^{13}C$ in $CH_4$. I agree with the authors' final conclusion that this is not currently feasible with sufficient accuracy to be useful, but this part of the analysis is over emphasised through the paper because it is clear from the start that this is the case. The sections on $^{13}CH_4$ could be much reduced while still coming to the same valid conclusion.

The paper is suitable for publication in AMT, but I recommend a major revision to restructure it to bring out the key messages from the large amount of detail.

**Specific comments**

P1 L15: Results obtained with the SEOM-IAS database **in the 4190-4349 cm$^{-1}$ window** show the lowest fit residuals

P2L40: Can you provide a reason why TCCON uses 6000 cm-1 and does not use this window? Is it because of the greater interference from water vapour? See also P3 L60.

P2L52: My understanding is that solar occultation refers to measurements from space (or at least above most of the atmosphere) where the ray path follows the sun as it sets or rises through the earth's limb between high elevation angle (exo-atmopshere) to low. TCCON would be better described as taking direct-beam solar absorption measurements from the ground.

P3L82: $^{12}CH_4$ is closer to 99% than 98%. This paragraph overstates the possibility of retrieving useful information on $\delta^{13}C$. As pointed out later (p8, discussion and conclusion), $\delta^{13}C$ in $CH_4$ varies only by around 1‰ from -47‰ *in situ* near the ground except close to strong sources, and we would expect total column amounts to vary much less than that. The possibility of measuring total column $\delta^{13}C$ in $CH_4$ with sufficient accuracy and precision to be useful for source attribution is not just challenging, it

is very remote indeed. Although the authors do mention the challenge, this paragraph holds out a false hope that could have been recognised from the beginning.

P4L99   Remove "While" – not  a sentence. (There are other instances…)

P4L123: This is ambiguously written – GFIT assumes a fixed profile shape for EACH trace gas – but they are different for each trace gas

P4L130: most spectral *features* are resolved, but many spectral  *lines* will overlap at the pressures and temperatures in the atmosphere.

P7 L188: Should $\sigma$(intra-window) rather read $\sigma$(inter-window)?

P7L192: A posteriori error appears twice.

P7L196:  the denominator should read $(^{13}CH_4/^{12}CH_4)_{VPDB}$ not just VPDB

P8L210: Using CO accuracy and precision requirements as a proxy for $^{13}CH_4$ because it has a similar total column amount is misleading and irrelevant. $^{13}CH_4$ measured with the target precision and accuracy for CO is geophysically/geochemically useless. We already know the value better than that -it  is very closely defined by the $^{12}CH_4$ column because $\delta^{13}C$ varies by less than 1‰ and $^{13}CH_4$ can be calculated from Eq. 2.  What really matters is how much $^{13}CH_4$ varies from this value. That requires far higher precision and accuracy. The text and objectives for $^{13}CH_4$ measurement should reflect this reality in this and the paragraph beginning on L219.

P8L218: See comments above.  this quoted uncertainty of <1‰ is required in in situ measurements near the surface – once the small variations near sources of varying $\delta^{13}C$ are mixed into the (almost unvarying) total column by the retrieval, the requirement is <<<1‰.

P9L232: There is ambiguity between the line parameters, which are intrinsic properties of the molecule, and the spectral lines as observed and analysed, which depend on both the intrinsic line parameters and the cxternal onditions – concentrations, pressure, temperature. The databases differ most likely in line positions, strengths, lower state energies and widths, as well as completeness – some lines may be missing in one list compared to another. These parameters are not, or are only weakly, pressure and temperature dependent themselves – they determine the positions, strengths and widths of absorption lines and their temperature- and pressure-dependences. – eg. the lower state energy is the main determinant of temperature dependence of a transition and hence absorption strength – it is not temperature dependent itself.

The max/min/mean of line intensities and lower state energies seems like a very crude measure for meaningful comparison, and I do not find 3.5.1 on P23 very helpful.  This section (3.5.1) could be put in the appendix and the main conclusion summarised in a much briefer section.

P10L270 section 2.4.3.  I suggest deleting this section.  The case that a different apodising function is used for measured and modelled spectra would be a gross error in the analysis and should never occur. If apodisation is applied in GFIT, it either matches that applied to the measured spectrum, or is applied internally to both and there is no mechanism by which it can be applied to one and not the other. There is little point in evaluating it.
On the other hand, there may be (and are) small instrument effects which make the ILS imperfect, and which could be viewed in a similar way to apodisation.  Such effects should be much smaller than the difference between an apodised and unapodised spectrum, even for a weak apodisation function.  A better ballpark measure of the effect of imperfect ILS would be to slightly adjust the instrument Field of View or phase error, which affect the width and symmetry of lines.

P12 Fig 2: This figure has limited value. It is impossible to distinguish the differences between the 4 databases through the four coloured traces, either through congestion, or poor choice of Y-axis scale. The most important information is in the different residuals (row 3). I suggest that the 5 single component spectra in each region, appropriately and comparatively Y-scaled to be easy to view, be moved to supplementary material/appendix or dropped, and Fig 2 replaced with only the CH4 spectra (4 columns), perhaps H2O as the major interferent, and one row each for the 4 residuals, having the same Y-scale in each window for comparability. The differences between databases would be much clearer in this view, supported by the statistics of Table 3. Ascension Is spectra do not need a figure in the appendix, comparable statistics to Table 3 would suffice (with a reason for the higher noise/poorer fits).

P16 Table 4. Rows 4 and 5 should read inter-window – see comment P7L188. If I understand correctly, this is the deviation across all windows, (intra-window would mean variation within one window, which seems meaningless).
From line 7, replace "b" with "bias".

Figure 5 could be removed to the appendix and just the essential differences to Tsukuba outlined in the text here. What is the reason for the higher uncertainties at Ascension?

Table 6 nicely illustrates the difficulty of retrieving useful information on $\delta^{13}$C in CH$_4$, given that we know a priori that the correct answer should be within 1‰ of -47‰. See earlier comments.

P24 3.5.2 I found this section long and difficult to assimilate. In Fig 9, what are the dotted lines? And in Fig 10, it is very hard to visualise or interpret the differences when both slope and intercept are varying and are co-dependent. It is much easier to interpret comparisons of a single metric, even if less rigorous. For example if the data of Figure 9 included the 1:1 line it would be much clearer to see any offset and slope differences and make an immediate comparative visual assessment. Fig 10 would not be necessary, but could be replaced by a table of comparative single statistics. Likewise Figs 11/12, 13/14 and 15/16.

In the case of the 2% CH4 bias, is this just a scaling of 2% to the apriori profile? I would not expect a large sensitivity in this case, because the retrieval also simply scales the a priori total profile without changing shape – only the initial starting point changes. Perhaps a better test would be to perturb the shape of the profile to assess the sensitivity, by changing the tropopause height or the tropospheric/stratospheric balance.

Sensitivity to water vapour is perhaps the most important here because water vapour is so variable.

P31 3.5.3. See earlier comment, I do not think this ILS sensitivity assessment is meaningful in practice in this form.

P35 L521: Long term averaging of $^{13}$CH4 measurements at a single TCCON site may well increase precision, but accuracy is more important and is not improved by averaging. Table 6 illustrates that the accuracy needs to improve by 1-2 orders of magnitude to be useful.

---

## Referee Comment (RC2) · Anonymous Referee #2 · 20 Jul 2020

The paper by Malina et al. reports on the consistency and quality of four spectroscopic databases for methane ($CH_4$) absorption parameters in the shortwave infrared spectral range. The evaluation tool is measurements of atmospheric absorption spectra collected in direct-sun geometry at high spectral resolution across a range of atmospheric path lengths and meteorological conditions. Topic-wise, the study is of interest to the readers of AMT since the respective spectroscopic parameters are relevant for ground-based as well as current and future satellite remote sensing experiments which aim at $CH_4$ column concentrations with sub-percent accuracy.

The study is based on state-of-the-art, high quality data. But, the paper is not written

well. As I already indicated in my technical review for the discussion phase, the paper provides too much specific information without guidance which piece is important and, on the other hand, the paper is too superficial in some of the important aspects. If to be published, the paper needs to be made more concise and to-the-point having in mind the interests of the atmospheric sciences community.

In that regard, I recommend dropping the entire 13CH4 discussion since it is essentially impossible (for the thinkable future) to perform a remote sensing experiment that yields errors low enough to make it useful for scientific purposes.

But I highly recommend discussing in more depth the following aspects. The key challenge of methane remote sensing is avoiding retrieval biases that correlate with geophysical conditions in a way that the errors mimic source (or sink) patterns. Reported and notorious candidates for such spurious errors are correlations between CH4 and H2O absorption (e.g. inducing tropical biases), correlations between CH4 and the lengths of the lightpath (e.g. inducing seasonal or high-latitude biases), and wrong CH4 profile shapes in the stratosphere. TCCON is known for requiring an empirical lightpath correction for low sun observations which, presumably, originates from deficient spectroscopic parameters. A meaningful evaluation of spectroscopic databases and retrieval windows requires addressing and quantifying these dependencies. The study here does not really include these aspects although it uses a range of spectra from two TCCON sites which could serve the purpose (e.g. plotting retrieval differences against H2O content or against path length). The study resorts to a sensitivity assessment with perturbed a priori and meteorological parameters which is not indicative for real atmospheric performance in particular since the perturbations are mere educated guesses. The most important sensitivity, the one to the stratospheric profile shape, is deliberately left out.

The required changes to the manuscript are at the edge of what seems possible under a major revision.

Specific comments:

P1 L18: What are "pressure cross sections"? Do you refer to the pressure dependence of molecular absorption cross sections?

P3 L72: "an average of three retrieved values from three TCCON fit windows" Why is CH4 separately retrieved from different windows and averaged a posteriori? If this is a simple arithmetic averaging, it tends to compromise the entire idea of using a maximum likelihood or an optimal estimation technique.

P3 L76: The discussion of the 13CH4 retrievals lacks the point that atmospheric gradients will be even smaller in column-average concentrations than in in-situ observations.

P4 L107: "least-squares optimal estimation" To the best of my knowledge, it is either least-squares (cost function without side-constraint, maximum likelihood principle) or optimal estimation (a priori departure as side-constraint, Bayesian statistics principle), not both together. From the list of retrieval parameters, it looks like the inversion problem is well suited for a pure least-squares technique. Optimal estimation would be required if the problem was ill-posed e.g. when retrieving concentration vertical profiles instead of column scaling factors.

P5 Table 2. Windows 1 and 5, 4 and 6 are only different because of technical details of the particular software used (apparently there can only be one target molecule per window). This is not of interest to the general reader and the discussion could be simplified.

P7 L175: The metrics should include the definition of the column-average mole fraction and its "X" notation. Later (section 3.3), the X-notation comes as a surprise.

P8 L210: Using CO absorption as a proxy for 13CH4 is inadequate, since precision and accuracy requirements for 13CH4 are factors higher.

P9 L256: I think the term "a priori errors" is misleading since not all parameters that are perturbed are retrieval parameters (for which an a priori exists).

The assumed perturbations for the parameters are not based on actual real-world problems reported previously. For example, simply assuming a 2% bias in the methane total column will not yield any significant error sensitivity because the retrieval of the total column is unconstrained. The a priori assumption that causes problems for the CH4 retrieval is the shape of the a priori profile e.g. assuming a mid-latitude stratospheric profile where there is in reality a polar vortex profile. Likewise a perturbation of the water vapor column by 10% cannot assess the actual real-world problem that spectroscopic interferences between CH4 and H2O might induce spurious gradients between the mid-latitudes and the tropics since the H2O content differs by factors, not just 10%. I would recommend replacing the entire perturbation study by a correlation analysis how CH4 retrievals from TCCON spectra (from Tukuba, Ascension Island, and possibly other stations to increase the parameter range) correlate with H2O content, CH4 stratospheric profile conditions, and path length dependencies and how these correlations are better or worse for the various spectroscopic databases.

P11 Fig.1: What is the spurious error (as a function of path length, season, latitude) that the difference of 12CH4 and 13CH4 averaging kernels could imply for delta-13CH4?

P12 Fig.2: The figure is much too small and contains unnecessary information such as the CO and HDO transmittances. The four spectroscopic databases are indistinguishable. I would recommend dropping all panels except for the residuals – if a residual is clearly correlated with a molecular absorber you can put a symbol on the residual peak.

P16 Table 4: The table is tough to understand. It would be helpful if the table indicated which window is 12CH4 and which one is 13CH4 since the definitions are far away. I recommend adding an extensive caption.

P21 Fig.7 and 8: Together with a very short and superficial discussion on page 22 the figures are largely useless (too small, point clouds masking each other). Drop for the sake of conciseness.

**[AMTD](...)**

Interactive
comment

P23 Why is the section called "a priori" error (similar question as for section title 2.4.2)? The section reports on the perturbation of the imposed temperature profile. Since temperature is not a retrieval parameter but a forward model parameter, there is no corresponding a priori. Probably, the title should rather be "parameter error".

P23 Fig 9: Units missing. Wouldn't it be the purpose of the sensitivity study to distinguish between the performance of the four databases with respect to temperature errors propagating into the retrieved CH4 columns? From Fig.9 I cannot find any conclusive hint in that regard. Fig. 10 is more useful.

P24 Fig 10: "perturbed pressure column retrievals" Isn't it temperature perturbation?

I do not comment on many editorial issues: acronyms need to be defined at first usage, jargon of the spectroscopic or TCCON community should be avoided, inconsistent use of parentheses when referencing, extensive use of the empty word "value".

---

## Author Comment (AC1) · 30 Sep 2020

The comment was uploaded in the form of a supplement:
https://amt.copernicus.org/preprints/amt-2020-86/amt-2020-86-AC1-supplement.pdf

---

## Author Comment (AC2) · 30 Sep 2020

Dear reviewer,

Thank you for reviewing our paper, and for the many helpful suggestions. In order to respond we have kept your original comments in black, our responses are in blue, and any proposed changes are in underlined blue.

The paper by Malina et al. reports on the consistency and quality of four spectroscopic databases for methane (CH4) absorption parameters in the shortwave infrared spectral range. The evaluation tool is measurements of atmospheric absorption spectra collected in direct-sun geometry at high spectral resolution across a range of atmospheric path lengths and meteorological conditions. Topic-wise, the study is of interest to the readers of AMT since the respective spectroscopic parameters are relevant for ground-based as well as current and future satellite remote sensing experiments which aim at CH4 column concentrations with sub-percent accuracy.

The study is based on state-of-the-art, high quality data. But, the paper is not written well. As I already indicated in my technical review for the discussion phase, the paper provides too much specific information without guidance which piece is important and, on the other hand, the paper is too superficial in some of the important aspects. If to be published, the paper needs to be made more concise and to-the-point having in mind the interests of the atmospheric sciences community.

Thank you for your comments here, please see our responses to the remainder of your comments which show how we have shortened the paper, while keeping the detail in the appendix.

In that regard, I recommend dropping the entire 13CH4 discussion since it is essentially impossible (for the thinkable future) to perform a remote sensing experiment that yields errors low enough to make it useful for scientific purposes.

Thank you for your comment here. Reviewer 1 suggested reducing the importance of 13CH4 retrievals in the paper. We have therefore removed most of the work relating to 13CH4 retrievals, but have included some of the results and some of the discussion related to the challenges of retrieving 13CH4. Despite these results showing the extreme challenge of retrieving usable values of 13CH4 at this time, we feel it is important to document these efforts. This will help any future studies investigating 13CH4 retrievals, allowing comparison between their efforts, and the capabilities at this time.

But I highly recommend discussing in more depth the following aspects. The key challenge of methane remote sensing is avoiding retrieval biases that correlate with geophysical conditions in a way that the errors mimic source (or sink) patterns. Reported and notorious candidates for such spurious errors are correlations between CH4 and H2O absorption (e.g. inducing tropical biases), correlations between CH4 and the lengths of the lightpath (e.g. inducing seasonal

or high-latitude biases), and wrong CH4 profile shapes in the stratosphere. TCCON is known for requiring an empirical lightpath correction for low sun observations which, presumably, originates from deficient spectroscopic parameters. A meaningful evaluation of spectroscopic databases and retrieval windows requires addressing and quantifying these dependencies. The study here does not really include these aspects although it uses a range of spectra from two TCCON sites which could serve the purpose (e.g. plotting retrieval differences against H2O content or against path length). The study resorts to a sensitivity assessment with perturbed a priori and meteorological parameters which is not indicative for real atmospheric performance in particular since the perturbations are mere educated guesses. The most important sensitivity, the one to the stratospheric profile shape, is deliberately left out.

Thank you for these important points, based on your (and reviewer 1's) recommendations we have restructured the section on the sensitivity analysis. The sensitivity analysis is split into two sections:
1. Firstly we have included the points you recommended above, namely the sensitivity of retrieved biases with respect to changing lightpath (solar zenith angle) and water vapour conditions over the course of a series of measurements at the Tsukuba and Ascension island TCCON sites. This analysis is performed by correlating water vapour and SZA changes with changes in retrieved 12CH4 biases, these results are included in the main text (section 3.5.1).
2. Secondly the majority of the original sensitivity analysis has been placed into Appendix C, the results of which have been summarised in the main text (section 3.5.2). We think that including the results of the sensitivity analysis is still important, these types of analyses are crucial in understanding the accuracy of retrieved values from satellites or ground instruments and are usually included in error budgets (Hu et al., 2016; Wunch et al., 2011). The point we make in this study, is that the error budgets calculated for sensitivity studies vary depending on the spectral window and the spectroscopic database, which is not normally considered in such studies and should be. We have modified the original sensitivity study, the aspect relating to instrument errors has been removed, and we have included a section relating to methane profile shape errors (as you suggested). For the profile shape errors, we used methane profiles from different seasons to induce errors due to stratospheric profiles and tropopause heights.

The required changes to the manuscript are at the edge of what seems possible under a major revision.

Specific comments:

P1 L18: What are "pressure cross sections"? Do you refer to the pressure dependence of molecular absorption cross sections?

Yes, pressure dependence was meant, however, we have re-written the abstract and pressure cross sections are no longer referred to.

P3 L72: "an average of three retrieved values from three TCCON fit windows" Why is CH4 separately retrieved from different windows and averaged a posteriori? If this is a simple arithmetic averaging, it tends to compromise the entire idea of using a maximum likelihood or an optimal estimation technique.

We missed a section of the documentation, which elaborates slightly. It is in-fact a weighted average, depending on the uncertainty of retrievals from each individual band. This sentence now reads:

....where a weighted average of three retrieved values from three TCCON fits windows.....

This is described in the Caltech TCCON wiki (https://tccon-wiki.caltech.edu/).

P3 L76: The discussion of the 13CH4 retrievals lacks the point that atmospheric gradients will be even smaller in column-average concentrations than in in-situ observations.

Thank you for this point, we have included discussion on the complexities of estimating 13CH4 column averaged gradients in section 2.3 the 'Analysis Criteria', we felt this was a more appropriate place for this discussion. The text in introduction is more for briefly introducing retrievals of 13CH4.

P4 L107: "least-squares optimal estimation" To the best of my knowledge, it is either least-squares (cost function without side-constraint, maximum likelihood principle) or optimal estimation (a priori departure as side-constraint, Bayesian statistics principle), not both together. From the list of retrieval parameters, it looks like the inversion problem is well suited for a pure least-squares technique. Optimal estimation would be required if the problem was ill-posed e.g. when retrieving concentration vertical profiles instead of column scaling factors.

Yes thank you, we have now described GFIT as:

...non-linear least squares fitting scheme....

P5 Table 2. Windows 1 and 5, 4 and 6 are only different because of technical details of the particular software used (apparently there can only be one target molecule per window). This is not of interest to the general reader and the discussion could be simplified.

We agree, this discussion has been simplified.

Please see updated discussion of Table 2.

P7 L175: The metrics should include the definition of the column-average mole fraction and its "X" notation. Later (section 3.3), the X-notation comes as a surprise.

Thank you for this point, we have decided to remove all X-notation from the paper. In this paper update, when we refer to retrieved trace concentrations we will refer to them as DMFs, for example "when we retrieved $^{12}CH_4$ DMFs we found".

P8 L210: Using CO absorption as a proxy for 13CH4 is inadequate, since precision and accuracy requirements for 13CH4 are factors higher.

We agree, the references to CO have been removed, and this discussion had been re-written as follows:

In terms of 13CH4 there are no published precision and accuracy requirements or statistics with TCCON. Fundamentally the final aim of retrieving 13CH4 is to calculate d13C. d13C has been used to differentiate between methane source types (Fisher et al. 2017, Nisbet et al. 2016, Rigby et al. 2017, Rella et al. 2015), and variations of this value has been linked with variations in the global methane budget (Rigby et al. 2017, Mcnorton et al. 2016). How much d13C varies in the total varies in the total column is a complex issue (Weidmann et al. 2017, Malina et al. 2018, 2019), in-situ studies (Nisbet et al. 2016, Rigby et al. 2017, Fisher et al. 2017) all show that an uncertainty of <<1‰ in d13C is required in order to determine natural annual variability at the surface. However, variability in d13C can be higher in the troposphere and stratosphere due to variability of the OH sink and the fractionation caused by OH (Rockmann et al. 2011, Buzan et al. 2016), with evidence that d13C can vary by up to 10‰ in different air parcels (Rockmann et al. 2011). Based on these factor, we assume a rough total column d13C variability of 1‰, which equates to a total uncertainty of <0.02 ppb on 13CH4 retrievals, or roughly 0.1% of the total column. This is clearly an unrealistic target for individual retrievals, given the uncertainty requirements for 12CH4 described above. Nevertheless precision errors will be low due to the nature of TCCON, and through the fact that TCCON sites are situated in a fixed position, allowing for long term averaging to reach a required precision target. Therefore one of the minor aims of this study is to identify how far away TCCON uncertainty (including systematic errors) is from the desired uncertainty of <1‰ d13C.

P9 L256: I think the term "a priori errors" is misleading since not all parameters that are perturbed are retrieval parameters (for which an a priori exists). The assumed perturbations for the parameters are not based on actual real-world problems reported previously. For example, simply assuming a 2% bias in the methane total column will not yield any significant error sensitivity because the retrieval of the total column is unconstrained. The a priori assumption that causes problems for the CH4 retrieval is the shape of the a priori profile e.g.

assuming a mid-latitude stratospheric profile where there is in reality a polar vortex profile. Likewise a perturbation of the water vapor column by 10% cannot assess the actual real-world problem that spectroscopic interferences between CH4 and H2O might induce spurious gradients between the mid-latitudes and the tropics since the H2O content differs by factors, not just 10%. I would recommend replacing the entire perturbation study by a correlation analysis how CH4 retrievals from TCCON spectra (from Tukuba, Ascension Island, and possibly other stations to increase the parameter range) correlate with H2O content, CH4 stratospheric profile conditions, and path length dependencies and how these correlations are better or worse for the various spectroscopic databases.

Based on your recommendation here we have undertaken the following actions:
1.  We now refer to "a priori errors" as "a priori and parameter errors", in order to distinguish between errors induced into the a priori profiles of methane and water vapour, and errors introduced into the profiles of temperature and pressure.
2.  We have now included a correlation analysis on how 12CH4 retrieval biases from Tsukuba and Ascension island correlate with local water vapour and solar zenith angle conditions, see section 3.5.1. This analysis shows that certain retrieval bands are sensitive to bias changes at high solar zenith angles. It also shows some sensitivities to water vapour variability, but only at high solar zenith angles.
3.  We have decided to keep the section on perturbation errors, we disagree with your statement that this is not based on real-world problems, indeed errors present in a priori and parameter profiles are included in the TCCON error budgets (Wunch et al., 2011). So it is possible that TCCON retrievals will have to deal with errors in the a priori profile of water vapour, and sensitivity to variability over the course of a series of measurements. We base our error magnitudes on what is expected for Sentinel 5P/TROPOMI retrievals (Hu et al., 2016). We have also included additional sensitivity analysis on incorporating incorrect methane atmospheric profiles into the retrievals, in order to understand the effect.

P11 Fig.1: What is the spurious error (as a function of path length, season, latitude) that the difference of 12CH4 and 13CH4 averaging kernels could imply for delta-13CH4?

Below we show the averaging kernels for a separate set of retrievals from Tsukuba and Ascension island, captured under different solar zenith angles and conditions. We note that the Tsukuba averaging kernels are very similar to those of Figure 1, this suggests that retrievals from Tsukuba show similar sensitivities with different retrieval types. The Ascension island averaging kernels suggest little change to 13CH4, but some changes in sensitivity in the lower troposphere for 12CH4. This implies that no additional spurious errors are caused by the differences in the 12CH4 and 13CH4 averaging kernels due to changing path lengths and seasons changing sensitivities, except those that already exist. Since the sensitivity of 12CH4 in the lower troposphere is very high in both cases, and the sensitivity of 13CH4 does not change.

Given that differences between the 12CH4 and 13CH4 averaging kernels do exist, this means that any changes in methane concentration in the lower troposphere will not be fully represented in the 13CH4 retrieval, thus causing bias. The averaging kernels for 13CH4 windows show slightly more difference in Ascension island than Tsukuba, meaning lower tropospheric changes will be represented slightly better in window 6 than in window 5 in Ascension island.

[Figure]

P12 Fig.2: The figure is much too small and contains unnecessary information such as the CO and HDO transmittances. The four spectroscopic databases are indistinguishable. I would recommend dropping all panels except for the residuals – if a residual is clearly correlated with a molecular absorber you can put a symbol on the residual peak.

Reviewer 1 also requested substantial changes to this figure, the following changes have been made to this figure.

All background absorber panels have been removed, only the transmissions of 12CH4 and 13CH4 have been kept, in order to indicate the differences between the spectral windows. Only the transmissions calculated from the TCCON spectroscopic database are included in order to reduce clutter. From this point only residual transmissions are shown, each row shows the residuals from a specific spectroscopic database.

[Figure]

P16 Table 4: The table is tough to understand. It would be helpful if the table indicated which window is 12CH4 and which one is 13CH4 since the definitions are far away. I recommend adding an extensive caption.

We agree, we have therefore identified which window is which, and expanded the caption for the tables.

P21 Fig.7 and 8: Together with a very short and superficial discussion on page 22 the figures are largely useless (too small, point clouds masking each other). Drop for the sake of conciseness.

We have removed this figure from the paper.

P23 Why is the section called "a priori" error (similar question as for section title 2.4.2)? The section reports on the perturbation of the imposed temperature profile. Since temperature is not a retrieval parameter but a forward model parameter, there is no corresponding a priori. Probably, the title should rather be "parameter error".

Section 3.5. has been renamed as "Sensitivity Analysis" which has been split into two parts. The first part section 3.5.1. named "Local condition variation" deals with the impact of varying solar zenith angles and water vapour on biases. The second part, section 3.5.2. has now been named "A priori and parameter errors" in order to indicate that we a perturbing the a priori and parameter profiles.

P23 Fig 9: Units missing. Wouldn't it be the purpose of the sensitivity study to distinguish between the performance of the four databases with respect to temperature errors propagating into the retrieved CH4 columns? From Fig.9 I cannot find any conclusive hint in that regard. Fig. 10 is more useful.

We have placed units in the titles of Figure 9, which is now in Appendix C. In order to address your comment about distinguishing between the performance of the perturbed and non-perturbed we have now included tables indicated the

biases between spectroscopic databases as well as the biases between windows. These results are compared against those identified in sect 3.3 in order to comment on the direct impact of implementing these a priori/parameter errors. These results are discussed in sect 3.5.2.

P24 Fig 10: "perturbed pressure column retrievals" Isn't it temperature perturbation?

Thank you yes, this has been corrected. This figure has also been moved to Appendix C.

I do not comment on many editorial issues: acronyms need to be defined at first usage, jargon of the spectroscopic or TCCON community should be avoided, inconsistent use of parentheses when referencing, extensive use of the empty word "value"

We have carefully re-read the paper to address your comments.

References

Hu, H., Hasekamp, O., Butz, A., Galli, A., Landgraf, J., Aan De Brugh, J., Borsdorff, T., Scheepmaker, R. and Aben, I.: The operational methane retrieval algorithm for TROPOMI, Atmos. Meas. Tech, 9, 5423–5440, doi:10.5194/amt-9-5423-2016, 2016.
Wunch, D., Toon, G. C., Blavier, J.-F. L., Washenfelder, R. A., Notholt, J., Connor, B. J., Griffith, D. W. T., Sherlock, V. and Wennberg, P. O.: The Total Carbon Column Observing Network, Phil. Trans. R. Soc. A, 369, 2087–2112, doi:10.1098/rsta.2010.0240, 2011.

---

## Referee Report (RR1)

**"On the consistency of methane retrievals using TCCON and multiple spectroscopic databases"**

This paper details the impact of using 5 different spectroscopic databases (GGG2014, GGG2020, GEISA2020, HITRAN2016, SEOM-IAS) on the retrieval of 12CH4 DMFs in 4 different spectral windows from TCCON measurements. The retrieval is performed with the GFIT retrieval algorithm from the GGG2014 environment. Retrievals are carried out on approximately a year of data at 4 TCCON stations (Ascension Island, Darwin, Ny Alesund, Tsukuba), chosen to have an optimal coverage of different atmospheric and measurement conditions (temperature, water vapour, SZA). A second objective of the paper is also to assess the impact of the spectroscopy and fit windows on the retrieval of 13CH4 (and indirectly, the d13C) and check for consistency of the results.

The scientific work is extensive, with a lot of data processing performed to represent retrievals under various atmospheric and measuring conditions. The presentation and analysis of the data is sometimes a little laborious, since so many different parameters are considered (spectroscopic database, retrieval window, TCCON sites). Still, the paper mostly presents the results adequately and underlines the important results. The results are scientifically relevant and interesting, and are certainly within the scope of AMT.

However, the paper sometimes lacks in style and clarity. It is obvious that it has been through some re-writes and major changes, and it should be proof-read and improved in terms of flow and clarity. I would recommend publication of the paper in AMT pending some minor revisions of the manuscript, as detailed below. Note that I started to note down many typos or problematic sentences in the first half of the paper but was not as systematic in the second half and would recommend to proof-read this more thoroughly.

**Comments**

1/ The abstract should be improved.

    a)   The third paragraph is especially difficult to read. It would be helpful to be more quantitative in your description of the impact. The sentence "We also find strong evidence that different windows in different spectroscopic databases exhibit different levels of sensitivity to changing local conditions such as light path length and water vapour" is technically correct, but it is so general it becomes almost meaningless.

    b)   You should mention the work on the 13CH4 retrieval and d13C determination.

2/ The introduction should be re-worked to improve clarity and fluidity:

a) I find it strange that a paper studying the impact of the spectroscopy on the retrieval of methane does not directly discuss the spectroscopy aspect of the retrieval in its introduction. Maybe add 1-2 lines to describe how it is relevant?

b) L44-50 should be at the end of the introduction, detailing the content of the paper. Up until that point, methane isotopologues have not been mentioned at all, maybe add a line about this aspect of the work as well beforehand.

c) I am not certain of the necessity of Table 1. One could either have a table with more information (a graph of the respective instruments and their spectral coverage), or a sentence in the text? Since you discuss TCCON windows shortly thereafter, maybe it would be a good opportunity to describe the TCCON windows used for the CH4 retrieval there (instead of referring to Table 4)?

d) L62-66: Maybe I misunderstood the point being made, but it seems simplistic to suggest that the retrieval window differences between TCCON and TROPOMI may be the major source of biases in the validation of the methane product, especially when many other factors will affect the quality of the retrieval on both GB and satellite platforms. This is surely an interesting point to study, but this seems to inflate the importance of the retrieval window on the validation.

e) L71: "We can infer some of the potential spectroscopic related biases in satellite retrievals…": yes, but since the resolution of TCCON and satellite instruments are quite different, TCCON should be much more affected than satellite retrievals, maybe something to mention.

f) L75: I think that Eq. 2 should be moved here. [?]

3/ L119: "GFIT assumes a fixed profile shape for each trace gas, …": you should mention how the a priori is determined

4/ L198-201: Is there already an estimate of the errors due to these assumptions? It would be interesting to have some figures on this in the paper.

5/ Section 3

Before analysing the retrieval data, I would *suggest* to first perform a comparison of synthetic spectra using fixed state vectors with different spectroscopic databases to look at the expected impact in terms of transmissions in various windows. It might also be interesting to look at the contribution of H2O and CH4 to the total transmission, and hence have a better feeling for which windows should be more sensitive to the water content (would help with the analysis of section3.3?)

6/ Section 3.1

A slightly more exhaustive analysis of these results would be interesting. Could you associate some of the more prominent residuals with species? For instance, in window 3, HITRAN and GEISA have large discrepancies around 6001 cm-1 compared with GGG, why is that? Are there clear improvements due to the use of non-Voigt parameters?

7/ Section 3.3

In this part of the analysis, you sometimes seem to confuse correlation and sensitivity, which are quite different, or only consider one of these aspects. For instance, on L346 you state that "results from Ascension Island, which operates at lower SZAs than any of the other TCCON sites considered in this study also indicates large correlations, suggesting further complexity". The truth is that the sensitivity (slope) of the linear fit to SZA is close to 0, so despite a large correlation, very little effect is observed.

On L348-349: "For window 1, there is weak sensitivity to water vapour variations at the Ny-Ålesund and the Darwin sites," where the sensitivity is substantial for Ny Alesund in Window 1, but correlation is weak.

Section 3.3 could also gain from an indication of statistical significance in the results. It is mentioned on L385 that large p-values were found in some cases, I would suggest identifying clearly results that are not significant, and state at what level you consider these not statistically significant.

**Typos and technical corrections**

Comments with a "[?]" at the end are for your own consideration

L1: "methane retrieving satellites" -> "methane retrieving satellites instruments" [?]

L3: "is as important as when TCCON…" -> "is as important now as when TCCON…"

L6: the 'S' in "Spectroscopic databases" should not be capitalized

L7: Leave out spectroscopic databases in "TCCON GGG2014 and GGG2020 spectroscopic databases;", as it is redundant [?]

L14: consider changing "~x3" by "up to three times" [?]

L16: "vapour. Such" -> the period should be a comma, and the Such should be lowercase. But you should consider changing this sentence as it is very long and uses 5 times the word "different"

L24: "bottom up" -> "bottom-up"

L25: "top down" -> "top-down"

L28: "S5P" -> "Sentinel 5-Precursor (S5P)" , "TROPOMI" -> "TROPOspheric Ozone Monitoring Instrument (TROPOMI)" . These acronyms have been defined in the abstract, but consider adding a proper acronym description in the full paper as well. [?]

L40: "will therefore be relying" -> "therefore rely"

L43: "from the 6000 cm-1 spectral range": a range has two values. Either define the window precisely, or reformulate

L44: "make" -> "perform"

L48: "…quantify the variations in retrieval abundances when using five separate spectroscopic databases,…" : this seems like the main aspect of this paper, it should probably be mentioned first.

L50: "are studies" -> "are studieD"

L50: consider a reformulation of this sentence

L52: (an weighted …" -> "(a weighted…)

L81: Either state "98.9% and 1.1%", or "99% and 1%", but it should add up to 100%. Considering the "roughly" in the sentence, I would go with 99% and 1%.

L86: "structured as follows," -> change the comma to a colon

L90: "…conclusions are shown…" -> "conclusions are drawn" [?]

L100: Add a comma after "In this study"

L100-101: "which is summarised briefly here." -> "briefly summarised here."

L101: "A forward" -> lowercase "a"

L104: "fit" -> "fitted" [?]

L105: after "In the case of GFIT", add a comma

L106-115: I am not convinced that these are essential to the papers and may be omitted. Maybe reconsider listing these, and put a reference instead [?]

L116: confusing wording: "Note that not all of the above are not routinely…" I imagine that you should omit the second "not".

L116: Start a new sentence with "For example" [?]

L124: "the 7885 cm-1 spectral range multiplied" -> a spectral range needs 2 numbers. Maybe use a formulation such as "in the vicinity of 7885 cm-1…" [?]

L127: Not sure if it's necessary to state the resolution again, which was already mentioned on L57

L131-132: not sure if this sentence needs commas after "window" and "study" or if they are optional [?]

Table 4 has a beginning o f a sixth row

L135: is there a reference for the "standardised process"?

L137: this sentence should not be its own paragraph

L158: "Further to exploring the…" -> maybe "Beyond exploring …" [?]

L167: "line mixing" -> "line-mixing"

L169: "They find a 1.1% difference in total methane…" -> is it a positive or negative bias? Maybe just add a "+" to emphasize [?]

L184: "standard methane window" -> either "standard methane windows" (there are 3 of them, right?) or, better yet "standard TCCON CH4 product"

L185-189: These are technically correct sentences but are a bit long and wordy. Maybe consider adding equations ? [?]

L194: This sentence should not start a new paragraph, but simply continue the previous one.

L196: there should not be a period before the "2)". Maybe a semi-colon?

L197-198: "apriori" or "a priori" -> be consistent, I have seen both in the paper

L202: "cross sections" -> "cross-sections"

L204-207: consider re-writing this sentence, it's a bit convoluted. Maybe with more than one sentence?

L208-209: you probably should not hyphenate an acronym

L213: leave out "metric"

L214: you repeat "varies in the total"

L216: "<<1‰ in…": are these French quotation marks (<<) or is it the symbol "much smaller than"? If it is the latter, it should be bigger, if it's the former, the close quotation mark is missing. Also, is this figure for a single measurement? monthly mean?

L221-223: "Nevertheless precision errors will be low due to the nature of TCCON, and through the fact that TCCON sites are situated in a fixed position, allowing for long term averaging to reach a required precision target." -> This sentence is unclear (especially the second part: "through the fact…"). Also, it should be "long-term", not "long term".

Figure 1: Would it be possible to plot or mention the instrumental uncertainty along the residuals?

L238: "shown in Fig 2," -> "presented in Fig 2," [?]

L239: "Fig 2 is the fit" -> "Fig 2 is that the fit"

L247: Maybe start a new paragraph with "There are differences in the…" as you focus more on the TCCON sites now, not on the windows?

L256: "Likely reasons" -> for what? If you refer to the last paragraph, you should stay in that paragraph, and state more clearly what you mean.

Figure 2: Maybe indicate in one plot (window4?) the number of spectra used for the statistics? I know this information is available in Table 2, but this is always nice to have this information on hand.

Figure 2 caption: maybe indicate the colour of each statistics in parenthesis: "Each subplot shows the RMSE (black) and chi2 values (blue) …"

L263+264: "fit" -> "fitted" [?]

L263: "there no specific" -> "there are no specific"

L267: "GESIA" -> "GEISA"

L270: "the the" -> "that the"

L270: "retrievals each database" -> "retrievals for each database"

L274-275: This is not a complete sentence, there seems to be missing something (maybe Firstly we present the results?)

L289: "from in" -> delete "in"

L290: "the standard in" -> "the standard retrieval in"

L304: "sides" -> "sites"

L327-330: This sentence encapsulates nicely the differing atmospheric conditions of the various TCCON stations. Maybe it should be moved to the introduction to better explain this part of the work [?]

Figure 8: This figure is not very easy to read, but I am not quite sure how to improve it. It does give a good overview of the results. Maybe instead it could be a pcolor-type graph for the slope of the linear fit and the intercept for each station and parameter sensitivity (each row of the graph could be a different spectroscopic database, each column a different window?). Just a suggestion

Figure 8 legend: "GEISA", not "GESIA"

L334: The description of Figure 8 is really not clear. Could you be more precise in what is being shown in the figure instead of "the qualitative distributions"?

L342: "indicate the retrieval" -> "indicate that the retrieval"

L345: "creep" -> maybe find a better word [?]

L349: "opposite databases" -> what are opposite databases?!

L362: "vapour. Thus" -> should be in the same sentence "vapour, thus"

L396: "in the calculation" -> its more in the analysis of the results than the actual calculation of the value…

Table 5: The table should be slightly improved in such a way that there is less text. Maybe use sub-rows for each station, one for each database (basically like it's done now, but just mention the database name once, not in each cell…). Also, would it be possible to either add the uncertainty on the d13C value, or mention an order of magnitude in the text? This would add important information to the discussion of L398-401 about the uncertainty.

L398: "that we calculate the mean uncertainty" -> "that the mean uncertainty…is between…"

L404: "show results" -> "are"

L407: "combination" -> "combinations" [?]

L420: "Further to this" -> "Furthermore" [?]

L447: "question that is of some interest to the community, "can we calculate realistic and constant 13C values from TCCON"." -> this formulation is a bit strange to me, maybe use colon after community (instead of a comma), or change the formulation to something like "question that is of some interest to the community, namely whether it is possible to calculate realistic and constant 13C values from TCCON."

L453: "calculate" -> "calculated"

L459: "this is an assumption based" -> "it is based"

---

## Referee Report (RR2)

By taking into account several suggestions made by the referees on the previous version, the quality of the revised manuscript now seems significantly improved. I therefore support publication of the manuscript in its present form.

I need to note that I do maintain my doubts concerning the argument raised by the authors in reply of my previous comment concerning the use of ground pressure for constructing the airmass reference. The authors state "However, we disagree with the approach of using the surface pressure and water vapor columns to calculate the VMRs ... we [would] introduce [site-to-site] biases from the pressure measurement."

It should not pose any difficulty to measure ground pressure within 0.5 mbar (in my understanding, the collection of a reliable ground pressure record at each site within this accuracy range is a prerequisite for operating the TCCON). So the reference airmass constructed via ground pressure is reliable on the ~0.5/1000 = 0.05% level. There is in addition a higher-order error via the spectroscopic determination of the water column, but TCCON should be capable of measuring this variable ~2% contribution to the total atmospheric column with an accuracy of ~ 2%. This gives rise to an additional uncertainty contribution to the ground-pressure approach of 2% \* 2% = 0.04%, so the resulting uncertainty budget of this approach becomes ~0.1%.

If this level of consistency is compared with the XAIR time series from the individual stations included in this work, then it is found that while the retrieved GGG2020 XAIR for Darwin is near unity, it ranges slightly lower for Tsukuba and Ny Alesund ~0.98 ... 0.99 and still lower for Ascension ~ 0.97. This is a scatter in the order of 1%, so an order of magnitude higher site-to-site bias than the estimated uncertainty for the route via ground pressure. I understand that errors in the target gas column and the oxygen column (their ratio being used for constructing Xgas) are expected to be significantly correlated, but the correlation needs to be very tight in order to be en par with the ground-pressure approach. For this reason, the discussion of the ground pressure approach would be appropriate in a spectroscopic study (1) for assessing the absolute band intensities and (2) for separating out airmass dependent disturbances introduced via the oxygen retrieval.

---

## Author Response (AR2)

Dear Editor,

Thank you for the current handling of the manuscript, please find our responses to your and the reviewers criticisms below. We have kept your comments in black, our responses are in blue, and specific changes are in underlined blue.

Dear Authors,

The referees have given their feedback on your manuscript, with 2 referees raising substantial criticism and insisting that publication of the manuscript in the present form is not possible. They recommend major changes for reaching publication quality.

While your investigation and the results are certainly interesting for the readers of AMT and especially the remote sensing community, I agree with the more critical reviewers' views that there remain major methodological questions which need to be addressed before the science case can be made. Likewise, the manuscript lacks clarity and is sometimes difficult to follow. Significant changes to the paper are therefore required before publication in AMT can be considered. Consequently, I decide that the paper can be reconsidered after major revisions.

Pleaase take all referee suggestions into account when improving the paper. Scientific significance and scientific quality are two core criteria for publication in AMT. It is therefore important that the methodology is improved and well explained. As outlined by the referees, there are some major questions to be addressed:

We hope our latest changes answer the current criticism.

1) How representative are the data that you show and do they support the conclusions? Why has the investigation been restricted to only two days at two different sites? Why does one day only cover less than two hours of measurement? Would observations at other network sites lead to the same or similar conclusions? Your study would gain much from including more sites and extending the period of observation and this is urgently recommended. It is difficult to understand why the investigation does not include more sites and a more significant time span in the first place.

We agree with this assessment and have now expanded the amount of data used, specifically we have expanded the amount of data used at the Ascension Island and Tsukuba sites to capture variations over the course of a year, represented by >1000 individual spectra in each case. We have also added results from two additional TCCON sites, Ny-Ålesund and Darwin, both with significantly different local conditions.

2) It is claimed that 'significant' biases between databases are observed, but this conclusion is partly due to ignoring database uncertainties. If I am not mistaken, methane line intensities in

HITRAN already have 20% relative uncertainties, for example. Given these high values, it is rather surprising that the agreement between databases is better than a few percent. A comparison of databases, such as in this manuscript, certainly needs that related uncertainties are discussed. Moreover, a bias is significant when it exceeds 2 or 3 standard uncertainties (depending on the desired probability level). Given the TCCON sigma values in Tables 4 and 5 and assuming a similar sigma for the other databases, the observed differences are not statistically significant even if the intensity uncertainties in the databases are neglected. But this is not obvious without attributing actual uncertainties to the observed bias values.

Thank you for these comments, we have decided to overhaul how we represent our biases. We now use Absolute mean errors and Root mean square errors with respect to the standard TCCON methane retrieval window from GGG2014. These values are normalized by the retrieval errors in this standard window, which include noise and forward model errors (etc). We therefore define any biases as being significant when they are larger than 1, thus indicating a bias larger than the error associated with the standard TCCON retrieval. Any values lower than 1, we assume as insignificant.

3) While the discussion of the 13-CH4 isotopologue takes a lot of space despite the conclusion that meaningful isotopologie retrievals cannot presently be made, it is not discussed quantitatively how much the inclusion of the new TROPOMI window possibly improves the isotopic retrieval and how the very different averaging kernels impact on determining the delta.

Thank you for this comment, we have now generally reduced the amount of space given to 13CH4 retrievals, however we do retain this analysis and have expended upon it (please see the new section 3.4). We discuss how 13CH4 retrievals using the TROPOMI window are far more constant and show less bias away from the accepted tropospheric average than 13CH4 retrievals using the standard window.

4) Presentation and discussion of the CH4 bias sensitivity (towards other parameters, such as water column, SZA, ...) is lacking clarity.

We have fundamentally overhauled this section (please see updated section 3.3), it has been cut down and simplified and is now easier to follow.

- Table 8, for example, displays a coefficient of determination and not a linear relationship and I would expect that it is $R^2$ between the bias (y) and the vapour column (x) (and not vice versa). The use of $R^2$ as a figure of merit should be explained in more detail. While a value of $R^2 > 0.5$ indicates that a variability of a parameter y can primarily be explained by the variability of parameter x, a value of 0.2 still indicates that y somewhat depends on x and that there is an interference.

This table has now been removed, and we use Pearson's correlation coefficient as opposed to the $R^2$ values to determine the existence of a linear relationship.

- What does it mean when you write (p 18) that relationships in Table 8 are unlikely to be purely down in water vapour?

This particular line has now been removed, however we do discuss in our updated section 3 how similar linear dependence is observed for water vapour and temperature variations. What we mean is that it may be difficult to completely disentangle the impact of water vapour and temperature dependences on the retrieval biases, since both water vapour and temperature and climb and descend during the course of a day.

- The conclusion at the end of the paragraph is also not clear and the claim that 'As with Table 8 the results from window 1 show the weakest relationships' is not entirely true. It seems that TCCON window 3 has almost always the lowest $R^2$, indicating that it is most consistent with TCCON window 4 - SEOM window 1 comes next.

This particular section has been removed.

How do you distinguish between systematic and undesirable correlations (due to spectroscopy, for example) and accidental correlations (spatial variation of the CH4 column that possibly correlates with SZA, etc.)?

We have tried to minimize such possibilities by performing the analysis using data over a whole year, and including 1000s of individual measurements. So that accidental correlations are averaged out over the long time period.

Finally, how do you explain that the bias depends on the water mixing ratio? Is there any hypothesis?

In our new section 2.4 we discuss how bias can depend on water vapour variability, largely due to differences in the apriori profile, the reality and the fact that GFIT is a scalar fit algorithm as opposed to a profile fit algorithm, potentially enhancing errors in the profile shape.

5) The overall presentation should be improved.

Besides these points that need thorough consideration, there are a few minor comments that should be looked at:

- The abstract talks about specific TROPOMI windows (plural form). In the text, there is only one such window, apart from the fact that the same window is given a different name when it concerns another isotopologue.

Thank you, this has been corrected.

- Bias values such as in Tables 4 and 5 should be given with sign and with uncertainties (standard deviation or error of the mean). Otherwise the numbers are difficult to interpret, especially since the numbers of digits shown falsely imply a very small uncertainty.

We found that the original set of metrics were confusing for the readers, and have removed these from the paper and have focused on two, identifying the bias w.r.t the standard TCCON methane window, normalized by the noise. These metrics are no-longer included in tables, but shown via bar charts which indicate relative differences more clearly.

- Why has there been only one hour of measurements been selected for Figure 3? Are there no other measurements on the same day? For studying a SZA dependency days with better coverage would be more favorable. Figure 3 also is the only instance in the paper, where windows 1 to 4 are designated by the centre wavenumber. This is confusing. Also why do many measurements come in pairs with gaps in between?

This figure has now been removed and replaced.

- Tables 4 and 5: it seems that GEISA yields the highest consistency in the inter-window comparison. This is an interesting point that requires diccussion.

This metric has now been replaced, and we do discuss instances of consistency in the databases in the new version of the paper.

- Section 3 lists a lot of results, but very little conclusions are drawn from the derived quantities. Often the message is missing. What does it mean when you write 'Windows 5 and 6 vary in their sensitivity, with the different spectroscopic databases showing similar sensitivity in window 5, but very different ones in window 6.'?

We have drawn further conclusions at the end of section 3, identifying that the variability of local conditions causes larger biases between spectral windows and spectroscopic databases.

- I am not convinced that there are 'significant' differences between different windows used for the retrieval of methane isotopologues.
Taking sigma_window of ~ 1.5 - 4 ppb for 13-CH4 as an uncertainty of measurement, an uncertainty on the order of 75 - 200 ‰ can be derived. This is not so much different from the observed variability in Table 7. Again, your definition of significance needs to be explained.

We have updated our definition of 'significant' differences, by normalizing biases by the retrieval noise. We therefore assume that bias values that exceed the

- One of the delta values in Table 7 is not compatible with the bias values shown in the Tables. Please check all values carefully.

This table has now been removed, and new values have been checked.

- Explain the choice of the 4th TCCON window as a reference. Wouldn't it be more consistent to use some average of TCCON windows as a reference? Or can the other windows be neglected as they are much narrower? The reader would likely be interested to know how each of the three windows contributes to the overall CH4 data product.

We agree with this comment, we now use the standard TCCON methane window from GGG2014 (a weighted average of the three standard windows) as our reference. We explain in the paper that this standard is a weighted average of the other windows, but TCCON literature does not elaborate on how these weighting are derived.

- The comparison of Tables 8 and 9 should not start by mentioning a close agreement, because the Ascension data from August are showing a quite contrasting picture. It is striking that the discussion does not discuss this difference.

These tables have now been removed from the paper.

- The correlation plots C9 etc. show discontinuities, most prominently for the 13C containing isotopologue. This is not really discussed.

This section has been removed, we found this was a bit of an information overload.

- Why is the Tsukuba April 2016 the reference in section 3.5.2 and Appendix C ? It has the smallest number of data points.

We have significantly overhauled the Tsukuba data, so this comment no longer applies.

- One of the most remarkable results is that inclusion of window 5 seems to reduce the uncertainty of 13CH4 retrieval. Could this improvement somehow be quantified?

We found the retrieval errors generated by TCCON for 13CH4 in the 6076 cm-1 window were double those of the 4265 cm-1 window, which will go a long way towards explaining the improvement in the d13C retrievals.

- Insignificant digits in Table 4 should be removed.

We now show all results to 3.s.f

Reviewer 1

Dear Reviewer,

Thank you for your updated comments, the paper has now been significantly overhauled due to reviewer comments, but our responses are below in blue.

The re-submitted manuscript is now much shorter and more readable. I recommend publication subject to a few technical corrections:
L81: d13C metric which is based on a ratio of these isotopologues (it is not a simple ratio)

Thank you, we have updated this sentence and removed references to ratio.

L87: the abundance of 13C is 1.1% and D is only 0.01% so the abundance of 12CH4 is closer to 99% than 98%

Thank you, corrected.

L313: This referal to Appendix A is perhaps to brutal, could you add a setance summarising the expalnation referred to App A?

Appendix A has been updated in the paper overhaul.

Table 8 caption: as not at? The meaning behind this caption is not clear.

The original Table 8 has been removed in the updated paper.

L446: This would not be a not a significant .... please correct.

This line has been removed due to the paper update.

Reviewer 2

Dear Reviewer,

Thank you for your comments, we have kept your original comments below, and we have added inline responses in blue.

The authors did a thorough job considering my comments from the previous stage of review. I still have some substantial concerns that I recommend addressing before publication:

Major comments

P11, section 3.1: I think the averaging kernel discussion falls short of what the problem for delta-13CH4 retrievals is. The averaging kernels for 12CH4 and 13CH4 are substantially different. That implies that a 13CH4 "anomaly", for example, close to the surface would be weighted differently in the total column than a proportionate 12CH4 anomaly. The difference in weighting would result in a spurious anomaly of the calculated delta-13CH4. My question from round 1 was what the magnitude of this averaging kernel effect on delta-13CH4 is and whether the effect is actually smaller than the targeted 1 permille (or how the effect is related to the huge variability in Table 7). I would think, that this presents a fundamental challenge in total column remote sensing of isotopologue ratios limiting the usefulness of delta-notations for that purpose. I recommend placing a caveat related to these considerations.

Thank you for this comment, we however have removed the Averaging Kernel discussion from this updated paper, since we were not sure as to the benefit of its inclusion and we have reduced the emphasis on 13CH4 retrievals in this update.

P17, Table 7: The consistency among the delta-13CH4 retrievals is awful and nowhere close to the required quality. What is the retrieval uncertainty of the delta-13CH4 values, what is the daily standard deviation? I actually recommend postponing a discussion of spectroscopic effects on delta-13CH4 to a stage where the retrieval procedure for 13CH4 will have been consolidated.

The retrieval uncertainty on 13CH4 retrievals is typically around 18-25% of the total column (so very large), this is a mean value over the entire available dataset in the new paper. However in the updated paper we provide d13C values averaged over thousands of retrievals, so the uncertainty on these values should be significantly reduced. We indicate this in section 3.4.

P18, Section 3.5: I have some concerns wrt. tables 8 and 9 and the related discussion:

Tables 8 and 9 have been removed in this current iteration of the paper.

- The coefficients of termination are substantial in many cases (I would consider r^2>0.2 substantial). So, I would object the statement that "Table 8 generally indicates that there is limited or no relationship …".

Thank you for this comment, however we have removed the use of coefficient of determination and use Pearson's coefficient instead, and we have overhauled this section of the paper.

- The coefficients have been calculated per day per station and it is the coefficients for the concentrations per se, not for concentration differences (e.g. wrt. a reference window). This probably implies that the parameter range (in particular for water vapor) is very small questioning the robustness of the calculated values.

Thank you for this important comment, we have now massively expanded the amount of data we are using, and make these comparisons against a reference, so these values are now much more robust.

- I also wonder about coefficients >0.5 for window 4. It essentially indicates that there is a very strong correlation which, however, could be due to spectroscopic errors or due to an actual correlation in the atmosphere on the particular day.

Thank you, similar to the comment above, we now make these comparisons using data over a whole year, so we have reduced the dependency on correlations on a specific day. However, w.r.t. the spectroscopic errors, these are the types of errors that we are trying to identify by making these window comparisons.

- I recommend evaluating a dataset that has substantial dynamic range in H2O (and SZA). Why did the authors actually choose Tsukuba and Ascension – there would be stations with more water variability (e.g. Darwin) and more seasonality (e.g. Park Falls)?

Agreed, we have now included two more stations, Darwin as suggested for water vapour, and we also included Ny-Ålesund as a contrast, with low water vapour and high SZAs.

- I would also recommend instead of showing tables of coefficients showing correlation plots that illustrate the parameter ranges and visually depict potential correlations. For example, there is a longstanding discussion of the airmass dependent bias in TCCON XCO2 (and related corrections, e.g. appendix E in Wunch et al., https://doi.org/10.1098/rsta.2010.0240, 2011). Is there a similar bias observed for XCH4 and does it dependent on the spectroscopic database and the retrieval window?

Thank you for this, we have included plots visualizing this (see Figure 5), but we found it difficult to determine correlations from this plot, so we have retained tables including r values for reference.

- It might also be useful to pick the window with the smallest correlations as a reference window and then, evaluate differences of concentrations.

We now use the standard methane retrieval window as our reference point for all calculations.

Minor comments

Throughout manuscript: It looks like a various places the revisions have not been carefully checked for correctness of replacements. Please double-check the entire manuscript for syntax and wording issues and make sure that there is a thorough proof-reading. (I stopped marking the most obvious syntax issues somewhere in the middle of the paper.)

Title: remove „isotopologue". While technically the usage of the word is of course correct, it is misleading in a sense that the reader might expect a focus on the rare isotopologues since when discussing the main isotopologue 12CH4 the word "isotopologue" is commonly not used. Along my previous comments and also consistent with the authors' revisions, the focus of the paper is clearly on the main isotopologue.

We agree, the title has now been modified to:

On the consistency of methane retrievals using TCCON and multiple spectroscopic databases

P1,L10: "We report …" - Sentence is very complicated and long (and maybe wrong grammar). Consider rephrasing and splitting.

Thank you, we have now changed the abstract and this sentence is no longer present.

P2,L21: "depending on the introduced error" – I do not really get the meaning of "introduced error": interferences with SZA, water vapor und profile shape are not simple hypothetical errors. They come from the actual dataset.

Thank you, we have removed this section from the paper.

P2,L27: "is not as well understand as is the case for carbon dioxide" – What is the metric for this comparison, is there a reference to be cited? Considering that the year-to-year variability of carbon uptake by the land biosphere is on the order of 10-20% of the anthropogenic emission total and that this variability is not really understood even on the global scale, I would remove such statements. In fact, methane with a single sink process dominating might even be considered the much simpler case.

We accept the generic statement of "not as well understood as CO2" is not useful, we have replaced with:

but the processes via which it enters and is removed from the atmosphere are still not well understood.

P2,L35: GOSAT is not mentioned in the list. Why is that? GOSAT has been operating since the year 2009 delivering probably the best methane retrievals ever available. Due to its wide spectral coverage it allows for proxy as well as full-physics methane retrievals and, GOSAT-2 covers the 4300cm-1 band. So, GOSAT is the gold standard for methane. It needs to be mentioned.

We had not mentioned GOSAT because it does not uses the 4265cm^-1 spectral range, a particular point of focus of this paragraph. However, we have now identified both GOSAT and GOSAT-2 in the paragraph.

P2,L45: "with SCIAMACHY" -> "with SCIAMACHY and GOSAT"

Changed, thank you

P2,L46: "season, assess" – Syntax.

Sentence changed to read as:

Spectra are taken from four different TCCON sites in order to assess the impact of varying atmospheric conditions at different global locations.

P3,L61: The main reason why TCCON is better quality than satellite data is the fact that direct sun viewing safely allows for neglecting atmospheric scattering processes. Thus, the lightpath through the atmosphere is known with much better accuracy than for the satellite sensors.

We have added this additional clarification.

P3,L65: "generating using" – Syntax.

Corrected to generated.

P3,L67: "accurate/stable" – What does "stable" refer to?

Corrected to:

window proves to be as accurate as the standard TCCON windows,

P3,L69 "will may use" – Syntax.

Removed

P3,L75: "Our finding will inform … and are …" – Syntax.

Sentence has been removed.

P3,L83: "industrial" – What does industrial refer to? Is it the fossil fuel production chain?

Yes, we have changed industrial to fossil fuels to be clearer.

P4,L113: "This synthetic …" -> "These synthetic …"

Corrected.

P5,L124: "Note that all of the above are not …" -> "Note that not all of the abover are …"

Corrected.

P6,L145: "The introduction of 13CH4 …" – This reads like the 13CH4 retrievals were the cause for comparing the spectroscopic databases. I think the study SEOM-IAS is actually much more important for the purposes of this paper.

We have rephrased this paragraph to identify the recent addition of 13CH4 as a point of interest.

P10,L263: "scalar retrieval algorithm" -> "scaling retrieval algorithm"

Corrected.

P12,L302 "line mixing/overlapping" -> "overlapping".

This section of the paper has been rewritten

P12, Fig.2: There is still substantial noise in the fit residual plots. Are these individual measurements or is it the co-added residuals from various measurements? If the former, would it make sense to co-add residuals?

These are individual retrieval plots, and while we do agree that co-adding would improve the noise, this is not typically done in TCCON retrievals, and we are trying as much as possible to replicated typical TCCON operations.

P13,Fig.3: The data at ~9.55 UTC (X12CH4 plot, upper panel) and ~9.75 UTC (error plot, lower panel) show substantial excursions. Why is that?

This figure has been changed in the new version of the paper.

P19, section 3.5.2: Following the recommendations in round 1 of review, the authors essentially moved all figures and tables related to that section into the appendix but kept a quite detailed discussion. If the discussion is really required and needs to stay in the manuscript, it would be good to have at least one summary figure or table in the main manuscript where the reader can refer to when going through the discussion.

These sections have been removed from this version of the paper.

P21,L450: The degrees of freedom for a profile retrieval of CH4 from TCCON will be small and thus, there will be a residual dependence on the a priori shape. I think a statement claiming that such an algorithm "would therefore not be subject to the methane profile errors" is too optimistic and hard evidence would be needed to convince me of that (quite general) statement.

We have changed this to read as:

Be less subject to methane profile errors

Reviewer 3

Dear Reviewer,

Thank you for your review, we have keep your original comments below in black and our responses in blue.

+ include more sites and motivate the choice of the sites, the authors should demonstrate that they encompass the range of possible situations for atmospheric methane, being therefore more useful to, e.g., the TCCON and satellite communities

We agree with this comment, we have included two more TCCON sites that represent a wide range of conditions such as SZA, temperature and water vapour. We elaborate on our choices in section 2.1.

+ do not limit the data sets to a few consecutive measurements, a day, or one or two months, especially if the underlying motivation is unclear or not provided. The resulting metrics will be more representative and robust

We agree, we have now included a significantly larger number of measurements from each site considered, typically >6000 over the course of a year.

+ try identifying more suitable spectral intervals for the 13CH4 retrievals; new spectroscopy has become available and more relevant windows might potentially be selected

While we agree that there is certainty room to improve on choosing spectral windows for 13CH4, we do not feel this paper is the correct study for determining the best windows for 13CH4 retrieval. This should be elaborated on in a separate study.

+ include model simulations or results (e.g., from Buzan et al., AMT, 9, 2016) in order to estimate a reasonable value for delta13C representative of a column measurement, this will be less speculative than the current approach relying on in situ surface measurements.

We agree that a more reasonable estimate of d13C for a column measurement could be used to make comparisons of d13C statistics against. However, we make the argument that most of our results indicate that we are not yet calculating consistent enough results to make this comparison worthwhile. We will therefore propose for future work, once more consistent results have been achieved across the spectroscopic databases, to make comparisons against a total column d13C value.

---

## Author Response (AR3)

Dear Reviewer,

Thank you for your comprehensive review of our paper, below is our response to your review. We have kept your original comments in black, our responses are in blue, and specific actions are indicated in underlined blue.

The work entitled "On the consistency of methane retrievals using TCCON and multiple spectroscopic databases" by Edward Molina et al. treats a highly relevant topic and suits well within the scope of AMT. While the empirical database used (time series of high-resolution FTIR spectra collected at four TCCON sites in different latitudes covering the year 2020) is a sound observational basis for the conducted evaluation of different spectroscopic databases and the overall structure of the paper is appropriate, there are significant shortcomings in the methodological concepts. The manuscript requires major revisions.

Main points of criticism, which need consideration, are:
The presented discussion based on methane dry air molar fractions (DMFs) ignores the fact that this product is based on the ratio of the target gas and molecular oxygen. In order to compensate for spectroscopic uncertainties in both methane and oxygen, the reported DMFs involves empirical airmass independent and airmass-dependent calibrations.
Therefore, when comparing the accurateness of methane band intensities or evaluating spurious air-mass dependencies triggered by imperfect description of pressure-broadening effects in different line lists, the effects introduced by the oxygen (which brings in both a considerable calibration bias and an airmass dependency) and the empirical post-corrections of the GGG processing chain need to be avoided. I believe that for the desired purpose molar fractions should instead be constructed from the column amount of the target gas, the water vapour column and the recorded ground pressure.

The empirical post-correction incorporated in GGG2014 is made to harmonize with the associated oxygen and methane line lists. It is therefore a trivial finding that the GGG2014 line list shows small airmass dependence and that other line lists create a calibration offset. I would expect from a study of the kind presented to demonstrate which methane line list has the best performance (correct calibration of band strengths and minimal SZA dependence of DMF). While the operational GGG2014 results (using the foreseen line lists and the corrections in the post-processing chain) present the best available reference for the true DMF value, the DMF values for the investigation of different line lists and micro windows should be constructed as described above. Note that using this procedure the GGG2014 methane line list can be tested in an equivalent manner to all other line lists versus the operational DMF as target quantity. This would reveal the actual calibration biases of the methane band intensities and identify the methane line list with the most realistic description of pressure broadening effects (which would create the least SZA dependence).

Thank you for this important point. We agree with the reviewer here that using the DMF values do introduce biases from the O2 spectroscopic parameters, and the empirical post-corrections. However, we disagree with the approach of using the surface pressure and water vapour

columns to calculate the VMRs. One of the key benefits of using the O2 column to calculate DMFs is that it is a constant in the atmosphere, meaning that we can directly compare the DMFs calculated from each site. If we calculate VMRs for each site, these are no longer directly comparable, since we introduce biases from the surface pressure measurement, and from the water column calculation (used for calculating the dry air). This means our comparisons would then become dependent on the uncertainties associated with water vapour and surface pressure from each site, which will be larger than O2. Further, the empirical post-correction is applied to all of the retrievals, meaning the applied bias will always be the same, rather than varying as would be the case with VMR calculations.

Fundamentally, although we agree with the point that using O2 introduces biases into our results, we think using VMR would add additional uncertainties that would make the results less valuable. We have now introduced this argument as a point in the discussion of the paper, and we propose an additional assessment using VMRs be caried out.

The other major shortcoming of the work is that the line lists are not investigated species-wise. The study would be of much higher value and more conclusive, if the water vapour line list would be investigated independently from the methane line list. I understand this would require a number of additional retrieval runs, but would be worth the effort as the work then would provide real guidance for selecting both the best methane and water vapour line lists. In its current form, it is difficult to see whether a temperature or humidity dependence is generated by shortcomings of the methane line list itself or whether it is just a consequence of a poor description of interfering water vapour lines (exceedingly important in window 1). I would strongly recommend to include this kind of extension and instead skip the included shallow discussion of methane isotopologue retrievals, which results in the implausible suggestion to use GEISA2020 for this purpose, although this line list on the whole does not appear to be a great choice even for the main isotopologue.

We agree with the reviewer that adding a similar investigation of water vapour (and probably HDO as a separate species) would greatly enhance this study, and is certainly a necessary next step. However, we think this would require an additional paper to describe this work, since this water vapour study would require an assessment as in-depth as that already shown in this paper. In addition to assessing the interfering water vapour lines in the methane windows, this study should also include an assessment of the water vapour windows nominally used in TCCON, which are significantly different to the methane windows. Therefore, since this paper is already of significant length and depth, we feel this study would yield a paper that tries to cover too much ground.

Detailed comments:
In several figures, the data seem to be clipped in a significant manner, so information might be lost (spectral residual in figure 1, bar charts in figures 2 and 7 (impossible to decide whether the bars end or are clipped), time series Figure 5 (I agree to use the same scale for all figures, but too many Tsukuba data points are lost with the current choice)).

Thanks you for this point, we have now modified the y-axes in all of these figures so that no data is clipped.

The implementation of the GGG2020 spectroscopic data is not described properly: are SDV and line-mixing parameters taken into account? While the authors state in section they use the "GGG2014 environment" and state in section 4 that the GGG2020 software uses "non-Voigt line shapes for methane" and could therefore yield "improved or different results", they on the other hand state in section 2.3 that they implemented the qSDHC model and first order line mixing. So I do not see why GGG2020 would generate other results out of the same line list. A proper comparison of the GGG2020 line list versus other lists in the modified GGG2014 environment certainly needs to include the non-Voigt extensions. The description needs to be clarified.

Thank you for highlighting this point of confusion, while GGG2020 does include non-Voigt parameters, the GGG2014 software used in this study was not modified to take advantage of them, the software was only modified to take advantage of the parameters in the SEOM-IAS database. We have now modified the relevant point in section 2.3 and the discussion to highlight the fact that these parameters are not used in this study, and may provide improved results if implemented in the future.

While figure 2 suggests that SEOM-IAS outperforms the other line lists in window 1, this is not supported by figure 1. I understand that figure 1 is just an example, but it would be better to show a typical example (it might be instructive to include in the appendix an equivalent figure for each site).

We agree, we have changed the example plot, and we have identified in Figure 2 (which has changed a little bit to reflect a wider spectra sample) that SEOM-IAS and GGG2020 show more or less equivalent results. We have also included typical spectra from each site in the appendix.

The chosen performance indicators for the spectral fit quality introduce dependence on the SNR of the measured spectra (this propagates also into the DMF indicators due to the variable retrieval error, if I understand correctly). Time series of SNR or at least typical SNR values achieved at each site should be provided. The SNR would be determined from transparent spectral sections in the vicinity of the micro windows (two values might be sufficient, but the SNR in the window 1 region might differ significantly from the other windows).

Thank you for this point, SNR was typically not calculated with GGG2014 as there was no consensus in the TCCON community as how to exactly calculate it for the TCCON retrievals, although we understand that GGG2020 will provide this opportunity. Unfortunately, the output from GGG2014 does not include any of the transparent spectral regions we would need to calculate SNR, and only show the micro-windows applied in the retrieval process. The raw spectra input into GGG2014 are in a binary format, and are there unable to be easily read and understood.

Two aspects of the shown TCCON methane time series seem suspicious: HITRAN2016, GEISA2020, and SEOM-IAS show a high bias with respect to GGG2014 at most sites. Surprisingly, this bias is absent in the Ascension time series. Is there an explanation for this behaviour? The Darwin XCH4 time series shows an uncommented 1% (!) step change to higher values from March onwards. Is this a geophysical signal or a problem with data quality? In addition to methane, the time series of XAIR as a quality indicator should be added as additional panel to the figure for all sites.

There are a number of points to discuss in this comment which we answer in the following numbered items.

1) W.r.t the lack of a bias in the Ascension Island results, we think there are a number of reasons for this. Firstly it is possible that the Ascension Island instrument is less noisy than the other TCCON instruments. Secondly and more likely, the Ascension Island data used in this study unlike the other sites was hand picked for 'golden days' only, so only high quality retrievals were used. In addition, the conditions at Ascension Island hardly vary, while the other sites experience more challenging and varying conditions, which could explain the differences.

2) The Darwin results are a little misleading, we studied this 1% step change carefully and only the first few weeks of March appear to be affected by this change. Which is apparent by the relative lack of variability in the retrieval results when compared to the remainder of the time series. All typical quality control indicators (XAir etc) do not flag this period as being of poor quality, and it is not easily apparent as to why this abnormality exists. Because none of the standard indicators do not suggest anything wrong with these retrievals, we have not filtered them from the results, but we have highlighted it in our updated paper.

3) We have now added XAir as an additional panel for each site as an indicator of the quality. We show XAir for GGG2014, 2020, HITRAN and GEISA2020 as SEOM-IAS does not have any spectral lines in the TCCON XAir retrieval window.

While a linear fit model is certainly appropriate for quantifying the impact of temperature, this probably is much less so for the airmass dependency. It would be good to add figures for each window and line list showing the typical SZA (or airmass) dependency (for clearly showing the effect, the daily data of a station time series would be normalized to a daily reference value at a chosen intermediate SZA typically covered by all sites, then the ratio would be plotted for the complete time series). Such figures would reveal the strength of the effect and inspire an appropriate functional form for fitting the dependency.

Thank you for this important point, we agree that SZA dependency is likely not a linear relationship. In order to further investigate possible fit models we plotted the histograms SZA, using the normalized residual between the retrieved window/spectroscopic database as weights. This helped identified the relative sensitivity of biases to SZA, which we generally found to be more of a second order relationship rather than a first order one. We decided not

to include the histograms into the paper, since we feel this would add numerous figures to an already lengthy paper. We have replaced the original figure 8, with the updated fit models, which clearly indicate non-linear relationships for SZA. Out of curiosity we applied second order fits to the other quantities, and found that in some cases temperature exhibited non-linear relationships. We have therefore included these fits into the updated figure 8 (now figure 9) into the paper as well.

Dear Reviewer,

Thank you for your comprehensive review of our paper, below is our response to your review. We have kept your original comments in black, our responses are in blue, and specific actions are indicated in underlined blue.

This paper details the impact of using 5 different spectroscopic databases (GGG2014, GGG2020, GEISA2020, HITRAN2016, SEOM-IAS) on the retrieval of 12CH4 DMFs in 4 different spectral windows from TCCON measurements. The retrieval is performed with the GFIT retrieval algorithm from the GGG2014 environment. Retrievals are carried out on approximately a year of data at 4 TCCON stations (Ascension Island, Darwin, Ny Alesund, Tsukuba), chosen to have an optimal coverage of different atmospheric and measurement conditions (temperature, water vapour, SZA). A second objective of the paper is also to assess the impact of the spectroscopy and fit windows on the retrieval of 13CH4 (and indirectly, the d13C) and check for consistency of the results.

The scientific work is extensive, with a lot of data processing performed to represent retrievals under various atmospheric and measuring conditions. The presentation and analysis of the data is sometimes a little laborious, since so many different parameters are considered (spectroscopic database, retrieval window, TCCON sites). Still, the paper mostly presents the results adequately and underlines the important results. The results are scientifically relevant and interesting, and are certainly within the scope of AMT.

However, the paper sometimes lacks in style and clarity. It is obvious that it has been through some re-writes and major changes, and it should be proof-read and improved in terms of flow and clarity. I would recommend publication of the paper in AMT pending some minor revisions of the manuscript, as detailed below. Note that I started to note down many typos or problematic sentences in the first half of the paper but was not as systematic in the second half and would recommend to proof-read this more thoroughly.

Thank you for your comments, we have responded to your criticisms in line below.

Comments

1/ The abstract should be improved.

a) The third paragraph is especially difficult to read. It would be helpful to be more quantitative in your description of the impact. The sentence "We also find strong evidence that different windows in different spectroscopic databases exhibit different levels of sensitivity to changing local conditions such as light path length and water vapour" is technically correct, but it is so general it becomes almost meaningless.

b) You should mention the work on the 13CH4 retrieval and d13C determination.

Thank you for this point, we have re-written the abstract to account for your comments a) and b) here.

2/ The introduction should be re-worked to improve clarity and fluidity:

a) I find it strange that a paper studying the impact of the spectroscopy on the retrieval of methane does not directly discuss the spectroscopy aspect of the retrieval in its introduction. Maybe add 1-2 lines to describe how it is relevant?

Thank you for pointing this out, we have now added the following text in the first paragraph to highlight the spectroscopic aspect.

The remote sensing of methane is fundamentally dependent on inferring the concentrations from the absorption of light in the atmosphere at wavelengths unique to methane, otherwise known as spectral lines. The position of these lines for a large number of gases are stored in large databases known as spectroscopic databases. These databases are a considerable source of error in the retrieving methane concentrations in the atmosphere, due to the uncertainty of the position and the magnitude of these spectral lines. Differences in the various available spectroscopic databases could lead to significant differences between satellite estimates of methane. Understanding these differences is an important step towards reducing these uncertainties in future satellite measurements.

b) L44-50 should be at the end of the introduction, detailing the content of the paper. Up until that point, methane isotopologues have not been mentioned at all, maybe add a line about this aspect of the work as well beforehand.

We have now moved this section to the end, and we have introduced the concept of isotopologues further up in the introduction text.

c) I am not certain of the necessity of Table 1. One could either have a table with more information (a graph of the respective instruments and their spectral coverage), or a sentence in the text? Since you discuss TCCON windows shortly thereafter, maybe it would be a good opportunity to describe the TCCON windows used for the CH4 retrieval there (instead of referring to Table 4)?

We have now removed Table 1, and describe the spectral ranges, we think the original spectral fits, and as you identify, the description of TCCON windows in Table 4 (now Table 3) is sufficient to describe the overlap of TROPOMI/UVNS and TCCON. We also introduce the TCCON methane windows in this section.

d) L62-66: Maybe I misunderstood the point being made, but it seems simplistic to suggest that the retrieval window differences between TCCON and TROPOMI may be the major source of biases in the validation of the methane product, especially when many other factors will affect the quality of the retrieval on both GB and satellite platforms. This is surely an interesting point to study, but this seems to inflate the importance of the retrieval window on the validation.

Thank you for this point, however, we do not feel that this section identifies windows differences as the main source of error between TCCON and TROPOMI. We are attempting to highlight that

the 4190-4340 cm-1 is a relatively unexplored region for methane, especially for TCCON, and that we can expect differences because of this. Indeed, one of the GGG developers identified that this spectral region has not been explored in TCCON historically because the spectroscopic parameters were not reliable.

e) L71: "We can infer some of the potential spectroscopic related biases in satellite retrievals…": yes, but since the resolution of TCCON and satellite instruments are quite different, TCCON should be much more affected than satellite retrievals, maybe something to mention.

Agreed, we have added the following sentence.

Note that the spectral resolution of TCCON is typically significantly higher than that of TROPOMI and other satellite instruments, which are unlikely to affected to the same degree as TCCON.

e) L75: I think that Eq. 2 should be moved here. [?]

Agreed, This equation has been moved here.

3/ L119: "GFIT assumes a fixed profile shape for each trace gas, …": you should mention how the a priori is determined

We have now identified the source of the a priori data in the sentence.

4/ L198-201: Is there already an estimate of the errors due to these assumptions? It would be interesting to have some figures on this in the paper.

This is an important point, the original paper describing TCCON (Wunch et al., 2011) does give these estimates in Figure 7. However they are only available for CO2, and are not shown for methane.

5/ Section 3 Before analysing the retrieval data, I would *suggest* to first perform a comparison of synthetic spectra using fixed state vectors with different spectroscopic databases to look at the expected impact in terms of transmissions in various windows. It might also be interesting to look at the contribution of H2O and CH4 to the total transmission, and hence have a better feeling for which windows should be more sensitive to the water content (would help with the analysis of section3.3?)

Thank you for the suggestions in this comment, firstly regarding the idea to investigate synthetic spectra, although this would be interesting, we feel the resources required to adequately describe and analyse the results, for a series of different locations (which is important in terms of the impact of a priori) would be significant.

We have added some figures to analyse how the calculated transmission of 12CH4 and H2O impact the total transmission. For example the figure below shows the % difference in calculated

12CH4 transmission w.r.t. GGG2014. We can see across all bands there are a number of differences, notably in window 1. We have also included an similar assessment for water vapour in the update to the paper. Comparing the figure below to Figure 1 in the original paper, we can see some of the large deviations matching in both figures.

We have updated section 3.1 to include an assessment on the specific transmission differences.

*Ny − Ålesund*

[Figure]

6/ Section 3.1 A slightly more exhaustive analysis of these results would be interesting. Could you associate some of the more prominent residuals with species? For instance, in window 3, HITRAN and GEISA have large discrepancies around 6001 cm-1 compared with GGG, why is that? Are there clear improvements due to the use of non-Voigt parameters?

Please see our response to the point above, where we tackle this issue. Specifically for the 6001 cm-1, we see (according to the figure above) that water vapour differences seem to be the main cause for the discrepancies. The figure above and the other analyses we have done, do not give obvious weight to the improvements of the non-Voigt parameters used in the SEOM-IAS database, apart from the constant lower magnitude RMSE values observed for SEOM-IAS

database in Figure 2. Likely because the 4200-4400 cm-1 spectral region is highly complex, and cannot be directly compared with the 5900-6150 cm-1 spectral region. We can now attribute some of the more characteristic differences, such as the one you mention at 6001 cm-1 for HITRAN and GEISA to water vapour rather than methane.

7/ Section 3.3 In this part of the analysis, you sometimes seem to confuse correlation and sensitivity, which are quite different, or only consider one of these aspects. For instance, on L346 you state that "results from Ascension Island, which operates at lower SZAs than any of the other TCCON sites considered in this study also indicates large correlations, suggesting further complexity". The truth is that the sensitivity (slope) of the linear fit to SZA is close to 0, so despite a large correlation, very little effect is observed.

Thank you for this point, Reviewer #4 pointed out that some of the relationships may not be linear, so we have rewritten this section of the paper, and taken your comments into account. Clearly contrasting the difference between correlation and sensitivity.

On L348-349: "For window 1, there is weak sensitivity to water vapour variations at the Ny-Ålesund and the Darwin sites," where the sensitivity is substantial for Ny Alesund in Window 1, but correlation is weak.

This section has now been re-written.

Section 3.3 could also gain from an indication of statistical significance in the results. It is mentioned on L385 that large p-values were found in some cases, I would suggest identifying clearly results that are not significant, and state at what level you consider these not statistically significant.

Only the GGG2020 values for the Tsukuba site were found to be statistically insignificant according to the p-test. We have therefore removed the GGG2020 Tsukuba results from the updated figures (10, 11 and 12), and have identified in the caption and with a text box in the corresponding section that the p-test was failed for these results. While highlighting that the p-test was passed for all other results.

Typos and technical corrections
Comments with a "[?]" at the end are for your own consideration

L1: "methane retrieving satellites" -> "methane retrieving satellites instruments" [?]

Changed.

L3: "is as important as when TCCON…" -> "is as important now as when TCCON…"

Changed.

L6: the 'S' in "Spectroscopic databases" should not be capitalized

*Changed.*

L7: Leave out spectroscopic databases in "TCCON GGG2014 and GGG2020 spectroscopic databases;", as it is redundant [?]

*Changed to "The spectroscopic databases include those native to TCCON GGG2014 and GGG2020"*

L14: consider changing "~x3" by "up to three times" [?]

*We changed this to "2.5", more in line with the results.*

L16: "vapour. Such" -> the period should be a comma, and the Such should be lowercase. But you should consider changing this sentence as it is very long and uses 5 times the word "different"

*The abstract has now been re-written.*

L24: "bottom up" -> "bottom-up"

*Corrected.*

L25: "top down" -> "top-down"

*Corrected.*

L28: "S5P" -> "Sentinel 5-Precursor (S5P)" , "TROPOMI" -> "TROPOspheric Ozone Monitoring Instrument (TROPOMI)" . These acronyms have been defined in the abstract, but consider adding a proper acronym description in the full paper as well. [?]

*Added.*

L40: "will therefore be relying" -> "therefore rely"

*Corrected.*

L43: "from the 6000 cm-1 spectral range": a range has two values. Either define the window precisely, or reformulate

*Changed to "spectral region".*

L44: "make" -> "perform"

*Corrected.*

L48: "…quantify the variations in retrieval abundances when using five separate spectroscopic databases,…" : this seems like the main aspect of this paper, it should probably be mentioned first.

Thank you, we have now identified how errors can creep in due to different spectroscopic databases in the first paragraph.

L50: "are studies" -> "are studieD"

Thank you, corrected.

L50: consider a reformulation of this sentence

This sentence is now reformulated as:

Building on this assessment, the sensitivity of the retrievals to variations in water vapour concentration and path length are studied. This allows for the assessment of how differing windows and spectroscopic databases are sensitive to variations in local conditions.

L52: (an weighted …" -> "(a weighted…)

Corrected, thank you.

L81: Either state "98.9% and 1.1%", or "99% and 1%", but it should add up to 100%. Considering the "roughly" in the sentence, I would go with 99% and 1%.

Agreed, changed to 99% and 1%.

L86: "structured as follows," -> change the comma to a colon

Corrected to colon, thank you.

L90: "…conclusions are shown…" -> "conclusions are drawn" [?]

Agreed, changed.

L100: Add a comma after "In this study"

Added.

L100-101: "which is summarised briefly here." -> "briefly summarised here."

Changed, thank you.

L101: "A forward" -> lowercase "a"

Changed, thank you.

L104: "fit" -> "fitted" [?]

Changed, thank you.

L105: after "In the case of GFIT", add a comma

Inserted, thank you.

L106-115: I am not convinced that these are essential to the papers and may be omitted. Maybe reconsider listing these, and put a reference instead [?]

We disagree with the reviewer here, these state vector elements are not typically listed, and are different from what is typically used in a satellite retrieval which we wish to emphasis. In addition, we wish to further emphasis the use of the 'continuum curvature' fit element which is not standard, even for TCCON.

L116: confusing wording: "Note that not all of the above are not routinely…" I imagine that you should omit the second "not".

Good catch, thank you, the 'not' has been removed.

L116: Start a new sentence with "For example" [?]

We have replaced 'for example' with 'especially' which we think fits better.

L124: "the 7885 cm-1 spectral range multiplied" -> a spectral range needs 2 numbers. Maybe use a formulation such as "in the vicinity of 7885 cm-1…" [?]

Corrected to spectral region.

L127: Not sure if it's necessary to state the resolution again, which was already mentioned on L57

Agreed, reference has been removed.

L131-132: not sure if this sentence needs commas after "window" and "study" or if they are optional [?]

Added a comma after "window", thank you.

Table 4 has a beginning o f a sixth row

We are not sure why this happened, however we have changed the format of the table, and this additional row no longer appears.

L135: is there a reference for the "standardised process"?

We have included a reference to the TCCON wiki page, however this requires an account and log-on. There is no other available documentation for this process.

L137: this sentence should not be its own paragraph L158: "Further to exploring the…" -> maybe "Beyond exploring …" [?]

Agreed, changed to "Beyond exploring".

L167: "line mixing" -> "line-mixing"

Corrected.

L169: "They find a 1.1% difference in total methane…" -> is it a positive or negative bias? Maybe just add a "+" to emphasize [?]

It's a positive bias, a "+" has been added.

L184: "standard methane window" -> either "standard methane windows" (there are 3 of them, right?) or, better yet "standard TCCON CH4 product"

Agreed, changed to "standard TCCON CH4 product".

L185-189: These are technically correct sentences but are a bit long and wordy. Maybe consider adding equations ? [?]

Agreed, we have replaced these sentences as equations.

L194: This sentence should not start a new paragraph, but simply continue the previous one.

We have removed the paragraph break.

L196: there should not be a period before the "2)". Maybe a semi-colon?

The period has be replaced with a semi-colon.

L197-198: "apriori" or "a priori" -> be consistent, I have seen both in the paper

We double checked this, and are now consistent.

L202: "cross sections" -> "cross-sections"

Corrected.

L204-207: consider re-writing this sentence, it's a bit convoluted. Maybe with more than one sentence?

We have re-written this sentence as follows:

These dependencies are quantified by non-linear regression analysis, consisting of fitting the variations of water vapour, SZA and measured temperature against the normalised difference between each methane isotopologue DMF case and the DMFs from the standard TCCON methane retrieval window. Here the normalisation factor is the uncertainty from the standard TCCON methane retrieval.

L208-209: you probably should not hyphenate an acronym L

We don't see this acronym, TCCON is hyphenated, but likely because the text is wrapped around.

213: leave out "metric"

Removed.

L214: you repeat "varies in the total"

Thank you, removed.

L216: "<< This sentence is unclear (especially the second part: "through the fact…"). Also, it should be "long-term", not "long term".

This sentence has now been re-written as follows:

However, TCCON currently represents the best chance of remotely measuring d13C, since precision errors are low and SNR is high Wunch et al ., (2011). In addition, because TCCON sites are situated in fixed positions long-term averaging is possible, which further reduces precision based errors.

Figure 1: Would it be possible to plot or mention the instrumental uncertainty along the residuals?

Individual error contributions are not provided by GGG2014, and it is not possible to identify them until trace gas uncertainty is provided.

L238: "shown in Fig 2," -> "presented in Fig 2," [?]

Changed, thank you.

L239: "Fig 2 is the fit" -> "Fig 2 is that the fit"

Changed, thank you.

L247: Maybe start a new paragraph with "There are differences in the…" as you focus more on the TCCON sites now, not on the windows?

Agreed, changed.

L256: "Likely reasons" -> for what? If you refer to the last paragraph, you should stay in that paragraph, and state more clearly what you mean.

We placed this section listening the reasons into the previous paragraph, and rephrased to, Likely reasons for this difference are.

Figure 2: Maybe indicate in one plot (window4?) the number of spectra used for the statistics? I know this information is available in Table 2, but this is always nice to have this information on hand.

We don't actually use all available data to generate the statistics in this plot (largely because this caused problems with my machine). We therefore generated these statistics from a subset of 500 measurements from each site, this is now indicated in the figure caption.

Figure 2 caption: maybe indicate the colour of each statistics in parenthesis: "Each subplot shows the RMSE (black) and chi2 values (blue) …"

Agreed, added.

L263+264: "fit" -> "fitted" [?]

Changed.

L263: "there no specific" -> "there are no specific"

Corrected.

L267: "GESIA" -> "GEISA"

Corrected.

L270: "the the" -> "that the"

Corrected.

L270: "retrievals each database" -> "retrievals for each database"

Corrected.

L274-275: This is not a complete sentence, there seems to be missing something (maybe Firstly we present the results?)

Good catch thank you, changed to "Firstly we analyse the results"

L289: "from in" -> delete "in"

Deleted.

L290: "the standard in" -> "the standard retrieval in"

Changed.

L304: "sides" -> "sites"

Corrected.

L327-330: This sentence encapsulates nicely the differing atmospheric conditions of the various TCCON stations. Maybe it should be moved to the introduction to better explain this part of the work [?]

Agreed, we have moved this sentence to section 2.1.

Figure 8: This figure is not very easy to read, but I am not quite sure how to improve it. It does give a good overview of the results. Maybe instead it could be a pcolor-type graph for the slope of the linear fit and the intercept for each station and parameter sensitivity (each row of the graph could be a different spectroscopic database, each column a different window?). Just a suggestion

We agree that the figure is challenging, however we think the following figure (now figure 10) captures the type of information the referee suggests here. We feel the point of figure 8 (now figure 9) gives a qualitative impression of the sensitivity of each database/window to variations in local conditions, and therefore remains useful. Especially now non-linear regression is used for fitting models, we see clear differences between non-linear sensitivities and linear sensitivities.

Figure 8 legend: "GEISA", not "GESIA"

Corrected.

L334: The description of Figure 8 is really not clear. Could you be more precise in what is being shown in the figure instead of "the qualitative distributions"?

We have added the following text to explore this figure in more detail.

Figure 9 qualitatively describes the nature of the sensitivity of each TCCON site/database/window to variations in local conditions. For Ny-Ålesund (row 1), there is a

mixture of non-linear and linear sensitivities to variations in water vapour and SZA. Windows 2, 3 and 4 for GEISA2019 indicate particularly significant non-linear sensitivities to SZA variations. Sensitivities to temperature variation are generally linear, although some indications of slight non-linear behaviour are apparent (GEISA2020). For Ny-Ålesund there are some cases where little sensitivity is observed, e.g. HITRAN, suggesting a wide range of responses in the databases/windows. In contrast to Ny-Ålesund, Darwin (row 2) shows limited sensitivity to local condition variations, with low magnitude linear gradients observed for most cases. There are some exceptions, notably HITRAN window 3 and GEISA2020 windows 3 and 4 in relation to SZA variations, were significant non-linear behaviour is observed. Tsukuba (row 3) again shows significantly different behaviour, with almost all databases/windows showing significant linear or non-linear sensitivity. Window 1 for SEOM-IAS, GGG2020, HITRAN and GEISA indicate significant negative linear relationships, with all other cases show a range of sensitivity. For variations in SZA, as with Ny-Ålesund and Darwin, HITRAN window 3 and GEISA2020 windows 3 and 4 suggest strong non-linear sensitivity to variations in SZA. Most of the other windows/databases indicate some linear/non-linear sensitivity, but not to the same degree as HITRAN window 3 and GEISA2020 windows 3 and 4. Temperature variations for Tsukuba indicate significant non-linear sensitivity for window 1 in most cases (except GGG2014), and in general show different results from those shown in Ny-Ålesund and Darwin. Finally for Ascension Island, we note almost no sensitivity to any local condition variation, except for HITRAN window 3 and GEISA2020 windows 3 and 4 with SZA variations, which have shown sensitivity in all cases.

L342: "indicate the retrieval" -> "indicate that the retrieval"

We have changed this section of the text, and this does not exist in the new text.

L345: "creep" -> maybe find a better word [?]

Changed to propagate.

L349: "opposite databases" -> what are opposite databases?!

This has been removed from the text.

L362: "vapour. Thus" -> should be in the same sentence "vapour, thus"

These words have been removed from the text in the updated paper.

L396: "in the calculation" -> its more in the analysis of the results than the actual calculation of the value…

Agreed, calculation has been changed to analysis.

Table 5: The table should be slightly improved in such a way that there is less text. Maybe use subrows for each station, one for each database (basically like it's done now, but just mention the database name once, not in each cell…). Also, would it be possible to either add the uncertainty

on the d13C value, or mention an order of magnitude in the text? This would add important information to the discussion of L398-401 about the uncertainty.

The table has now been split into two, and modified to include less text.
W.r.t the uncertainty, we discuss the retrieved uncertainty of 13CH4 in more detail, and come to the conclusion the precision errors are not the limiting factor in calculating d13C. The sentences indicated have been updated, as shown below.

Tables 4 and 5 indicates a wide range of results, suggesting either significant differences in spectroscopic parameters or large retrieval uncertainty. GGG identifies the mean uncertainty of 13CH4 retrievals to between 0.5 - 2 ppb (~2.5-10%) depending on the database and TCCON site. However, given that these uncertainties can be averaged over a long period of time, they should reduce significantly (by $\sim$ x200 in the case of Darwin), meaning that the precision of 13CH4 retrievals should be very high (e.g. <0.006 ppb). Therefore precision errors cannot explain the differences in d13C values shown in Tables 4 and 5, meaning differences in the spectroscopic databases are the key sources of errors in 13CH4 retrievals. This therefore suggests that knowledge of 13CH4 retrievals spectroscopic parameters must be improved before serious attempts at remote sensing of 13CH4 can be made.

L398: "that we calculate the mean uncertainty" -> "that the mean uncertainty…is between…"

Changed.

L404: "show results" -> "are"

Corrected.

L407: "combination" -> "combinations" [?]

Corrected.

L420: "Further to this" -> "Furthermore" [?]

Corrected.

L447: "question that is of some interest to the community, "can we calculate realistic and constant 13C values from TCCON"." -> this formulation is a bit strange to me, maybe use colon after community (instead of a comma), or change the formulation to something like "question that is of some interest to the community, namely whether it is possible to calculate realistic and constant 13C values from TCCON."

We have reformulated this sentence as follows.

this study touches on a question that is of some interest to the community, namely whether it is possible to calculate realistic and constant d13C values from TCCON.

L453: "calculate" -> "calculated"

Corrected.

L459: "this is an assumption based" -> "it is based"

Corrected.

References:

[revised manuscript text omitted]

---

## Author Response (AR4)

Dear Dr Janssen,

Thank you for your thorough efforts on this paper to date, for this response we have kept your comments in black, and our changes in blue. We have also re-read the paper again to try and remove any stray errors.

Dear authors,

I am happy to announce that all referees recommend your article for publication. Unfortunately, the current version does still contain too many glitches and phrases that are difficult to understand. I therefore ask you to correct the following list of remarks and suggestions. Please highlight any changes with respect to the current version. Since there might be oversights from my part, I also would like to ask you to carefully reread your corrected manuscript for further errors. Once you have updated the manuscript, the article can proceed to the next stages.

With kind regards

Christof JANSSEN

L 17 : suggesting that further

'That' added.

L 20 : verb missing : 'indicates different' instead of 'indicating differing'

Changed to 'indicates different'.

L 24 : Delete phrase With the aim of

Deleted.

L 25 : ‰ or % ?

This should be 100‰, changed to this.

L 27 : What does TEXT mean ?

Apologies, thought this was a section dealt with by the copy/edit team. The copyright statement has been removed.

L 79 : similar studies in the past

Changed.

L 81 : replace 'seasons. We' by 'seasons we'

Changed to 'seasons, we'.

L 84 : write 'unlikely to be affected at the'

Changed to 'unlikely to be affected to the'.

L 87 : replace '(see Eq. 1)' by 'which is defined as' and move Move eq 1 directly behind

Done.

L 94 : Replace 'This is a quantity that has been used' by 'd13C has been used'

Changed.

L 94 : Do you mean global studies ? If not, than I would prefer to remove this word here.

Globally has been removed.

L 127 : specify 'variations' as 'variations (standard deviations)'

Changed.

T 2 : For clarity, please replace ±sigma with ± 1 sigma

Changed.

T 2 : Check unities. They are missing on the sigma values. The best is to add '/ ppmv', etc. in the table header

Changed.

L 149 : remove 'attempt to'

Removed.

L 156 : Phrase starting with 'Combined' is incomplete. Join phrases ending and starting in this line

Replaced 'Combined' with 'These are adjusted with'

L 168 : remove 'below' as Tables can flaot.

Removed.

L 172 : Correct grammar. A verb is missing in subordinate phrase starting with 'where'

This sentence has been replaced with the following "TCCON methane products are the result of a standardised process, where the final reported values are calculated from a weighted average of three retrieved values from windows 2, 3 and 4 as described in Table 3".

L 177 : The database included with the updated GGG2020 software, referred to in this study as GGG2020, which includes numerous updates to the GGG2014 spectroscopic parameters. -> The database included with the updated GGG2020 software which includes numerous updates to the GGG2014 spectroscopic parameters and is referred to as GGG2O2O in this study.

Changed as suggested.

L 180 : 2020 is the current release, write HITRAN2016 release instead of current release HITRAN2016

The reference to the current release has been removed.

L 182 : lines shapes -> line shapes

Corrected.

L 189 : differentiate with them -> differentiate between them

Corrected.

L 191 : Some work has been performed previously comparing spectroscopic databases -> Some work on comparing spectroscopic databases has been performed previously
Changed.

L 202 : parameters, the remaining -> parameters. The remaining

Changed as suggested.

L 205 : speed dependent -> speed dependance

Corrected,

L 206 : Please indicate the spectral bands used by Mendonca et al. The utility of using these parameters likely depends on a particular band.

Add the following clarification "(albeit in the 5880-6145 cm-1 spectral region)"

L 210 : We note the -> We note that the

Corrected.

L 217 : Make sure the equation terminates wih a semicolon, not with a full stop.

Changed.

L 218 : Remove full stop.

Changed.

L 221 : Please sepell out w.r.t.

All instances of w.r.t. have been spelled out.

L 229 : impact -> impacts

Changed.

L 233 : parameters 1) -> parameters: 1)

Changed.

L 236 : all trace gas fitting is unclear (fiting of all trace gases ?). Please rephrase

Changed to "The GFIT retrieval algorithm is a scaling retrieval algorithm", to simplify.

L 240 : Please refer to line strengths or line intensities and not cross sections

Changed to 'line strengths'.

L 250 : use 'on these bias and precision values'.

Changed.

L 251 : instead does not seem to make any sense her. You could use 'still' here

We removed the 'instead' it is unnecessary.

L 252 : Remove Fundamentally

Removed.

L 254 : Insert full stop after '2018, 2019)' and use capital I in in-situ

Changed.

L 258 : Please explain 0.02 ppb.

Added the following at the end of the relevant sentence "calculated using Eq. 1 (Malina et al., 2018).

L 278 : is more complex region -> covers a more complex region.

Changed.

L 296 : Start new phase before 'this is likely'

Changed to "results. This is likely because".

Figure 2 caption. Remove blue colour.

Removed.

L 303 : write out w.r.t.

As the similar comment above.

L 305 : Firstly without a secondly is considered bad style.

Firstly has been removed.

Please replace further in line 307.

Removed.

L 314 & 315 : remove all commas

Removed.

L 323 : Write Fig. 8

Done.

L 323 : Phrase unclear. Do you mean : We also find that the Xair value indicates the retrieval quality with only ... ?

Sentence changed to the following structure "We also find the retrieved XAir values indicate good quality retrievals, with only a small number falling outside the acceptable range."

L 325 : see comments on L323 & 303

Changed.

L 326 : Norm Abs Mean has been abbreviated and defined differently in Eq 4. Please use consistent notations (NAmean)

Changed to NAM, as suggested by the comment below.

L 327 : This has been sais before. It is sufficient to write '... reference retrievals. Thus we assume any ...'

Removed extraneous text.

L 330 : Use consistent notation Norm Abs Mean. I propose to use NAM throughout, but corect Eq 4 as well.

NAM is now used through the whole paper.

L 371 : write 'knowledge of spectroscopic parameters in window 1 is not as good as in other windows'

Changed.

L 375 : replace 2017), however these' by '2017). However, these'

Changed.

L 377 : Replace 'The implication being' by 'This implies'

Changed.

L 389 : 'We therefore fit' should start a new phrase.

Changed the sentence to "To account for non-linearity, we fit the normalised residual DMF values with a second order model, to expose any potential non-linearity. "

L 407 : 'and 12. Where' -> 'and 12 where'

Changed.

L 411 : 'to be the presence' -> 'to indicate the presence'

Changed.

L 411 : 'First considering' -> 'We first consider'

Changed.

L 414 : relationship in singular

Changed.

L 415 : add missing 'of'

Added.

L 423 : w.r.t.

As above comment.

L 425 : shows -> show

Changed.

L 427 : much larger than for/in any other case.

Changed.

L 427 : write 'especially in the Ny-Alesund retrievals'

Changed.

L 431 : Fig.

Changed.

L 434 : The sentence starting with 'with' is gramatically incomplete.

Sentence changed to "The windows and spectroscopic databases all show similar results, with no clear 'winner' or 'loser'".

L 439 : remove variation

Removed.

L 444 : Rephrase, eg Given that both Tsukuba and darwin have large variations …., it is interesting that …

Sentence changed to "Given both Tsukuba and Darwin both have large variations in background water vapour, this is an interesting result".

L 457 : suggests -> suggest

Changed.

L 481 : biases, the -> biases. The

Changed.

L 482 : results, however -> results. However,

Changed.

L 483 : The other sites, -> The other sites

Changed.

L 486 & 487 : Three consecutive uses of 'clear'. Please improve

Sentence changed to "In general, there is no obvious case of one window or database showing increased sensitivity over and above any of the others (although Ascension Island is typically less sensitive).".

L 492 : We note when -> We note that when

Changed.

L 493 : indicating these results -> indicating that these results

Changed.

L 494 : Start new phrase efter combinations.

Done.

L 497 : write out w.r.t.

As comments above.

L 500 : the expected result. -> expected.

Changed.

L 502 : Eq 1), -> Eq 1)

Changed.

L 505 : write out wrt

As comments above.

L 507 : Tables 4 and 5 indicate

Changed.

L 508 : Better write 'GGG yields a mean uncertainty of 13CH4 retrievals between ...'

Changed.

L 510 : by ~x200 -> by about 200 times

Changed.

L 513 : delete retrievals

Removed.

L 513 : at remote -> of remote

Changed.

L 515 : , considering -> and considering

Changed.

L 516 : The phrase is confusing. Please restructure

Phrase changed to "these combinations yield surprisingly consistent results across all sites and windows"

L 519 : (at least 2x any other -> (at least twice than for any other

Changed.

L 537 : Reconsider the phrase starting with 'Overall' as it is not clear.

Sentence changed to "Overall the results in Tables 4 and 5 suggest GEISA2020 has the most consistent 13CH4 retrievals across all windows and sites, and relatively low bias levels."

L 539 : , however, Window -> . Window 4, however,

Changed.

L 542 : Remove 'in this paper' (we are not at the conclusion of the paper yet) and terminate the phrase after biases.

Removed, and changed.

L 543 : however it -> However it

Changed.

L 544 & 545 : you probably meant to write 'due to simultaneous variation of a number of ...', Delete simultaneously

Deleted.

L 594 : delete in each case. This seems redundant.

Agreed, removed.

L 610 : analysed is misspelled

Corrected.

L 610 : Terminate phrase after databases.

Changed.

L 603 : too many commas

Changed.

L 607 : write out w.r.t.

As comments above.

L 610 : suggesting that TCCON ….

Changed.

L 611 : bias to -> bias with respect to

Changed.

L 614 : suggest the GEISA -> suggests that the GEISA

Changed.

L 620 : show – shows

Changed.

L 620 : write out w.r.t

As comments above

L 623 : remove comma

Removed.

L 624 : Raplace coma by full stop.

Replaced with semi-colon, which seemed more appropriate.

L 627 : Please rephrase. What do you want to say here ? What is a unique formation ?

Changed sentence to "Suggesting that the weightings normally used to generate TCCON methane products should depend on TCCON site and season."

---

## Author Response (AR5)

Dear Dr Janssen,

Thank you for your efforts; we respond to your final changes in blue, and keep your original comments in black.

Dear Authors,

Thank you for complying with the suggested changes. I would like to congratulate for your manuscript finally being accepted for publication in AMT. Please prepare the documents for the editorial team by applying the four following technical corrections:

L 92 : move the phrase "where VPDB refers to Vienna Pee Dee Belemnite, an international reference standard for 13C assessment." directly behind eq 1.

Text has been moved to directly behind eq 1.

Table 2 : Replace ° by °C

Replaced.

L 224 : Remove the whole line.

Removed.

L 504 : remove the Guillemet before the < sign.

Removed.